# Cluster-aware Semi-supervised Learning: Relational Knowledge Distillation Provably Learns Clustering

**Yijun Dong**[*]
Courant Institute of Mathematical Sciences
New York University
New York, NY
yd1319@nyu.edu

**Kevin Miller**[*]
Oden Institute for Computational
Engineering & Science
University of Texas at Austin
Austin, TX
ksmiller@utexas.edu

**Qi Lei**
Courant Institute of Mathematical Sciences
& Center of Data Science
New York University
New York, NY
ql518@nyu.edu

**Rachel Ward**
Oden Institute for Computational
Engineering & Science
University of Texas at Austin
Austin, TX
rward@math.utexas.edu

## Abstract

Despite the empirical success and practical significance of (relational) knowledge distillation that matches (the relations of) features between teacher and student models, the corresponding theoretical interpretations remain limited for various knowledge distillation paradigms. In this work, we take an initial step toward a theoretical understanding of relational knowledge distillation (RKD), with a focus on semi-supervised classification problems. We start by casting RKD as spectral clustering on a population-induced graph unveiled by a teacher model. Via a notion of clustering error that quantifies the discrepancy between the predicted and ground truth clusterings, we illustrate that RKD over the population provably leads to low clustering error. Moreover, we provide a sample complexity bound for RKD with limited unlabeled samples. For semi-supervised learning, we further demonstrate the label efficiency of RKD through a general framework of cluster-aware semi-supervised learning that assumes low clustering errors. Finally, by unifying data augmentation consistency regularization into this cluster-aware framework, we show that despite the common effect of learning accurate clusterings, RKD facilitates a "global" perspective through spectral clustering, whereas consistency regularization focuses on a "local" perspective via expansion.

## 1 Introduction

The immense volume of training data is a vital driving force behind the unparalleled power of modern deep neural networks. Nevertheless, data collection can be quite resource-intensive; in particular, data labeling may be prohibitively costly. With an aim to lessen the workload associated with data labeling, semi-supervised learning capitalizes on more easily accessible unlabeled data via diverse approaches of pseudo-labeling. For instance, data augmentation consistency regularization [Sohn et al., 2020, Berthelot et al., 2019] generates pseudo-labels from the current model predictions on carefully designed data augmentations. Alternatively, knowledge distillation [Hinton et al., 2015] can be leveraged to gauge representations of unlabeled samples via pretrained teacher models.

---

[*]Equal contribution.

37th Conference on Neural Information Processing Systems (NeurIPS 2023).

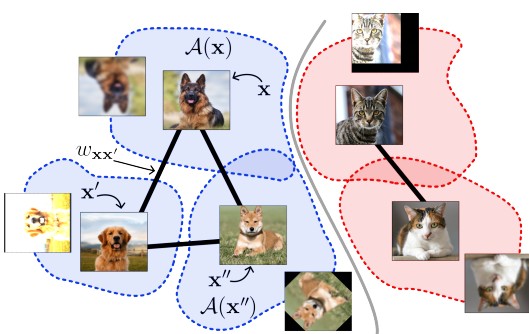

Figure 1: The complementary perspectives of RKD and DAC regarding the data. RKD learns the pairwise relations among data (*e.g.*, edge $w_{\mathbf{xx}'}$ that characterizes the similarity between $\mathbf{x}$ and $\mathbf{x}'$) over the population "globally" via spectral clustering, as illustrated in Section 4. Alternatively, DAC discovers the "local" clustering structure through the expansion of neighborhoods based on overlapping augmentation sets (*e.g.*, $\mathcal{A}(\mathbf{x}) \cap \mathcal{A}(\mathbf{x}'') \neq \emptyset$), as elaborated in Section 6.

Meanwhile, as the scale of training state-of-the-art models explodes, computational costs and limited data are becoming significant obstacles to learning from scratch. In this context, (data-free) knowledge distillation [Liu et al., 2021, Wang and Yoon, 2021] (where pretrained models are taken as teacher oracles without access to their associated training data) is taking an increasingly crucial role in learning efficiently from powerful existing models.

Despite the substantial progresses in knowledge distillation (KD) in practice, its theoretical understanding remains limited, possibly due to the abundant diversity of KD strategies. Most existing theoretical analyses of KD [Ji and Zhu, 2020, Allen-Zhu and Li, 2020, Harutyunyan et al., 2023, Hsu et al., 2021, Wang et al., 2022] focus on feature matching between the teacher and student models [Hinton et al., 2015], while a much broader spectrum of KD algorithms [Liu et al., 2019, Park et al., 2019, Chen et al., 2021, Qian et al., 2022] demonstrates appealing performance in practice. Recently, *relational knowledge distillation* (RKD) [Park et al., 2019, Liu et al., 2019] has achieved remarkable empirical successes by matching the inter-feature relationships (instead of features themselves) between the teacher and student models. This drives us to focus on RKD and motivates our analysis with the following question:

*What perspective of the data does relational knowledge distillation learn, and how efficiently?*

**(RKD learns spectral clustering.)** In light of the connection between RKD and instance relationship graphs (IRG) [Liu et al., 2019][2], intuitively, RKD learns the ground truth geometry through a graph induced by the teacher model. We formalize this intuition from a spectral clustering perspective (Section 4). Specifically, for a multi-class classification problem, we introduce a notion of *clustering error* (Definition 3.1) that measures the difference between the predicted and ground truth class partitions. Through low clustering error guarantees, we illustrate that RKD over the population learns the ground truth partition (presumably unveiled by the teacher model) via *spectral clustering* (Section 4.1). Moreover, RKD on sufficiently many unlabeled samples learns a similar partition with low clustering error (Section 4.2). By appealing to standard generalization analysis [Wainright, 2019, Bartlett and Mendelson, 2003, Wei and Ma, 2019] for unbiased estimates of the parameterized RKD loss, we show that the *unlabeled sample complexity of RKD* is dominated by a polynomial term in the model complexity (Theorem 4.2, Theorem 4.3).

As one of the most influential and classical unsupervised learning strategies, clustering remains an essential aspect of modern learning algorithms with weak/no supervision. For example, contrastive learning [Caron et al., 2020, Grill et al., 2020, Chen and He, 2021] has proven to learn good representations without any labels by encouraging (spectral) clustering [HaoChen et al., 2021, Lee et al., 2021, Parulekar et al., 2023]. However, the effects of clustering in the related schemes of semi-supervised learning (SSL) are less well investigated. Therefore, we pose a follow-up question:

*How does cluster-aware semi-supervised learning improve label efficiency and generalization?*

---

[2]An IRG is a graph with samples as vertices and sample relationships, characterized by their mutual distances in a feature space, as edges. RKD encourages the student model to learn such IRGs from the teacher model.

**(Cluster-aware SSL is label-efficient.)** We formulate a cluster-aware SSL framework (Section 5) where assuming a low clustering error (Definition 3.1) endowed by learning from unlabeled data, the label complexity scales linearly in the number of clusters, up to a logarithmic factor (Theorem 5.1).

Since clustering is a rather generic notion that can be facilitated from various perspectives, to better understand spectral clustering as learned by RKD, we compare the clustering mechanism of RKD with that of another common example of cluster-aware SSL—data augmentation consistency (DAC) regularization, while exploring the following question:

*What are the similarities and discrepancies between different cluster-aware strategies?*

**(RKD and DAC learn clustering from complementary perspectives.)** We unify the existing analysis [Yang et al., 2023] tailored for DAC regularization [Sohn et al., 2020] into the cluster-aware SSL framework (Section 6). In particular, while facilitating clustering similarly to RKD, DAC reduces the clustering error through an expansion-based mechanism characterized by "local" neighborhoods induced by data augmentations (Theorem 6.2). This serves as a complementary perspective to RKD which approximates spectral clustering on a graph Laplacian that reflects the underlying "global" structure of the population (Figure 1).

## 2 Related Works

**Relational knowledge distillation.** Since the seminal work of Hinton et al. [2015], knowledge distillation has become a foundational method for time and memory-efficient deep learning. Recent years have seen different KD strategies. Some works learn to directly match the output (response) [Hinton et al., 2015, Chen et al., 2017, Ba and Caruana, 2014] or intermediate layers (features) [Romero et al., 2014, Zagoruyko and Komodakis, 2016a, Kim et al., 2018] of the teacher network. More recently, Park et al. [2019] introduced the idea of relational knowledge distillation (RKD), and concurrent work [Liu et al., 2019] introduced the "instance relationship graph" (IRG). They essentially presented similar ideas to learn the inter-feature relationship between samples instead of matching models' responses or features individually. A comprehensive survey of relational (relation-based) KD can be found in [Gou et al., 2021].

Despite the empirical success stories, the theoretical understanding of KD remained limited. Prior theoretical work is mostly constrained to linear classifier/nets [Phuong and Lampert, 2019, Ji and Zhu, 2020] or neural tangent kernel (NTK) analysis [Harutyunyan et al., 2023]. Hsu et al. [2021] leverages knowledge distillation to improve the sample complexity analysis of large neural networks with potentially vacuous generalization bounds.

**Data augmentation consistency regularization.** Data augmentation serves as an implicit or explicit regularization to improve sample efficiency and avoid overfitting. Initially, people applied simple transformations to labeled samples and add them to the training set [Krizhevsky et al., 2017, Simard et al., 2002, Simonyan and Zisserman, 2014, He et al., 2016, Cubuk et al., 2018]. Traditional transformations preserve image semantics, including (random) perturbations, distortions, scales, crops, rotations, and horizontal flips. Subsequent augmentations may alter the labels jointly with the input samples, e.g. Mixup [Zhang et al., 2017], Cutout [DeVries and Taylor, 2017], and Cutmix [Yun et al., 2019]. Theoretical works have demonstrated different functionalities of data augmentation such as incorporating invariances [Mei et al., 2021], amplifying strong features [Shen et al., 2022], or reducing variance during the learning procedure [Chen et al., 2020].

Recent practices explicitly add consistency regularization to enforce prediction or representation similarities [Laine and Aila, 2016, Bachman et al., 2014, Sohn et al., 2020]. Not only does this way incorporates unlabeled samples, but Yang et al. [2023] also theoretically demonstrated other benefits: explicit consistency regularization reduces sample complexity more significantly and handles misspecification better than simply adding augmented data into the training set.

**Graph-based learning.** Although we leverage graphs to model the underlying clustering structure of the population, the RKD algorithm that we study applies to generic data distributions and does not involve any graph constructions over the training data. The latter is crucial for modern learning settings as graph construction can be prohibitively expensive on common large-scale datasets.

Nevertheless, assuming that explicit graph construction is affordable/available, there exists a rich history of nonparametric graph-based semi-supervised classification models for transductive learn-

ing [Zhu et al., 2003, Bertozzi and Merkurjev, 2019, Zhou et al., 2003, Belkin et al., 2004, Belkin and Niyogi, 2004]. These methods have seen success in achieving high accuracy results in the low-label rate regime [Calder et al., 2020, Bertozzi and Merkurjev, 2019]. Furthermore, assuming inherited graph structures of data, recent progress in graph neural networks (GNN) [Zhou et al., 2020] and graph convolutional networks (GCN) [Welling and Kipf, 2016] has provided a connection between these classical graph-based methods and powerful deep learning models for a variety of problem settings [Zhou et al., 2020].

# 3 Problem Setup

**Notations.** For any event $e$, let $\mathbb{1}\{e\} = 1$ if $e$ is true, and $\mathbb{1}\{e\} = 0$ otherwise. For any positive integers $n, K \in \mathbb{N}$, we denote $[n] = \{1, \ldots, n\}$ and $\Delta_K \triangleq \left\{ (p_1, \ldots, p_K) \in [0,1]^K \,\middle|\, \sum_{k=1}^K p_k = 1 \right\}$. For a finite set $\mathcal{X}$, $|\mathcal{X}|$ denotes the size of $\mathcal{X}$, and $\mathrm{Unif}(\mathcal{X})$ denotes the uniform distribution over $\mathcal{X}$ (*i.e.*, $\mathrm{Unif}(\mathbf{x}) = 1/|\mathcal{X}|$ for all $\mathbf{x} \in \mathcal{X}$). For a distribution $P$ over $\mathcal{X}$ and any $n \in \mathbb{N}$, $P^n$ represents the joint distribution over $\mathcal{X}^n$ such that $\{\mathbf{x}_i\}_{i \in [n]} \sim P^n$ is a set of $n$ *i.i.d.* samples. Given any matrix $\mathbf{M} \in \mathbb{R}^{n \times k}$ ($n \geq k$ without loss of generality), let $\sigma(\mathbf{M}) = \{\sigma_i(\mathbf{M}) \mid i \in [k]\}$ be the singular values of $\mathbf{M}$ with $\sigma_1(\mathbf{M}) \geq \cdots \geq \sigma_k(\mathbf{M})$. While for any symmetric $\mathbf{M} \in \mathbb{R}^{n \times n}$, let $\lambda(\mathbf{M}) = \{\lambda_i(\mathbf{M}) \mid i \in [n]\}$ be the eigenvalues of $\mathbf{M}$ such that $\lambda_1(\mathbf{M}) \leq \cdots \leq \lambda_n(\mathbf{M})$. Given any labeling function $y : \mathcal{X} \to [K]$, let $\overrightarrow{y} : \mathcal{X} \to \{0,1\}^K$ be its one-hot encoding.

## 3.1 Spectral Clustering on Population-induced Graph

We consider a $K$-class classification problem over an unknown data distribution $P : \mathcal{X} \times [K] \to [0,1]$ (while overloading $P$ for the probability measure over $\mathcal{X}$ without ambiguity). Let $y_* : \mathcal{X} \to [K]$ be the ground truth classification that introduced a natural partition $\{\mathcal{X}_k\}_{k \in [K]}$ where $\mathcal{X}_k \triangleq \{\mathbf{x} \in \mathcal{X} \mid y_*(\mathbf{x}) = k\}$ such that $\bigcup_{k=1}^K \mathcal{X}_k = \mathcal{X}$ and $\mathcal{X}_k \cap \mathcal{X}_{k'} = \emptyset$ for all $k \neq k'$. Meanwhile, we specify a function class $\mathcal{F} \subset \{f : \mathcal{X} \to \mathbb{R}^K\}$ to learn from where each $f \in \mathcal{F}$ is associated with a prediction function $y_f(\mathbf{x}) \triangleq \mathrm{argmax}_{k \in [K]} f(\mathbf{x})_k$.

We characterize the geometry of the data population[3] $\mathcal{X}$ with an undirected weighted graph $G_\mathcal{X} = (\mathcal{X}, \mathbf{W}(\mathcal{X}))$ such that: (i) For every vertex pair $\mathbf{x}, \mathbf{x}' \in \mathcal{X}$, the edge weight $w_{\mathbf{x}\mathbf{x}'} \geq 0$ characterizes the similarity between $(\mathbf{x}, \mathbf{x}')$. For example, $w_{\mathbf{x}\mathbf{x}'}$ between a pair of samples from the *same* class is larger than that between a pair from *different* classes (ii) $P(\mathbf{x}) = w_\mathbf{x} \triangleq \sum_{\mathbf{x}' \in \mathcal{X}} w_{\mathbf{x}\mathbf{x}'}$ such that $\sum_{\mathbf{x} \in \mathcal{X}} \sum_{\mathbf{x}' \in \mathcal{X}} w_{\mathbf{x}\mathbf{x}'} = 1$, intuitively implying that a more representative instance $\mathbf{x}$ (corresponding to a larger $w_\mathbf{x}$) has a higher probability of being sampled. We remark that such population-induced graph $G_\mathcal{X}$ is reminiscent of the population augmentation graph for studying contrastive learning[4] [HaoChen et al., 2021].

Let $\mathbf{W}(\mathcal{X})$ with $\mathbf{W}_{\mathbf{x}\mathbf{x}'}(\mathcal{X}) = w_{\mathbf{x}\mathbf{x}'}$ be the weighted adjacency matrix of $G_\mathcal{X}$; let $\mathbf{D}(\mathcal{X})$ be the diagonal matrix of the weighted degrees $\{w_\mathbf{x}\}_{\mathbf{x} \in \mathcal{X}}$. The (normalized) graph Laplacian takes the form $\mathbf{L}(\mathcal{X}) = \mathbf{I} - \overline{\mathbf{W}}(\mathcal{X})$ where $\overline{\mathbf{W}}(\mathcal{X}) \triangleq \mathbf{D}(\mathcal{X})^{-1/2} \mathbf{W}(\mathcal{X}) \mathbf{D}(\mathcal{X})^{-1/2}$ is the normalized adjacency matrix. Then, the *spectral clustering* [von Luxburg, 2007] on $G_\mathcal{X}$ can be expressed as a (rank-$K$) Nyström approximation of $\overline{\mathbf{W}}(\mathcal{X})$ in terms of the weighted outputs $\mathbf{D}(\mathcal{X})^{\frac{1}{2}} f(\mathcal{X})$:

$$\mathcal{F}_{\mathbf{L}(\mathcal{X})} \triangleq \underset{f \in \mathcal{F}}{\mathrm{argmin}} \left\{ R(f) \triangleq \left\| \overline{\mathbf{W}}(\mathcal{X}) - \mathbf{D}(\mathcal{X})^{\frac{1}{2}} f(\mathcal{X}) f(\mathcal{X})^\top \mathbf{D}(\mathcal{X})^{\frac{1}{2}} \right\|_F^2 \right\}. \tag{1}$$

To quantify the alignment between the ground truth class partition and the clustering predicted by $f \in \mathcal{F}$, we introduce the notion of clustering error.

---

[3] For demonstration purposes, we assume an arbitrarily large, but finite, population $|\mathcal{X}| < \infty$ (as in HaoChen et al. [2021], *e.g.*, $\mathcal{X} = \mathbb{F}^d$ where $\mathbb{F}$ is the set of all floating-point numbers with finite precision). Meanwhile, our analyses can be generalized to any compact set $\mathcal{X}$ via functional analysis tools (*e.g.*, replacing the adjacency matrix $\mathbf{W}(\mathcal{X})$ with an adjacency operator).

[4] The key difference between the population augmentation graph [HaoChen et al., 2021] and the population-induced graph described here is that for analyzing RKD, we consider a graph over the original, instead of the augmented, population. It's also worth highlighting that we introduce such graph structure only for the sake of analysis, while our data distribution is generic. Therefore, we focus on learning from a general function class $\mathcal{F}$ without constraining to graph neural networks.

**Definition 3.1** (Clustering error). *Given any $f \in \mathcal{F}$, we define the majority labeling*

$$\widetilde{y}_f\left(\mathbf{x}\right) \triangleq \underset{k \in [K]}{\arg\max}\, \mathbb{P}_{\mathbf{x}' \sim P(\mathbf{x})}\left[y_*(\mathbf{x}') = k \mid y_f(\mathbf{x}') = y_f(\mathbf{x})\right], \qquad (2)$$

*along with the minority subsets associated with $f$: $M\left(f\right) \triangleq \{\mathbf{x} \in \mathcal{X} \mid \widetilde{y}_f(\mathbf{x}) \neq y_*\left(\mathbf{x}\right)\}$ such that $P\left(M(f)\right)$ quantifies the clustering error of $f$. For any $\mathcal{F}' \subseteq \mathcal{F}$, let $\mu\left(\mathcal{F}'\right) \triangleq \sup_{f \in \mathcal{F}'} P\left(M\left(f\right)\right)$.*

Intuitively, $M(f)$ characterizes the difference between $K$-partition of $\mathcal{X}$ by the ground truth $y_*$ and by the prediction function $y_f$ associated with $f$; while $\mu\left(\mathcal{F}'\right)$ quantifies the worse-case clustering error of all functions $f \in \mathcal{F}'$. In Section 4.1, we will demonstrate that spectral clustering on $G_{\mathcal{X}}$ (Equation (1)) leads to provably low clustering error $\mu\left(\mathcal{F}_{\mathbf{L}(\mathcal{X})}\right)$.

### 3.2 Relational Knowledge Distillation

For RKD, we assume access to a proper teacher model $\psi : \mathcal{X} \to \mathcal{W}$ (for a latent feature space $\mathcal{W}$) that induces a graph-revealing kernel: $k_\psi\left(\mathbf{x}, \mathbf{x}'\right) = \frac{w_{\mathbf{xx}'}}{w_{\mathbf{x}} w_{\mathbf{x}'}}$. Then, the spectral clustering on $G_{\mathcal{X}}$ in Equation (1) can be interpreted as the *population RKD loss*:

$$R\left(f\right) = \mathbb{E}_{\mathbf{x}, \mathbf{x}' \sim P(\mathbf{x})^2}\left[\left(k_\psi\left(\mathbf{x}, \mathbf{x}'\right) - f(\mathbf{x})^\top f(\mathbf{x}')\right)^2\right].$$

While with only limited unlabeled samples $\mathbf{X}^u = \left\{\mathbf{x}_j^u \mid j \in [N]\right\} \sim P(\mathbf{x})^N$ in practice, we consider the analogous *empirical RKD loss*:

$$\mathcal{F}_{\mathbf{L}(\mathbf{X}^u)} \triangleq \underset{f \in \mathcal{F}}{\arg\min}\left\{\widehat{R}_{\mathbf{X}^u}\left(f\right) \triangleq \frac{2}{N} \sum_{i=1}^{N/2} \left(f\left(\mathbf{x}_{2i-1}^u\right)^\top f\left(\mathbf{x}_{2i}^u\right) - k_\psi\left(\mathbf{x}_{2i-1}^u, \mathbf{x}_{2i}^u\right)\right)^2\right\}, \qquad (3)$$

where $\widehat{R}_{\mathbf{X}^u}(f)$ serves as an unbiased estimate for $R(f)$ (Proposition D.1).

Further, for *semi-supervised setting* with a small set of labeled samples $(\mathbf{X}, \mathbf{y}) = \left\{(\mathbf{x}_i, y_i)\right\}_{i \in [n]} \sim P\left(\mathbf{x}, y\right)^n$ (usually $n \ll N$) *independent of* $\mathbf{X}^u$, let $\ell : \mathbb{R}^K \times [K] \to \{0, 1\}$ be the zero-one loss: $\ell\left(f(\mathbf{x}), y\right) = \mathbb{1}\left\{y_f(\mathbf{x}) \neq y\right\}$. We denote $\mathcal{E}\left(f\right) \triangleq \mathbb{E}_{(\mathbf{x}, y) \sim P}\left[\ell\left(f(\mathbf{x}), y\right)\right]$ and $\widehat{\mathcal{E}}\left(f\right) \triangleq \frac{1}{n}\sum_{i=1}^n \ell\left(f(\mathbf{x}_i), y_i\right)$ as the population and empirical losses, respectively, and consider a proper learning setting with the ground truth $f_* \triangleq \arg\min_{f \in \mathcal{F}} \mathcal{E}\left(f\right)$. Then, SSL with RKD aims to find $\min_{f \in \mathcal{F}} \widehat{\mathcal{E}}(f) + \widehat{R}_{\mathbf{X}^u}(f)$. Alternatively, for an overparameterized setting[5] with $\left(\arg\min_{f \in \mathcal{F}} \widehat{\mathcal{E}}\left(f\right)\right) \cap \mathcal{F}_{\mathbf{L}(\mathbf{X}^u)} \neq \emptyset$, SSL with RKD can be formulated as: $\min_{f \in \mathcal{F}_{\mathbf{L}(\mathbf{X}^u)}} \widehat{\mathcal{E}}\left(f\right)$.

## 4 Relational Knowledge Distillation as Spectral Clustering

In this section, we show that minimizing either the population RKD loss (Section 4.1) or the empirical RKD loss (Section 4.2) leads to low clustering errors $\mu\left(\mathcal{F}_{\mathbf{L}(\mathcal{X})}\right)$ and $\mu\left(\mathcal{F}_{\mathbf{L}(\mathbf{X}^u)}\right)$, respectively.

### 4.1 Relational Knowledge Distillation over Population

Starting with minimization of the population RKD loss $f \in \mathcal{F}_{\mathbf{L}(\mathcal{X})} = \arg\min_{f \in \mathcal{F}} R\left(f\right)$ (Equation (1)), let $\lambda\left(\mathbf{L}\left(\mathcal{X}\right)\right) = \left(\lambda_1, \ldots, \lambda_{|\mathcal{X}|}\right)$ be the eigenvalues of the graph Laplacian in the ascending order, $0 = \lambda_1 \leq \cdots \leq \lambda_{|\mathcal{X}|}$ with an arbitrary breaking of ties.

A key assumption of RKD is that the teacher model $\psi$ (with the corresponding graph-revealing kernel $k_\psi$) is well aligned with the underlying ground truth partition, formalized as below:

**Assumption 4.1** (Appropriate teacher models). *For $G_{\mathcal{X}}$ unveiled through the teacher model $k_\psi\left(\cdot, \cdot\right)$, we assume $\lambda_{K+1} > 0$ (i.e., $G_{\mathcal{X}}$ has at most $K$ disconnected components); while the ground truth classes $\{\mathcal{X}_k\}_{k \in [K]}$ are well-separated by $G_{\mathcal{X}}$ such that $\alpha \triangleq \frac{\sum_{k \in [K]} \sum_{\mathbf{x} \in \mathcal{X}_k} \sum_{\mathbf{x}' \notin \mathcal{X}_k} w_{\mathbf{xx}'}}{2 \sum_{\mathbf{x} \in \mathcal{X}} \sum_{\mathbf{x}' \in \mathcal{X}} w_{\mathbf{xx}'}} \ll 1$ (i.e., the fraction of weights of inter-class edges is sufficiently small).*

---

[5]Overparameterization is a ubiquitous scenario when learning with neural networks: $f \in \mathcal{F}$ is parameterized by $\theta \in \Theta$ with $\dim(\Theta) > N + n$ such that the minimization problem has infinitely many solutions.

In particular, $\alpha \in [0, 1]$ reflects the extent to which the ground truth classes $\{\mathcal{X}_k\}_{k \in [K]}$ are separated from each other, as quantified by the fraction of edge weights between nodes of disparate classes. In the ideal case, $\alpha = 0$ when the ground truth classes are perfectly separated such that every $\mathcal{X}_k$ is a connected component of $G_{\mathcal{X}}$, while $\lambda_{K+1} > \lambda_K = 0$.

Meanwhile, to reduce the generic function class $\mathcal{F}$ down to a class of reasonable prediction functions, we introduce the following regularity conditions on the boundedness and margin:

**Assumption 4.2** ($\beta$-skeleton boundedness and $\gamma$-margin). *For any $f \in \mathcal{F}$ with $P(M(f) \cap \mathcal{X}_k) \leq P(\mathcal{X}_k)/2$ for all $k \in [K]$, we assume there exists a skeleton subset $\mathbf{S} = [\mathbf{s}_1; \dots; \mathbf{s}_K] \in \mathcal{X}^K$ with[6] $\mathbf{s}_k = \arg\max_{\mathbf{x} \in \mathcal{X} \setminus M(f)} f(\mathbf{x})_k$ such that $y_f(\mathbf{s}_k) = k$ for every $k \in [K]$ and $\mathrm{rank}(f(\mathbf{S})) = K$.*

    *(i) We say that $f$ is $\beta$-skeleton bounded if $\sigma_1(f(\mathbf{S})) \leq \beta$ for some reasonably small $\beta$.*
    *(ii) Let the $k$-th margin of $f$ be $\gamma_k \triangleq f(\mathbf{s}_k)_k - \max_{\mathbf{x} \in M(f):y_f(\mathbf{x}) \neq k} f(\mathbf{x})_k$. We say that $f$ has a $\gamma$-margin if $\min_{k \in [K]} \gamma_k > \gamma$ for some sufficiently large $\gamma > 0$.*

Intuitively, $\mathbf{S}$ can be viewed as a set of $K$ samples where $f$ makes the "most confident" prediction in each class. Assumption 4.2 ensures the boundedness of predictions made on such a skeleton subset $\|f(\mathbf{S})\|_2 \leq \beta$, as well as a margin $\gamma$ by which the "most confident" prediction of $f$ in each class $\mathbf{s}_k$ can be separated from the minority samples predicted to lie in other classes $\{\mathbf{x} \in M(f)|y_f(\mathbf{x}) \neq k\}$. As a toy example, when $y_f : \mathcal{X} \to [K]$ is surjective, and the columns in $f(\mathcal{X}) \in \mathbb{R}^{|\mathcal{X}| \times K}$ consist of identity vectors of the predicted clusters $f(\mathbf{x})_k = \mathbb{1}\{y_f(\mathbf{x}) = k\}$, Assumption 4.2 is satisfied with $\beta = 1$ and $\gamma = 1$. Meanwhile, the following Remark 4.1 highlights that although Assumption 4.2 cannot be guaranteed with spectral clustering alone, it is generally satisfied in the semi-supervised learning setting with additional supervision/regularization (*cf.* Example C.1).

**Remark 4.1** (Limitation of spectral clustering alone). *Notice that for a generic function class $\mathcal{F}$, spectral clustering alone is not sufficient to guarantee Assumption 4.2, especially the existence of a skeleton subset $\mathbf{S}$ or a large enough margin $\gamma$. As counter-exemplified in Example C.1, Equation (1) can suffer from large clustering error $\mu(\mathcal{F}_{\mathbf{L}(\mathcal{X})})$ when applied alone and failing to satisfy Assumption 4.2.*

*To learn predictions with accurate clustering, RKD requires either (i) additional supervision/regularization in end-to-end settings like semi-supervised learning or (ii) further fine-tuning in unsupervised representation learning settings like contrastive learning [HaoChen et al., 2021]. For end-to-end learning in practice, RKD is generally coupled with standard KD (i.e., feature matching) and weak supervision [Park et al., 2019, Liu et al., 2021, Ma et al., 2022], both of which help the learned function $f$ satisfy Assumption 4.2 with a reasonable margin $\gamma$.*

Throughout this work, we assume $\mathcal{F}$ is sufficiently regularized (with weak supervision in Section 5 and consistency regularization in Section 6) such that Assumption 4.2 always holds. Under Assumption 4.1 and Assumption 4.2, the clustering error of RKD over the population $\mu(\mathcal{F}_{\mathbf{L}(\mathcal{X})})$ is guaranteed to be small given a good teacher model $\psi$ that leads to a negligible $\alpha$:

**Theorem 4.1** (RKD over population, proof in Appendix C.1). *Under Assumption 4.1 and Assumption 4.2 for every $f \in \mathcal{F}_{\mathbf{L}(\mathcal{X})}$, the clustering error with the population (Equation (1)) satisfies*

$$\mu(\mathcal{F}_{\mathbf{L}(\mathcal{X})}) \leq 2\max\left(\frac{\beta^2}{\gamma^2}, 1\right) \cdot \frac{\alpha}{\lambda_{K+1}}.$$

Theorem 4.1 suggests that the clustering error over the population is negligible under mild regularity assumptions (*i.e.*, Assumption 4.2) when (i) the ground truth classes are well-separated by $G_{\mathcal{X}}$ revealed through the teacher model $k_\psi(\cdot, \cdot)$ (*i.e.*, $\alpha \ll 1$ in Assumption 4.1) and (ii) the $(K+1)$th eigenvalue $\lambda_{K+1}$ of the graph Laplacian $\mathbf{L}(\mathcal{X})$ is not too small. As we review in Appendix C.3, the existing result Lemma C.4 [Louis and Makarychev, 2014] unveils the connection between $\lambda_{K+1}$ and the sparsest $K$-partition (Definition C.1) of $G_{\mathcal{X}}$. Intuitively, a reasonably large $\lambda_{K+1}$ implies that the partition of the $K$ ground truth classes is the "only" partition of $G_{\mathcal{X}}$ into $K$ components by removing a *sparse* set of edges (Corollary C.5).

---

[6]Recall the majority labeling (Equation (2)) and the associated minority subset $M(f)$ which intuitively describes the difference between clustering of the $y_*$ and that of $y_f$. We denote $f(\mathbf{x}) = [f(\mathbf{x})_1, \dots, f(\mathbf{x})_K]$.

## 4.2 Relational Knowledge Distillation on Unlabeled Samples

We now turn our attention to a more practical scenario with limited unlabeled samples and bound the clustering error $\mu\left(\mathcal{F}_{\mathbf{L}(\mathbf{X}^u)}\right)$ granted by minimizing the empirical RKD loss (Equation (3)).

To cope with $\mathcal{F} \ni f : \mathcal{X} \to \mathbb{R}^K$, we recall the notion of Rademacher complexity for vector-valued functions from Maurer [2016]. Let $\mathrm{Rad}(\rho)$ be the Rademacher distribution (*i.e.*, with uniform probability $\frac{1}{2}$ over $\{-1, 1\}$) and $\boldsymbol{\rho} \sim \mathrm{Rad}^{N \times K}$ be a random matrix with *i.i.d.* Rademacher entries. Given any $N \in \mathbb{N}$, with $f(\mathbf{x})_k$ denoting the $k$-th entry of $f(\mathbf{x}) \in \mathbb{R}^K$, we define

$$\mathfrak{R}_N\left(\mathcal{F}\right) \triangleq \mathbb{E}_{\substack{\mathbf{X}^u \sim P(\mathbf{x})^N \\ \boldsymbol{\rho} \sim \mathrm{Rad}^{N \times K}}} \left[ \sup_{f \in \mathcal{F}} \frac{1}{N} \sum_{i=1}^{N} \sum_{k=1}^{K} \rho_{ik} \cdot f\left(\mathbf{x}_i^u\right)_k \right] \tag{4}$$

as the Rademacher complexity of the vector-valued function class $\mathcal{F} \ni f : \mathcal{X} \to \mathbb{R}^K$.

Since the empirical RKD loss is an unbiased estimate of its population correspondence (Proposition D.1), we can leverage the standard generalization analysis in terms of Rademacher complexity to provide an unlabeled sample complexity for RKD.

**Theorem 4.2** (Unlabeled sample complexity of RKD, proof in Appendix D.2). *Assume there exist $B_f, B_{k_\psi} > 0$ such that $\|f(\mathbf{x})\|_2^2 \le B_f$ and $k_\psi\left(\mathbf{x}, \mathbf{x}'\right) \le B_{k_\psi}$ for all $\mathbf{x}, \mathbf{x}' \in \mathcal{X}$, $f \in \mathcal{F}$. Given any $f_{|\mathbf{X}^u} \in \mathcal{F}_{\mathbf{L}(\mathbf{X}^u)}$, $f_{|\mathcal{X}} \in \mathcal{F}_{\mathbf{L}(\mathcal{X})}$, $\delta \in (0, 1)$, with probability at least $1 - \delta/2$ over $\mathbf{X}^u \sim P(\mathbf{x})^N$,*

$$R\left(f_{|\mathbf{X}^u}\right) - R\left(f_{|\mathcal{X}}\right) \le 16\sqrt{2B_f}\left(B_f + B_{k_\psi}\right)\mathfrak{R}_{N/2}\left(\mathcal{F}\right) + 2\left(B_{k_\psi} + B_f\right)^2 \sqrt{\frac{\log\left(4/\delta\right)}{N}}.$$

Theorem 4.2 implies that the unlabeled sample complexity of RKD is dominated by the Rademacher complexity of $\mathcal{F}$. In Appendix D.3, we further concretize Theorem 4.2 by instantiating $\mathfrak{R}_{N/2}\left(\mathcal{F}\right)$ via existing Rademacher complexity bounds for neural networks [Golowich et al., 2018].

With $\Delta \to 0$ as $N$ increasing (Theorem 4.2), the clustering error of minimizing the empirical RKD loss can be upper bounded as follows.

**Theorem 4.3** (RKD on unlabeled samples, proof in Appendix D.4). *Under Assumption 4.1 and Assumption 4.2 for every $f \in \mathcal{F}_{\mathbf{L}(\mathbf{X}^u)}$, given any $\delta \in (0, 1)$, if there exists $\Delta < \left(1 - \lambda_K\right)^2$ such that $R\left(f_{|\mathbf{X}^u}\right) \le R\left(f_{|\mathcal{X}}\right) + \Delta$ for all $f_{|\mathbf{X}^u} \in \mathcal{F}_{\mathbf{L}(\mathbf{X}^u)}$ and $f_{|\mathcal{X}} \in \mathcal{F}_{\mathbf{L}(\mathcal{X})}$ with probability at least $1 - \delta/2$ over $\mathbf{X}^u \sim P(\mathbf{x})^N$, then error of clustering with the empirical graph (Equation (3)) satisfies the follows: for any $K_0 \in [K]$ such that $\lambda_{K_0} < \lambda_{K+1}$ and $C_{K_0} \triangleq \frac{\left(1 - \lambda_{K_0}\right)^2 - \left(1 - \lambda_K\right)^2}{\Delta} = O(1)$,*

$$\mu\left(\mathcal{F}_{\mathbf{L}(\mathbf{X}^u)}\right) \le 2\max\left(\frac{\beta^2}{\gamma^2}, 1\right) \cdot \left(\frac{\alpha}{\lambda_{K+1}} + \frac{\left(1 + \left(K - K_0\right)C_{K_0}\right)\Delta}{\left(1 - \lambda_{K_0}\right)^2 - \left(1 - \lambda_{K+1}\right)^2}\right).$$

Theorem 4.3 ensures that given sufficient unlabeled samples (proportional to the Rademacher complexity of $\mathcal{F}$) such that $\Delta \to 0$, the clustering error $\mu\left(\mathcal{F}_{\mathbf{L}(\mathbf{X}^u)}\right)$ from empirical RKD is nearly as low as $\mu\left(\mathcal{F}_{\mathbf{L}(\mathcal{X})}\right)$ from population RKD, up to an additional error term that scales linearly in $\Delta$.

# 5 Label Efficiency of Cluster-aware Semi-supervised Learning

In this section, we demonstrate the label efficiency of learning from a function class with low clustering error (Definition 3.1), like the ones endowed by RKD discussed in Section 4.

Specifically, given any cluster-aware function subclass $\mathcal{F}' \subseteq \mathcal{F}$ with low clustering error (*e.g.*, $\mathcal{F}_{\mathbf{L}(\mathcal{X})}$ and $\mathcal{F}_{\mathbf{L}(\mathbf{X}^u)}$), for a set of $n$ *i.i.d.* labeled samples $(\mathbf{X}, \mathbf{y}) \sim P(\mathbf{x}, y)^n$, we have the following generalization guarantee:

**Theorem 5.1** (Label complexity, proof in Appendix B.1). *Given any cluster-aware $\mathcal{F}' \subseteq \mathcal{F}$ with $\mu(\mathcal{F}') \ll 1$, assuming that $(\mathbf{X}, \mathbf{y})$ contains at least one sample per class, for any $\delta \in (0, 1)$, with probability at least $1 - \delta/2$ over $(\mathbf{X}, \mathbf{y}) \sim P(\mathbf{x}, y)^n$, $\widehat{f} \in \arg\min_{f \in \mathcal{F}'} \widehat{\mathcal{E}}(f)$ satisfies*

$$\mathcal{E}\left(\widehat{f}\right) - \mathcal{E}\left(f_*\right) \le 4\sqrt{\frac{2K\log(2n)}{n}} + 2\mu\left(\mathcal{F}'\right) + \sqrt{\frac{2\log\left(4/\delta\right)}{n}}, \tag{5}$$

*e.g., conditioned on $\mathbf{X}^u$, $\widehat{f}_{|\mathbf{X}^u} \in \arg\min_{f \in \mathcal{F}_{\mathbf{L}(\mathbf{X}^u)}} \widehat{\mathcal{E}}(f)$ satisfies Equation (5) with $\mu\left(\mathcal{F}_{\mathbf{L}(\mathbf{X}^u)}\right)$.*

Theorem 5.1 implies that with low clustering error (*e.g.*, endowed by unsupervised methods like RKD in Theorem 4.1/Theorem 4.3, DAC in Theorem 6.2), the labeled sample complexity is as low as $\widetilde{O}(K)$, linear in the number of clusters and asymptotically optimal[7] up to a logarithmic factor.

**Remark 5.1** (Class imbalance, elaborated in Appendix B.2). *In Theorem 5.1, while the label complexity scales as $n = \widetilde{O}(K)$ with* i.i.d. *sampling, we meanwhile assume that $(\mathbf{X}, \mathbf{y})$ contains at least one labeled sample per class. Intuitively, when the $K$ classes are balanced (i.e., $|\mathcal{X}_k| = |\mathcal{X}|/K$ for all $k \in [K]$), an analogy to the coupon collector problem implies that $n = O(K \log(K))$ i.i.d. labeled samples are sufficient in expectation. Nevertheless, with class imbalance, the label complexity can be much worse (e.g., when $|\mathcal{X}_K| \ll |\mathcal{X}_k|$ for all $k \in [K-1]$, collecting one label $K$ takes $1/P(\mathcal{X}_K) \gg K$ labeled samples). In Appendix B.2, we show that such label inefficiency can be circumvented by leveraging a cluster-aware prediction $f \in \mathcal{F}'$ and drawing $O(\log(K))$ labeled samples uniformly from each of the $K$ predicted clusters, instead of* i.i.d. *from the entire population.*

**Remark 5.2** (Coreset selection). *While Theorem 5.1 yields an asymptotically optimal label complexity guarantee with* i.i.d. *sampling, it is of significant practical value to be more judicious in selecting labeled samples, especially for the low-label-rate regime (e.g., $n = O(K)$). Therefore, in the experiments (Appendix A), we further investigate—alongside* i.i.d. *sampling[8]—a popular coreset selection method [Bilmes, 2022, Krause and Golovin, 2014], where the coreset $(\mathbf{X}, \mathbf{y})$ is statistically more representative of the data distribution $P$.*

*In particular, coreset selection can be recast as a facility location problem [Krause and Golovin, 2014] characterized by the teacher model: $\max_{\mathbf{X} \subset \mathcal{X}} \sum_{\mathbf{x}' \in \mathcal{X}} \max_{\mathbf{x} \in \mathbf{X}} k_\psi(\mathbf{x}, \mathbf{x}')$, whose optimizers can be approximated heuristically via the stochastic greedy [Mirzasoleiman et al., 2015] submodular optimization algorithm [Schreiber et al., 2020]. Intuitively, the facility location objective encourages the coreset $\mathbf{X}$ to be representative of the entire population $\mathcal{X}$ in a pairwise similarity sense, and the coreset approximation identified by the stochastic greedy method is nearly optimal (in the facility location objective) up to a multiplicative constant [Mirzasoleiman et al., 2015].*

## 6 Data Augmentation Consistency Regularization as Clustering

In light of the broadness of the notion of clustering, investigating the discrepancy between different types of cluster-awareness is crucial for understanding the effect of spectral clustering brought by RKD beyond the low clustering error demonstrated in Section 4. In this section, we leverage the existing theoretical tools from Wei and Ma [2019], Cai et al. [2021] to unify DAC regularization into the cluster-aware semi-supervised learning framework in Theorem 5.1. Specifically, we demonstrate the effect of DAC as a "local" expansion-based clustering [Yang et al., 2023], which works in complement with the "global" spectral clustering facilitated by RKD.

Start by recalling the formal notion of expansion-based data augmentation (Definition 6.1) and data augmentation consistency (DAC) error (Definition 6.2) from Wei et al. [2021]:

**Definition 6.1** (Expansion-based data augmentation ([Wei et al., 2021] Definition 3.1)). *For any $\mathbf{x} \in \mathcal{X}$, we consider a set of class-invariant data augmentations $\mathcal{A}(\mathbf{x}) \subset \mathcal{X}$ such that $\{\mathbf{x}\} \subsetneq \mathcal{A}(\mathbf{x}) \subseteq \mathcal{X}_{y_*(x)}$. We say $\mathbf{x}, \mathbf{x}' \in \mathcal{X}$ lie in neighborhoods of each other if their augmentation sets have a non-empty intersection: $\mathrm{NB}(\mathbf{x}) \triangleq \{\mathbf{x}' \in \mathcal{X} \mid \mathcal{A}(\mathbf{x}) \cap \mathcal{A}(\mathbf{x}') \neq \emptyset\}$ for $\mathbf{x} \in \mathcal{X}$; and $\mathrm{NB}(S) \triangleq \bigcup_{\mathbf{x} \in S} \mathrm{NB}(\mathbf{x})$ for $S \subseteq \mathcal{X}$. Then, we quantify* strength *of such data augmentations via the $c$-expansion property (c > 1): for any $S \subseteq \mathcal{X}$ such that $P(S \cap \mathcal{X}_k) \leq P(\mathcal{X}_k)/2 \,\forall\, k \in [K]$,*

$$P(\mathrm{NB}(S) \cap \mathcal{X}_k) > \min\{c \cdot P(S \cap \mathcal{X}_k), P(\mathcal{X}_k)\} \;\forall\, k \in [K].$$

**Definition 6.2** (DAC error [Wei et al., 2021] (3.3)). *For any $f \in \mathcal{F}$ and $\mathcal{F}' \subseteq \mathcal{F}$, let*

$$\nu(f) \triangleq \mathbb{P}_{\mathbf{x} \sim P(\mathbf{x})}\left[\exists\, \mathbf{x}' \in \mathcal{A}(\mathbf{x}) \text{ s.t. } y_f(\mathbf{x}) \neq y_f(\mathbf{x}')\right], \quad \nu(\mathcal{F}') \triangleq \sup_{f \in \mathcal{F}'} \nu(f).$$

---

[7]Notice that assuming at least one labeled sample per class (*i.e.*, $n \geq K$) is necessary for any generalization guarantees. Otherwise, depending on the missing classes, the excess risk in Equation (5) can be arbitrarily bad.

[8]For the low-label-rate experiments, to ensure consistent performance, we follow the common practice [Calder et al., 2020] and slightly deviate from the assumption by sampling *i.i.d.* from each ground truth class instead.

We adopt the theoretical framework in Wei et al. [2021] for generic neural networks with smooth activation function $\phi$ and consider

$$\mathcal{F} = \left\{ f(\mathbf{x}) = \mathbf{A}_p \phi\left(\cdots \mathbf{A}_2 \phi(\mathbf{A}_1 \mathbf{x}) \cdots\right) \,\middle|\, \mathbf{A}_\iota \in \mathbb{R}^{d_\iota \times d_{\iota-1}} \,\forall\, \iota \in [p],\ d = \max_{\iota=0,\cdots,p} d_\iota \right\}.$$

With such function class $\mathcal{F}$ of $p$-layer neural networks with maximum width $d$ and weights $\{\mathbf{A}_\iota\}_{\iota=1}^p$, we recall the notion of *all-layer margin* from Wei et al. [2021] Appendix C.2. By decomposing $f = f_{2p-1} \circ \cdots \circ f_1$ such that $f_{2\iota-1}(\mathbf{z}) = \mathbf{A}_\iota \mathbf{z}$ for all $\iota \in [p]$ and $f_{2\iota}(\mathbf{z}) = \phi(\mathbf{z})$ for all $\iota \in [p-1]$, we consider a perturbation $\boldsymbol{\delta} = (\boldsymbol{\delta}_1, \cdots, \boldsymbol{\delta}_{2p-1})$ (where $\boldsymbol{\delta}_{2\iota-1}, \boldsymbol{\delta}_{2\iota} \in \mathbb{R}^{d_\iota}$) to each layer of $f$:

$$f_1(\mathbf{x}, \boldsymbol{\delta}) = f_1(\mathbf{x}, \boldsymbol{\delta}_1) = f_1(\mathbf{x}) + \boldsymbol{\delta}_1 \|\mathbf{x}\|_2,$$
$$f_\iota(\mathbf{x}, \boldsymbol{\delta}) = f_\iota(\mathbf{x}, \boldsymbol{\delta}_1, \cdots, \boldsymbol{\delta}_\iota) = f_\iota\left(f_{\iota-1}(\mathbf{x}, \boldsymbol{\delta})\right) + \boldsymbol{\delta}_\iota \|f_{\iota-1}(\mathbf{x}, \boldsymbol{\delta})\|_2$$

such that $f(\mathbf{x}, \boldsymbol{\delta}) = f_{2p-1}(\mathbf{x}, \boldsymbol{\delta})$. Then, the all-layer margin $m : \mathcal{F} \times \mathcal{X} \times [K] \to \mathbb{R}_{\geq 0}$ is defined as the minimum norm of $\boldsymbol{\delta}$ that is sufficient to perturb the classification of $f(\mathbf{x})$:

$$m(f, \mathbf{x}, y) \triangleq \min_{\boldsymbol{\delta}} \sqrt{\sum_{\iota=1}^{2p-1} \|\boldsymbol{\delta}_\iota\|_2^2} \quad s.t. \quad \operatorname*{argmax}_{k \in [K]} f(\mathbf{x}, \boldsymbol{\delta})_k \neq y. \tag{6}$$

Moreover, Wei et al. [2021] introduced the *robust margin* for the expansion-based data augmentation $\mathcal{A}$ (Definition 6.1), which intuitively quantifies the worst-case preservation of the label-relevant information in the augmentations of $\mathbf{x}$, measured with respect to $f$:

$$m_{\mathcal{A}}(f, \mathbf{x}) \triangleq \min_{\mathbf{x}' \in \mathcal{A}(\mathbf{x})} m(f, \mathbf{x}', y_f(\mathbf{x})). \tag{7}$$

We say the expansion-based data augmentation $\mathcal{A}$ is *margin-robust* with respect to the ground truth $f_*$ if $\min_{\mathbf{x} \in \mathcal{X}} m_{\mathcal{A}}(f_*, \mathbf{x}) > 0$ is reasonably large.

Then, we cast DAC regularization[9] as reducing the function class $\mathcal{F}$ via constraints on the robust margins at the $N$ unlabeled samples $\mathbf{X}^u$: for some $0 < \tau < \min_{\mathbf{x} \in \mathcal{X}} m_{\mathcal{A}}(f_*, \mathbf{x})$,

$$\mathcal{F}^\tau_{\mathbf{X}^u} \triangleq \left\{ f \in \mathcal{F} \mid m_{\mathcal{A}}(f, \mathbf{x}_i^u) \geq \tau \,\forall\, i \in [N] \right\}. \tag{8}$$

Leverage the existing unlabeled sample complexity bound for DAC error from Wei et al. [2021]:

**Proposition 6.1** (Unlabeled sample complexity of DAC ([Wei et al., 2021] Theorem 3.7)). *Given any $\delta \in (0, 1)$, with probability at least $1 - \delta/2$ over $\mathbf{X}^u$,*

$$\nu\left(\mathcal{F}^\tau_{\mathbf{X}^u}\right) \leq \widetilde{O}\left(\frac{\sum_{\iota=1}^p \sqrt{d}\,\|\mathbf{A}_\iota\|_F}{\tau \sqrt{N}}\right) + O\left(\sqrt{\frac{\log(1/\delta) + p \log(N)}{N}}\right),$$

*where $\widetilde{O}(\cdot)$ suppresses polylogarithmic factors in $N$ and $d$.*

The clustering error of Equation (8) is as low as its DAC error, up to a constant factor that depends on the augmentation strength, characterized by the $c$-expansion (Definition 6.1):

**Theorem 6.2** (Expansion-based clustering, proof in Appendix E.3). *For Equation (8) with $\mathcal{A}$ admitting $c$-expansion (Definition 6.1), the clustering error is upper bounded by the DAC error in Proposition 6.1:*

$$\mu\left(\mathcal{F}^\tau_{\mathbf{X}^u}\right) \leq \max\left\{\frac{2}{c-1}, 2\right\} \cdot \nu\left(\mathcal{F}^\tau_{\mathbf{X}^u}\right).$$

In Theorem 6.2, (i) $c$ characterizes the augmentation strength (*i.e.*, the perturbation on label-irrelevant information), whereas (ii) $\tau$ quantifies the margin robustness (*i.e.*, the preservation of label-relevant information) for the expansion-based data augmentation. Ideally, we would like both $c$ and $\tau$ to be reasonably large, while there exist trade-offs between the augmentation strength and margin robustness (*e.g.*, overly strong augmentations inevitably perturb label-relevant information).

---

[9]In Appendix E.2, we further bridge the conceptual notion of DAC regularization in Equation (8) with common DAC regularization algorithms in practice like FixMatch [Sohn et al., 2020].

**Remark 6.1** (Clustering with DAC v.s. RKD). *As Figure 1 demonstrated, in contrast to RKD that unveils the spectral clustering of a population-induced graph from a "global" perspective, DAC alternatively learns a "local" clustering structure through the expansion of neighborhoods* $\mathrm{NB}(\cdot)$ *characterized by sets of data augmentations* $\mathcal{A}(\cdot)$ *(Definition 6.1).*

*Therefore, DAC and RKD provide complementary perspectives for each other. On one hand, the "local" clustering structure revealed by DAC effectively ensures a reasonable margin in Assumption 4.2 (Remark 4.1). On the other hand, when the augmentation strength $c$ is insufficient (i.e., $\mathcal{A}(\cdot)$ is not expansive enough) for DAC to achieve a low clustering error $\mu\left(\mathcal{F}_{\mathbf{X}^u}^\tau\right)$ (Theorem 6.2), the supplementary "global" perspective of RKD connects the non-overlapping neighborhoods in the same classes and brings better classification accuracies, as we empirically verified in Appendix A.*

## 7 Discussions, Limitations, and Future Directions

This work provides a theoretical analysis of relational knowledge distillation (RKD) in the semi-supervised classification setting from a clustering perspective. Through a cluster-aware semi-supervised learning (SSL) framework characterized by a notion of low clustering error, we demonstrate that RKD achieves a nearly optimal label complexity (linear in the number of clusters) by learning the underlying geometry of the population as revealed by the teacher model via spectral clustering. With a unified view of data augmentation consistency (DAC) regularization in the cluster-aware SSL framework, we further illustrate the complementary "global" and "local" perspectives of clustering learned by RKD and DAC, respectively.

As an appealing future direction, domain adaptation in knowledge distillation is a common scenario in practice where the training data of the teacher and student models come from different distributions. Although distributional shifts can implicitly reflect the alignment between the ground truth and the teacher model (Assumption 4.1) in our analysis, an explicit and specialized study on RKD in the domain adaptation setting may lead to better theoretical guarantees and deeper insights.

Another avenue for future exploration involves RKD on different (hidden) layers of the neural network. In this work, we consider RKD on the output layer following the standard practice [Park et al., 2019], whereas RKD on additional hidden layers has been shown to further improve the performance [Liu et al., 2019]. While the analysis in Section 4 remains valid for hidden-layer features of low dimensions ($\approx K$), Assumption 4.1 tends to fail (*i.e.*, $\alpha$ can be large) for high-dimensional features, which may require separate and careful consideration.

## Acknowledgement

YD was supported in part by the Office of Naval Research N00014-18-1-2354, NSF DMS-1952735, DOE ASCR DE-SC0022251, and UT Austin Graduate School Summer Fellowship. KM was supported by the Peter J. O'Donnell Jr. Postdoctoral Fellowship at the Oden Institute at the University of Texas, Austin. QL acknowledges the support of NYU Research Catalyst Prize and Whitehead Fellowship for Junior Faculty in Biomedical and Biological Sciences. RW was supported in part by AFOSR MURI FA9550-19-1-0005, NSF DMS-1952735, NSF IFML grant 2019844, NSF DMS-N2109155, and NSF 2217033. The authors wish to thank IFML for generously providing computational resources, as well as the anonymous reviewers for their helpful comments and suggestions.

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

# A Experiments

In this section, we present experimental results on CIFAR-10/100 [Krizhevsky and Hinton, 2009] to demonstrate the efficacy of combining DAC and RKD (*i.e.*, the "local" and "global" perspectives of clustering) for semi-supervised learning in the low-label-rate regime. The experiment code can be found at https://github.com/dyjdongyijun/Semi_Supervised_Knowledge_Distillation.

We consider the (data-free) knowledge distillation setting, wherein the student model cannot access the training data of the teacher model. That is, the student models only have access to limited training samples, with very few labels, drawn from a potentially different data distribution than which the teacher model is pretrained on. More precisely, given a labeled sample set $(\mathbf{X}, \mathbf{y})$ of size $n$ and an unlabeled sample set $\mathbf{X}^u$ of size $N$ $(n \ll N)$[10], we train a student model (WideResNet [Zagoruyko and Komodakis, 2016b]) in the low-label-rate regime with as few as 4 labeled samples per class (*i.e.*, $n = 4K$, or $n = 40$ for CIFAR-10 and $n = 400$ for CIFAR-100), along with $N = 25000, 40000$, and $50000$ unlabeled samples (*i.e.*, $50\%, 80\%, 100\%$ of the entire CIFAR-10/100 training set).

Then, combining DAC and RKD, we solve the following objective in the experiments:

$$
\min_{f \in \mathcal{F}} \underbrace{\widehat{\mathcal{L}}^{\mathrm{CE}}_{(\mathbf{X}, \mathbf{y})}(f)}_{\text{Supervised cross-entropy}} + \underbrace{\lambda_{\mathrm{DAC}} \cdot \widehat{\mathcal{L}}^{\mathrm{DAC}}_{\mathbf{X}^u}(f)}_{\text{Unsupervised DAC loss}} + \underbrace{\lambda_{\mathrm{RKD}} \cdot \widehat{R}_{\mathbf{X}^u}(f)}_{\text{empirical RKD loss}},
$$

where $\widehat{\mathcal{L}}^{\mathrm{CE}}_{(\mathbf{X}, \mathbf{y})}(f) = \frac{1}{n} \sum_{i=1}^{n} \ell_{\mathrm{CE}}(y_i | f(\mathbf{x}_i))$ denotes the supervised cross-entropy loss with $\ell_{\mathrm{CE}}(y | f(\mathbf{x})) = -\log \left( \mathtt{softmax}(f(\mathbf{x}))_y \right)$. The DAC and RKD losses are as described in Equation (9) and Equation (3), respectively. For the regularization hyper-parameters, we have $\lambda_{\mathrm{DAC}} = 1$ and $\lambda_{\mathrm{RKD}} = 0$ for DAC (only), while $\lambda_{\mathrm{RKD}} = 0.001$ for DAC + RKD.

**DAC via FixMatch.** We take FixMatch [Sohn et al., 2020]–a state-of-the-art algorithm in the low-label-rate regime–as the semi-supervised learning baseline. By combining RKD with FixMatch, we show that RKD brings additional boosts to the classification accuracy, especially with unsuitable data augmentations and/or limited unlabeled data.

Following the formulation and implementation in Sohn et al. [2020], FixMatch involves a pair of weak and strong data augmentations $(\mathbf{x}^w, \mathbf{x}^s)$ for each unlabeled sample $\mathbf{x}^u$. For a current model $f$, the DAC loss given by FixMatch can be expressed as

$$
\widehat{\mathcal{L}}^{\mathrm{DAC}}_{\mathbf{X}^u}(f) \triangleq
$$
$$
\mathrm{mean}\left( \left\{ \ell_{\mathrm{CE}}(y_f(\mathbf{x}_i^w) | f(\mathbf{x}_i^s)) \;\middle|\; \max\left( \mathtt{softmax}\left( \frac{f(\mathbf{x}_i^w)}{T} \right) \right) \geq \tau_{\mathrm{DAC}}, \; i \in [N] \right\} \right), \tag{9}
$$

where $T = 1$ is a fixed temperature hyper-parameter. $\tau_{\mathrm{DAC}}$ is the threshold below which the pseudo-label gauged on $\mathbf{x}_i^w$ is considered "with low confidence" and therefore discarded. We set $\tau_{\mathrm{DAC}} = 0.95$ for CIFAR-10 and $\tau_{\mathrm{DAC}} = 0.8$ for CIFAR-100.

In FixMatch, the weak augmentations include random flips and crops, whereas the strong augmentations additionally involve RandAugment [Cubuk et al., 2020]. To investigate the scenarios with unsuitable data augmentations and/or limited unlabeled data, we conduct experiments with various (i) augmentation strength (*i.e.*, magnitude of transformations) $m = 2, 6, 10$ in RandAugment [Cubuk et al., 2020] and (ii) unlabeled sample size $N = 25000, 40000, 50000$ (*i.e.*, $50\%, 80\%, 100\%$ of the entire CIFAR-10/100 training set).

**RKD.** We warm up with the *in-distribution* case in the CIFAR-10 experiments (Table 1) where the teacher and student models are (pre)trained on the same dataset (*i.e.*, CIFAR-10). Concretely, the teacher model $\psi : \mathcal{X} \to \mathcal{W}$ is a Densenet 161 [Huang et al., 2017, Phan, 2021] pretrained on the entirety of the CIFAR-10 training set (via supervised ERM).

Taking a step further, we consider an *out-of-distribution* scenario in the CIFAR-100 experiments (Table 2) where the teacher model is pretrained on ImageNet [Deng et al., 2009] via an unsupervised contrastive learning method–SwAV [Caron et al., 2020]. Intuitively, we choose teacher models

---

[10]It is worth mentioning that, in practice, we relax the assumption of the independence between $(\mathbf{X}, \mathbf{y})$ and $\mathbf{X}^u$ and recycle $\mathbf{X}$ in $\mathbf{X}^u$.

pretrained via SwAV as it encourages both data clustering and consistency of cluster assignment under data augmentations without involving supervision, which tends to provide representations that generalize better in a cluster-aware semi-supervised learning setting.

Given the teacher models, the output (response) of each sample is taken as the corresponding teacher feature such that $\mathcal{W} = \mathbb{R}^K$. In light of the empirical success of RKD based on simple kernel functions [Park et al., 2019, Liu et al., 2019], we focus on the (shifted) cosine kernel, $k_\psi(\mathbf{x}, \mathbf{x}') = 1 + \frac{\psi(\mathbf{x})^\top \psi(\mathbf{x}')}{\|\psi(\mathbf{x})\|_2 \|\psi(\mathbf{x}')\|_2}$, in the experiments.

With data augmentation (*e.g.*, when coupling with FixMatch), we assume that weak augmentations (*i.e.*, random flips and crops) bring little perturbations on responses of the teacher model (*i.e.*, the teacher features), intuitively as the predictions made by a good teacher model ought to be invariance under weak augmentations. Thereby, we can considerably reduce memory burden and accelerate the RKD process by storing the teacher features $\psi(\mathbf{x})$ of the original data $\mathbf{x} \in \mathcal{X}$ offline, instead of inferring every augmented sample $\mathbf{x}' \in \mathcal{A}(\mathbf{x})$ with the large teacher model $\psi(\mathbf{x}')$ online.

**Labeled data selection.** As alluded to in Remark 5.2, the success of low-label-rate (i.e., $n \ll N$) semi-supervised classification naturally depends on the quality of the (extremely) limited amount of labeled data to represent the underlying population. We perform experiments in which the selection of labeled data is non-adaptive (Uniform) or adaptive (StochasticGreedy). The non-adaptive setting, Uniform, selects labeled data uniformly in each ground truth class[11]. This contrasts with the strictly *i.i.d.* sampling assumption used in our theoretical results but constitutes a reasonable alternative that has been used in other low-label rate semi-supervised classification works (*e.g.*, [Calder et al., 2020]).

We also utilize a coreset selection strategy (StochasticGreedy) for identifying "representative" inputs to use as labeled data in an adaptive manner. This labeled set is selected via a stochastic greedy [Mirzasoleiman et al., 2015] optimization problem to (approximately) minimize a facility location objective[12]. An in-depth discussion of coreset selection methods is outside the scope of the current work, but we refer the interested reader to the enlightening references [Bilmes, 2022, Krause and Golovin, 2014, Mirzasoleiman, 2017] regarding submodular optimization methods in machine learning.

Table 1: Top-1 accuracy on CIFAR-10.

| Label selection | Sample sizes $(n, N)$ | SSL algorithm | Augmentation Strength | | |
| | | | *High* | *Medium* | *Weak* |
| --- | --- | --- | --- | --- | --- |
| Uniform | (40, 50000) | FixMatch | $89.47 \pm 3.83$ | $87.89 \pm 1.43$ | $81.36 \pm 1.41$ |
| | | FixMatch + RKD | $\mathbf{89.88 \pm 0.41}$ | $\mathbf{89.10 \pm 1.14}$ | $\mathbf{84.61 \pm 1.23}$ |
| | (40, 40000) | FixMatch | $87.60 \pm 1.21$ | $85.30 \pm 1.35$ | $83.74 \pm 3.07$ |
| | | FixMatch + RKD | $\mathbf{90.19 \pm 0.19}$ | $\mathbf{88.65 \pm 1.81}$ | $\mathbf{86.68 \pm 1.06}$ |
| | (40, 25000) | FixMatch | $81.73 \pm 1.27$ | $81.43 \pm 2.98$ | $76.84 \pm 1.98$ |
| | | FixMatch + RKD | $\mathbf{87.06 \pm 2.04}$ | $\mathbf{87.24 \pm 0.39}$ | $\mathbf{82.75 \pm 2.35}$ |
| StochasticGreedy | (40, 50000) | FixMatch | $85.48 \pm 1.67$ | $89.39 \pm 0.82$ | $78.98 \pm 0.52$ |
| | | FixMatch + RKD | $\mathbf{91.19 \pm 0.26}$ | $\mathbf{90.00 \pm 0.64}$ | $\mathbf{87.11 \pm 4.33}$ |
| | (40, 40000) | FixMatch | $84.51 \pm 1.34$ | $84.88 \pm 6.68$ | $70.70 \pm 5.32$ |
| | | FixMatch + RKD | $\mathbf{85.68 \pm 1.19}$ | $\mathbf{88.31 \pm 1.36}$ | $\mathbf{85.12 \pm 3.49}$ |
| | (40, 25000) | FixMatch | $79.74 \pm 8.02$ | $73.97 \pm 1.98$ | $73.86 \pm 3.29$ |
| | | FixMatch + RKD | $\mathbf{83.72 \pm 4.73}$ | $\mathbf{82.09 \pm 3.27}$ | $\mathbf{80.65 \pm 3.67}$ |

**Results.** From Table 1 and Table 2, we observe that the incorporation of RKD with FixMatch (*i.e.*, adding the "global" perspective of RKD to the "local" perspective of DAC) generally brings additional improvements to the classification accuracy, especially when the data augmentation strength is inappropriate and/or the number of unlabeled data is limited.

---

[11]For fair comparison and consistent performance among different configurations in the experiments, we use a separate random number generator with a fixed random seed to ensure that the same set of labeled samples is selected for all experiments.

[12]We use the Apricot [Schreiber et al., 2020] submodular optimization codebase in Python.

Table 2: Top-1 accuracy on CIFAR-100.

| Label selection | Sample sizes $(n, N)$ | SSL algorithm | Augmentation Strength | | |
| --- | --- | --- | --- | --- | --- |
| | | | *High* | *Medium* | *Weak* |
| Uniform | (400, 50000) | FixMatch | $46.85 \pm 1.68$ | $45.67 \pm 0.51$ | $45.84 \pm 0.35$ |
| | | FixMatch + RKD | $\mathbf{49.33 \pm 0.42}$ | $\mathbf{47.97 \pm 0.37}$ | $\mathbf{47.65 \pm 0.17}$ |
| | (400, 40000) | FixMatch | $44.11 \pm 0.51$ | $42.98 \pm 0.56$ | $42.74 \pm 0.42$ |
| | | FixMatch + RKD | $\mathbf{46.24 \pm 0.64}$ | $\mathbf{45.78 \pm 0.94}$ | $\mathbf{45.13 \pm 1.08}$ |
| | (400, 25000) | FixMatch | $36.60 \pm 0.03$ | $35.86 \pm 0.51$ | $33.59 \pm 1.54$ |
| | | FixMatch + RKD | $\mathbf{37.87 \pm 0.98}$ | $\mathbf{36.69 \pm 0.41}$ | $\mathbf{35.69 \pm 1.51}$ |
| StochasticGreedy | (400, 50000) | FixMatch | $47.45 \pm 0.30$ | $48.64 \pm 0.35$ | $46.79 \pm 1.05$ |
| | | FixMatch + RKD | $\mathbf{50.41 \pm 1.41}$ | $\mathbf{50.67 \pm 1.62}$ | $\mathbf{49.09 \pm 1.31}$ |
| | (400, 40000) | FixMatch | $45.93 \pm 0.21$ | $44.78 \pm 1.51$ | $44.62 \pm 0.72$ |
| | | FixMatch + RKD | $\mathbf{47.62 \pm 0.67}$ | $\mathbf{47.28 \pm 0.56}$ | $\mathbf{46.07 \pm 0.86}$ |
| | (400, 25000) | FixMatch | $39.40 \pm 0.31$ | $39.01 \pm 0.68$ | $38.22 \pm 0.42$ |
| | | FixMatch + RKD | $\mathbf{40.88 \pm 0.61}$ | $\mathbf{40.73 \pm 0.46}$ | $\mathbf{39.20 \pm 0.73}$ |

**Training details.**    Throughout the experiments, we used weight decay $0.0005$. We train the student model via stochastic gradient descent (SGD) with Nesterov momentum $0.9$ for $2^{17}$ iterations (batches) with a batch size $64 * 8 = 2^9$ (consisting of 64 labeled samples and $64 * 7$ unlabeled samples). The initial learning rate is $0.03$, decaying with a cosine scheduler. The test accuracies are evaluated $2^7$ times in total evenly throughout the $2^{17}$ iterations (*i.e.*, one test evaluation after every $2^{10}$ iterations) on an EMA model with a decay rate $0.999$. The average and standard deviation of the best test accuracy (*i.e.*, early stopping with the maximum patience 128) are reported for each experiment over 3 arbitrary random seeds. Both CIFAR-10/100 experiments are conducted on one NVIDIA A40 GPU.

# B    Proofs and Discussions for Section 5

## B.1    Proof of Theorem 5.1

**Lemma B.1** (Rademacher complexity with cluster-aware functions (generalization of Yang et al. [2023] Theorem 8))**.** *For a fixed $\mathcal{F}_{\mathbf{L}(\mathbf{X}^u)}$, let*

$$\ell \circ \mathcal{F}_{\mathbf{L}(\mathbf{X}^u)} = \left\{ \ell(f(\cdot), \cdot) : \mathcal{X} \times \mathcal{Y} \to \{0, 1\} \mid f \in \mathcal{F}_{\mathbf{L}(\mathbf{X}^u)} \right\}$$

*be the loss function class. Then, its Rademacher complexity can be upper-bounded by*

$$\mathfrak{R}_n \left( \ell \circ \mathcal{F}_{\mathbf{L}(\mathbf{X}^u)} \right) \leq \sqrt{\frac{2K \log(2n)}{n}} + 2\mu \left( \mathcal{F}_{\mathbf{L}(\mathbf{X}^u)} \right). \tag{10}$$

*Proof of Lemma B.1.*  Given a set of *i.i.d.* samples $(\mathbf{X}, \mathbf{y}) \sim P(\mathbf{x}, y)^n$, let

$$\widehat{\mathfrak{S}}_{(\mathbf{X}, \mathbf{y})} \left( \ell \circ \mathcal{F}_{\mathbf{L}(\mathbf{X}^u)} \right) \triangleq \left| \left\{ \left( \ell\left( f(\mathbf{x}_1), y_1 \right), \ldots, \ell\left( f(\mathbf{x}_n), y_n \right) \right) \mid f \in \mathcal{F}_{\mathbf{L}(\mathbf{X}^u)} \right\} \right|.$$

denote the number of distinct patterns over $(\mathbf{X}, \mathbf{y})$ in $\ell \circ \mathcal{F}_{\mathbf{L}(\mathbf{X}^u)}$. Then, by Massart's finite lemma, the empirical Rademacher complexity with respect to $(\mathbf{X}, \mathbf{y})$ is upper bounded by

$$\widehat{\mathfrak{R}}_{(\mathbf{X}, \mathbf{y})} \left( \ell \circ \mathcal{F}_{\mathbf{L}(\mathbf{X}^u)} \right) \leq \sqrt{\frac{2 \log \left( \widehat{\mathfrak{S}}_{(\mathbf{X}, \mathbf{y})} \left( \ell \circ \mathcal{F}_{\mathbf{L}(\mathbf{X}^u)} \right) \right)}{n}}.$$

By the concavity of $\sqrt{\log(\cdot)}$, we know that,

$$
\begin{aligned}
\mathfrak{R}_n\left(\ell \circ \mathcal{F}_{\mathbf{L}(\mathbf{X}^u)}\right) =& \mathbb{E}_{(\mathbf{X},\mathbf{y})}\left[\widehat{\mathfrak{R}}_{(\mathbf{X},\mathbf{y})}\left(\ell \circ \mathcal{F}_{\mathbf{L}(\mathbf{X}^u)}\right)\right] \\
\leq& \mathbb{E}_{(\mathbf{X},\mathbf{y})}\left[\sqrt{\frac{2\log\left(\widehat{\mathfrak{S}}_{(\mathbf{X},\mathbf{y})}\left(\ell \circ \mathcal{F}_{\mathbf{L}(\mathbf{X}^u)}\right)\right)}{n}}\right] \\
\leq& \sqrt{\frac{2\log\left(\mathbb{E}_{(\mathbf{X},\mathbf{y})}\left[\widehat{\mathfrak{S}}_{(\mathbf{X},\mathbf{y})}\left(\ell \circ \mathcal{F}_{\mathbf{L}(\mathbf{X}^u)}\right)\right]\right)}{n}}.
\end{aligned}
\tag{11}
$$

Since $P\left(M(f)\right) \leq \mu\left(\mathcal{F}_{\mathbf{L}(\mathbf{X}^u)}\right) \leq \frac{1}{2}$ for all $f \in \mathcal{F}_{\mathbf{L}(\mathbf{X}^u)}$, by union bound, we have

$$
\begin{aligned}
& \mathbb{E}_{(\mathbf{X},\mathbf{y})}\left[\widehat{\mathfrak{S}}_{(\mathbf{X},\mathbf{y})}\left(\ell \circ \mathcal{F}_{\mathbf{L}(\mathbf{X}^u)}\right)\right] \\
\leq& \sum_{\iota=0}^{n}\binom{n}{\iota}\mu\left(\mathcal{F}_{\mathbf{L}(\mathbf{X}^u)}\right)^{\iota}\left(1-\mu\left(\mathcal{F}_{\mathbf{L}(\mathbf{X}^u)}\right)\right)^{n-\iota} \cdot \binom{n-\iota+1}{\min\{K, n-\iota\}-1}2^{K+\iota} \\
\leq& (2n)^K\sum_{\iota=0}^{n}\binom{n}{\iota}\left(2\mu\left(\mathcal{F}_{\mathbf{L}(\mathbf{X}^u)}\right)\right)^{\iota}\left(1-\mu\left(\mathcal{F}_{\mathbf{L}(\mathbf{X}^u)}\right)\right)^{n-\iota} \\
=& (2n)^K\left(1-\mu\left(\mathcal{F}_{\mathbf{L}(\mathbf{X}^u)}\right)+2\mu\left(\mathcal{F}_{\mathbf{L}(\mathbf{X}^u)}\right)\right)^n \\
\leq& (2n)^K e^{n\mu\left(\mathcal{F}_{\mathbf{L}(\mathbf{X}^u)}\right)},
\end{aligned}
\tag{12}
$$

where $\iota$ counts the number of minority samples in the $n$ samples. Plugging in this last line into Equation (11) yields Equation (10). $\qquad\square$

*Proof of Theorem 5.1.* Since $\ell \circ \mathcal{F}_{\mathbf{L}(\mathbf{X}^u)}$ is 1-bounded with the zero-one loss $\ell$, Lemma F.1 implies that with probability at least $1 - \delta/2$ over $(\mathbf{X}, \mathbf{y})$,

$$
\mathcal{E}\left(\widehat{f}_{|\mathbf{X}^u}\right) - \mathcal{E}\left(f_*\right) \leq 4\mathfrak{R}_n\left(\ell \circ \mathcal{F}_{\mathbf{L}(\mathbf{X}^u)}\right) + \sqrt{\frac{2\log\left(4/\delta\right)}{n}}.
$$

Then, incorporating the upper bound of $\mathfrak{R}_n\left(\ell \circ \mathcal{F}_{\mathbf{L}(\mathbf{X}^u)}\right)$ from Lemma B.1 completes the proof. $\quad\square$

### B.2 Cluster-wise Labeling with Cluster-aware Predictions

Recall the notions of majority labeling $\widetilde{y}_f(\cdot)$ (Equation (2)) and minority subset $M(f)$ (Definition 3.1). We first specify the cluster-aware predictions that are useful for labeling with the following non-degeneracy assumption.

**Assumption B.1** $((m_0, c_0)$-non-degenerate predictions)**.** *For any prediction function $f \in \mathcal{F}$, we say $f$ is $(m_0, c_0)$-non-degenerate if the following properties are satisfied:*

*(i) $y_f : \mathcal{X} \to [K]$ is surjective (i.e., $\{y_f(\mathbf{x}) \mid \mathbf{x} \in \mathcal{X}\} = [K]$)*

*(ii) $\left|\mathcal{X}_k^f\right| \geq m_0$ for all $k \in [K]$, where we define $\mathcal{X}_k^f \triangleq \{\mathbf{x} \in \mathcal{X} \mid \widetilde{y}_f(\mathbf{x}) = k\}$*

*(iii) There exists some $c_0 \geq 2$ such that $c_0 \cdot P\left(M(f) \cap \mathcal{X}_k^f\right) \leq P\left(\mathcal{X}_k^f\right)$ for all $k \in [K]$.*

It is worth pointing out that although the majority labeling $\widetilde{y}_f(\cdot)$ depends on the ground truth and therefore remains unknown without label acquisition, under Assumption B.1, the associated partition $\left\{\mathcal{X}_k^f\right\}_{k\in[K]}$ over $\mathcal{X}$ depends only on $f$ and is known without labeling. This is because $y_f(\cdot)$ and $\widetilde{y}_f(\cdot)$ induce the same partition over $\mathcal{X}$ when both $y_f(\cdot)$ and $\widetilde{y}_f(\cdot)$ are non-degenerate (*i.e.*, being surjective functions onto $[K]$, a necessary condition of Assumption B.1 (i) and (ii)). We also notice that Assumption B.1 (iii) can be generally satisfied with a reasonably large $c$ when learning via Equation (3) with a sufficiently small $\mu\left(\mathcal{F}_{\mathbf{L}(\mathbf{X}^u)}\right)$ (*cf.* Theorem 4.1 or Theorem 4.3). For example, when $c_0 \cdot \mu\left(\mathcal{F}_{\mathbf{L}(\mathbf{X}^u)}\right) \leq \min_{k \in K} P\left(\mathcal{X}_k^f\right)$, Assumption B.1 (iii) is automatically satisfied.

With the cluster-aware predictions in Assumption B.1, we show that $O\left(\log(K)\right)$ *i.i.d.* labeled samples per predicted cluster are sufficient to ensure at least one labeled sample per class. That is, the cluster-wise labeling with $(m_0, c_0)$-non-degenerate predictions guarantees a label complexity of $O\left(K\log(K)\right)$ even with class imbalance.

**Proposition B.2** (Cluster-wise labeling). *Given any $(m_0, c_0)$-non-degenerate prediction function $f \in \mathcal{F}$ (Assumption B.1) with sufficiently large $m_0 \gg \log_{c_0}(2K)$ and $c_0 \geq 2$, for each $k \in [K]$, acquire labels of $m$ ($m \leq m_0$) unlabeled samples $\{\mathbf{x}_{ki}^u\}_{i \in [m]} \sim \mathrm{Unif}\left(\mathcal{X}_k^f\right)^m$ drawn i.i.d. from $\mathcal{X}_k^f$. Then, for any $\delta \in \left(\frac{2K}{c_0^{m_0}}, 1\right)$, when $m \geq \log_{c_0}\left(\frac{2K}{\delta}\right)$, $\{y_*\left(\mathbf{x}_{ki}^u\right)\}_{k \in [K], i \in [m]} = [K]$ have at least one ground truth label per class with probability at least $1 - \delta$.*

*Proof of Proposition B.2.* We first observe that

$$
\begin{aligned}
\mathbb{P}\left[\{y_*\left(\mathbf{x}_{ki}^u\right)\}_{k \in [K], i \in [m]} = [K]\right] = & \mathbb{P}\left[\forall\, k' \in [K],\ \exists\, k \in [K], i \in [m] \text{ s.t. } y_*\left(\mathbf{x}_{ki}^u\right) = k'\right] \\
\geq & \mathbb{P}\left[\forall\, k \in [K],\ \exists\, i \in [m] \text{ s.t. } y_*\left(\mathbf{x}_{ki}^u\right) = k\right] \\
= & \prod_{k=1}^{K} \mathbb{P}_{\{\mathbf{x}_{ki}^u\}_{i \in [m]} \sim \mathrm{Unif}\left(\mathcal{X}_k^f\right)^m}\left[\exists\, i \in [m] \text{ s.t. } y_*\left(\mathbf{x}_{ki}^u\right) = k\right],
\end{aligned}
$$

where

$$
\begin{aligned}
\mathbb{P}\left[\exists\, i \in [m] \text{ s.t. } y_*\left(\mathbf{x}_{ki}^u\right) = k\right] = & 1 - \mathbb{P}\left[\forall\, i \in [m],\ y_*\left(\mathbf{x}_{ki}^u\right) \neq k\right] \\
= & 1 - \left(\mathbb{P}_{\mathbf{x} \sim \mathrm{Unif}\left(\mathcal{X}_k^f\right)}\left[y_*\left(\mathbf{x}\right) \neq \widetilde{y}_f\left(\mathbf{x}\right)\right]\right)^m \\
= & 1 - \left(\mathbb{P}_{\mathbf{x} \sim \mathrm{Unif}\left(\mathcal{X}_k^f\right)}\left[\mathbf{x} \in M(f)\right]\right)^m \\
= & 1 - \left(\frac{P\left(M(f) \cap \mathcal{X}_k^f\right)}{P\left(\mathcal{X}_k^f\right)}\right)^m.
\end{aligned}
$$

Since Assumption B.1 implies $P\left(M(f) \cap \mathcal{X}_k^f\right)\big/P\left(\mathcal{X}_k^f\right) \leq \frac{1}{c_0}$ for all $k \in [K]$, we have

$$
\mathbb{P}\left[\{y_*\left(\mathbf{x}_{ki}^u\right)\}_{k \in [K], i \in [m]} = [K]\right] \geq \left(1 - \left(\frac{1}{c_0}\right)^m\right)^K \geq \exp\left(-\frac{2K}{c_0^m}\right) \geq 1 - \frac{2K}{c_0^m},
$$

where the second and the third inequalities follow from the facts that $1 - q \geq \exp\left(-2q\right)$ for all $0 \leq q \leq \frac{1}{2}$ and $\exp\left(-q\right) \geq 1 - q$ for all $0 \leq q \leq 1$, respectively. It is straightforward to observe that when $m \geq \log_{c_0}\left(\frac{2K}{\delta}\right)$, we have $\frac{2K}{c_0^m} \leq \delta$. With $\min_{k \in [K]}\left|\mathcal{X}_k^f\right| = m_0$, taking up to $m \leq m_0$ unlabeled samples for each $k \in [K]$, we can achieve $\delta$ as small as $\delta_{\min} = \frac{2K}{c_0^{m_0}}$. $\qquad\square$

## C   Proofs and Discussions for Section 4.1

In Appendix C.1, we proceed with the proof of low clustering error guarantee (Theorem 4.1) for minimizing the population RKD loss (Equation (1)). In Appendix C.2, we extend the discussion in Remark 4.1 on the limitation of spectral clustering alone in the end-to-end setting via an illustrative toy counter-example. In Appendix C.3, we clarify the connection between our results and corresponding spectral graph theoretic notions of sparsest $k$-partitions.

### C.1   Proof of Theorem 4.1

Given any labeling function $y : \mathcal{X} \to [K]$, let $\overrightarrow{y} : \mathcal{X} \to \{0, 1\}^K$ be the one-hot encoded labeling function such that for $y\left(\mathcal{X}\right) \in [K]^{|\mathcal{X}|}$, $\overrightarrow{y}\left(\mathcal{X}\right) \in \{0, 1\}^{|\mathcal{X}| \times K}$. Further, given $f \in \mathcal{F}$, we denote $\mathbf{U}_{\mathcal{X}} \triangleq \mathbf{D}\left(\mathcal{X}\right)^{1/2}\overrightarrow{\widetilde{y}_f}\left(\mathcal{X}\right)$, $\mathbf{F}_{\mathcal{X}} \triangleq \mathbf{D}\left(\mathcal{X}\right)^{1/2} f\left(\mathcal{X}\right)$, and $\mathbf{Y}_{\mathcal{X}} \triangleq \mathbf{D}\left(\mathcal{X}\right)^{1/2}\overrightarrow{y_*}\left(\mathcal{X}\right)$ such that $\mathbf{U}_{\mathcal{X}}, \mathbf{F}_{\mathcal{X}}, \mathbf{Y}_{\mathcal{X}} \in \mathbb{R}^{|\mathcal{X}| \times K}$.

Since $\mathbf{L}(\mathcal{X}) = \mathbf{I} - \overline{\mathbf{W}}(\mathcal{X})$, the classical argument with Gershgorin circle theorem [Golub and Van Loan, 2013, Theorem 7.2.1] implies that $0 \preceq \mathbf{L}(\mathcal{X}) \preceq 2\mathbf{I}$, whereas $k_\psi$ being positive semi-definite further implies that $\overline{\mathbf{W}}(\mathcal{X}) \succeq 0$ and therefore $0 \preceq \mathbf{L}(\mathcal{X}) \preceq \mathbf{I}$ (see the proof of Claim C.3). Consider the spectral decomposition of $\mathbf{L}(\mathcal{X}) = \mathbf{I} - \overline{\mathbf{W}}(\mathcal{X})$,

$$\mathbf{L}(\mathcal{X}) = \sum_{i=1}^{|\mathcal{X}|} \lambda_i \cdot \mathbf{v}_i \mathbf{v}_i^\top \quad \text{s.t.} \quad \mathbf{D}(\mathcal{X})^{-1/2} \mathbf{W}(\mathcal{X}) \mathbf{D}(\mathcal{X})^{-1/2} = \sum_{i=1}^{|\mathcal{X}|} (1 - \lambda_i) \cdot \mathbf{v}_i \mathbf{v}_i^\top, \quad (13)$$

where $\left\{ \mathbf{v}_i \in \mathbb{R}^{|\mathcal{X}|} \right\}_{i \in [|\mathcal{X}|]}$ are the orthonormal eigenvectors associated with $0 = \lambda_1 \leq \cdots \leq \lambda_{|\mathcal{X}|} \leq 1$, respectively, that form a basis of $\mathbb{R}^{|\mathcal{X}|}$.

**Lemma C.1.** *Given any $f \in \mathcal{F}$ that satisfies Assumption 4.2, we have*

$$P(M(f)) = \mathbb{E}_{\mathbf{x} \sim P}\left[\mathbb{1}\{\widetilde{y}_f(\mathbf{x}) \neq y_*(\mathbf{x})\}\right] \leq 2 \max\left(\frac{\beta^2}{\gamma^2}, 1\right) \cdot \min_{\mathbf{Z} \in \mathbb{R}^{K \times K}} \|\mathbf{Y}_\mathcal{X} - \mathbf{F}_\mathcal{X} \mathbf{Z}\|_F^2.$$

*Proof of Lemma C.1.* Without ambiguity, we overload the notation of each sample $\mathbf{x} \in \mathcal{X}$ as the corresponding index in $[|\mathcal{X}|]$ such that, given any sample subset $S = \{\mathbf{s}_i\}_{i \in [|S|]} \subset \mathcal{X}$, following the MATLAB notation for matrix indexing, $\mathbf{\Pi}_S \triangleq \mathbf{I}_{|\mathcal{X}|}(:, S) \in \mathbb{R}^{|\mathcal{X}| \times |S|}$ where $\mathbf{I}_{|\mathcal{X}|}$ denotes the $|\mathcal{X}| \times |\mathcal{X}|$ identity matrix. With respect to $S$, we also denote $\mathbf{F}_S \triangleq \mathbf{F}_\mathcal{X}(S, :) = \mathbf{\Pi}_S^\top \mathbf{F}_\mathcal{X}$ and $\mathbf{Y}_S \triangleq \mathbf{Y}_\mathcal{X}(S, :) = \mathbf{\Pi}_S^\top \mathbf{Y}_\mathcal{X}$.

We first show that for any $\mathbf{S} = [\mathbf{s}_1; \ldots; \mathbf{s}_K] \in \mathcal{X}^K$ that satisfies $\operatorname{rank}(\mathbf{F}_S) = K$,

$$\left\|\mathbf{Y}_\mathcal{X} - \mathbf{F}_\mathcal{X} \mathbf{F}_S^{-1} \mathbf{Y}_S\right\|_F^2 \leq 2 \min_{\mathbf{Z} \in \mathbb{R}^{K \times K}} \|\mathbf{Y}_\mathcal{X} - \mathbf{F}_\mathcal{X} \mathbf{Z}\|_F^2. \quad (14)$$

To see this, we start by replacing $\mathbf{Z}$ on the right-hand side of Equation (14) with the corresponding least square solution such that

$$\min_{\mathbf{Z} \in \mathbb{R}^{K \times K}} \|\mathbf{Y}_\mathcal{X} - \mathbf{F}_\mathcal{X} \mathbf{Z}\|_F^2 = \left\|\mathbf{Y}_\mathcal{X} - \mathbf{F}_\mathcal{X} \mathbf{F}_\mathcal{X}^\dagger \mathbf{Y}_\mathcal{X}\right\|_F^2$$

where $\mathbf{F}_\mathcal{X}^\dagger = \left(\mathbf{F}_\mathcal{X}^\top \mathbf{F}_\mathcal{X}\right)^{-1} \mathbf{F}_\mathcal{X}^\top$. Then, we can decompose the left-hand side of Equation (14) via the orthogonal projection $\mathbf{F}_\mathcal{X} \mathbf{F}_\mathcal{X}^\dagger$:

$$\left\|\mathbf{Y}_\mathcal{X} - \mathbf{F}_\mathcal{X} \mathbf{F}_S^{-1} \mathbf{Y}_S\right\|_F^2 = \left\|\left(\mathbf{I} - \mathbf{F}_\mathcal{X} \mathbf{F}_\mathcal{X}^\dagger\right)\left(\mathbf{Y}_\mathcal{X} - \mathbf{F}_\mathcal{X} \mathbf{F}_S^{-1} \mathbf{Y}_S\right)\right\|_F^2 + \left\|\mathbf{F}_\mathcal{X} \mathbf{F}_\mathcal{X}^\dagger \left(\mathbf{Y}_\mathcal{X} - \mathbf{F}_\mathcal{X} \mathbf{F}_S^{-1} \mathbf{Y}_S\right)\right\|_F^2$$

$$= \left\|\mathbf{Y}_\mathcal{X} - \mathbf{F}_\mathcal{X} \mathbf{F}_\mathcal{X}^\dagger \mathbf{Y}_\mathcal{X}\right\|_F^2 + \|(\mathbf{P}_F - \mathbf{P}_S) \mathbf{Y}_\mathcal{X}\|_F^2,$$

where $\mathbf{P}_F \triangleq \mathbf{F}_\mathcal{X} \mathbf{F}_\mathcal{X}^\dagger = \mathbf{F}_\mathcal{X}\left(\mathbf{F}_\mathcal{X}^\top \mathbf{F}_\mathcal{X}\right)^{-1} \mathbf{F}_\mathcal{X}^\top$ and $\mathbf{P}_S \triangleq \mathbf{F}_\mathcal{X}\left(\mathbf{\Pi}_S^\top \mathbf{F}_\mathcal{X}\right)^{-1} \mathbf{\Pi}_S^\top$. By observing that

$$\mathbf{P}_S \mathbf{P}_F = \mathbf{F}_\mathcal{X}\left(\mathbf{\Pi}_S^\top \mathbf{F}_\mathcal{X}\right)^{-1} \mathbf{\Pi}_S^\top \mathbf{F}_\mathcal{X}\left(\mathbf{F}_\mathcal{X}^\top \mathbf{F}_\mathcal{X}\right)^{-1} \mathbf{F}_\mathcal{X}^\top = \mathbf{P}_F = \mathbf{P}_F \mathbf{P}_F,$$

we can upper bound $\|(\mathbf{P}_F - \mathbf{P}_S) \mathbf{Y}_\mathcal{X}\|_F^2$ such that

$$\|(\mathbf{P}_F - \mathbf{P}_S) \mathbf{Y}_\mathcal{X}\|_F^2 = \|(\mathbf{P}_F - \mathbf{P}_S)(\mathbf{I} - \mathbf{P}_F) \mathbf{Y}_\mathcal{X}\|_F^2$$

$$\leq \|(\mathbf{I} - \mathbf{P}_F) \mathbf{Y}_\mathcal{X}\|_F^2$$

$$= \left\|\mathbf{Y}_\mathcal{X} - \mathbf{F}_\mathcal{X} \mathbf{F}_\mathcal{X}^\dagger \mathbf{Y}_\mathcal{X}\right\|_F^2,$$

which leads to Equation (14).

Now, we consider the sample subset $\mathbf{S} \in \mathcal{X}^K$ described in Assumption 4.2. For given $f \in \mathcal{F}$, partitioning $\mathcal{X}$ into the majority and minority subsets, $\mathcal{X} \backslash M(f)$ and $M(f)$, respectively yields

$$\left\|\mathbf{Y}_\mathcal{X} - \mathbf{F}_\mathcal{X} \mathbf{F}_S^{-1} \mathbf{Y}_S\right\|_F^2$$

$$= (1 - P(M(f))) \cdot \mathbb{E}_{\mathbf{x} \sim P(\mathbf{x})}\left[\left\|\overrightarrow{y_*}(\mathbf{x}) - \overrightarrow{y_*}(\mathbf{S})^\top f(\mathbf{S})^{-\top} f(\mathbf{x})\right\|_2^2 \,\Big|\, \mathbf{x} \notin M(f)\right]$$

$$+ P(M(f)) \cdot \mathbb{E}_{\mathbf{x} \sim P(\mathbf{x})}\left[\left\|\overrightarrow{y_*}(\mathbf{x}) - \overrightarrow{y_*}(\mathbf{S})^\top f(\mathbf{S})^{-\top} f(\mathbf{x})\right\|_2^2 \,\Big|\, \mathbf{x} \in M(f)\right]$$

$$\geq P(M(f)) \cdot \mathbb{E}_{\mathbf{x} \sim P(\mathbf{x})}\left[\left\|\overrightarrow{y_*}(\mathbf{x}) - \overrightarrow{y_*}(\mathbf{S})^\top f(\mathbf{S})^{-\top} f(\mathbf{x})\right\|_2^2 \,\Big|\, \mathbf{x} \in M(f)\right].$$

Since $\mathbf{s}_k \notin M(f)$, we have $\overrightarrow{y_*}(\mathbf{S}) = \overrightarrow{\widetilde{y}_f}(\mathbf{S})$ whose rows lie in the canonical bases of $\{\mathbf{e}_k \in \mathbb{R}^K\}_{k \in [K]}$; whereas for $\mathbf{x} \in M(f)$, we have $y_*(\mathbf{x}) \neq \widetilde{y}_f(\mathbf{x})$. Therefore, in the case when $y_*(\mathbf{x}) = y_*(\mathbf{s}_k)$ for some $k \in [K]$,

$$\left\| \overrightarrow{y_*}(\mathbf{x}) - \overrightarrow{y_*}(\mathbf{S})^\top f(\mathbf{S})^{-\top} f(\mathbf{x}) \right\|_2^2$$

$$= \left\| f(\mathbf{S})^{-\top} \left( f(\mathbf{S})^\top \overrightarrow{y_*}(\mathbf{S}) \overrightarrow{y_*}(\mathbf{x}) - f(\mathbf{x}) \right) \right\|_2^2$$

$$\geq \sigma_K \left( f(\mathbf{S})^{-\top} \right)^2 \cdot \left\| f(\mathbf{x}) - \sum_{k \in [K]: y_*(\mathbf{x}) = y_*(\mathbf{s}_k)} f(\mathbf{s}_k) \right\|_2^2,$$

where $\sigma_K \left( f(\mathbf{S})^{-\top} \right) = 1/\beta$, and with $\widetilde{y}_f(\mathbf{s}_k) = y_*(\mathbf{s}_k) = y_*(\mathbf{x}) \neq \widetilde{y}_f(\mathbf{x})$ implying $y_f(\mathbf{s}_k) \neq y_f(\mathbf{x})$,

$$\left\| f(\mathbf{x}) - \sum_{k \in [K]: y_*(\mathbf{x}) = y_*(\mathbf{s}_k)} f(\mathbf{s}_k) \right\|_2^2 \geq \gamma^2,$$

we have

$$\left\| \overrightarrow{y_*}(\mathbf{x}) - \overrightarrow{y_*}(\mathbf{S})^\top f(\mathbf{S})^{-\top} f(\mathbf{x}) \right\|_2^2 \geq \frac{\gamma^2}{\beta^2}.$$

Meanwhile, in the case when $y_*(\mathbf{x}) \neq y_*(\mathbf{s}_k)$ for all $k \in [K]$, we have $\overrightarrow{y_*}(\mathbf{S}) \overrightarrow{y_*}(\mathbf{x}) = \mathbf{0}$, and therefore

$$\left\| \overrightarrow{y_*}(\mathbf{x}) - \overrightarrow{y_*}(\mathbf{S})^\top f(\mathbf{S})^{-\top} f(\mathbf{x}) \right\|_2^2 \geq \left\| \overrightarrow{y_*}(\mathbf{x})^\top \left( \overrightarrow{y_*}(\mathbf{x}) - \overrightarrow{y_*}(\mathbf{S})^\top f(\mathbf{S})^{-\top} f(\mathbf{x}) \right) \right\|_2^2 = 1.$$

Overall, we've shown that

$$\mathbb{E}_{\mathbf{x} \sim P(\mathbf{x})} \left[ \left\| \overrightarrow{y_*}(\mathbf{x}) - \overrightarrow{y_*}(\mathbf{S})^\top f(\mathbf{S})^{-\top} f(\mathbf{x}) \right\|_2^2 \,\Big|\, \mathbf{x} \in M(f) \right] \geq \min \left( \frac{\gamma^2}{\beta^2}, 1 \right),$$

which leads to

$$P(M(f)) \leq \max \left( \frac{\beta^2}{\gamma^2}, 1 \right) \left\| \mathbf{Y}_{\mathcal{X}} - \mathbf{F}_{\mathcal{X}} \mathbf{F}_S^{-1} \mathbf{Y}_S \right\|_F^2. \tag{15}$$

Finally, combining Equation (14) and Equation (15) completes the proof. $\square$

**Lemma C.2.** *With $\mathbf{y}_k \in [0,1]^{|\mathcal{X}|}$ denoting the $k$-th column of $\mathbf{Y}_{\mathcal{X}} \triangleq \mathbf{D}(\mathcal{X})^{1/2} \overrightarrow{y_*}(\mathcal{X}) \in [0,1]^{|\mathcal{X}| \times K}$,*

$$\sum_{k \in [K]} \mathbf{y}_k^\top \mathbf{L}(\mathcal{X}) \mathbf{y}_k = \alpha.$$

*Proof of Lemma C.2.* It is sufficient to observe that, by construction, for every $k \in [K]$,

$$\mathbf{y}_k^\top \mathbf{L}(\mathcal{X}) \mathbf{y}_k = \mathbf{y}_k^\top \mathbf{y}_k - \mathbf{y}_k^\top \mathbf{D}(\mathcal{X})^{-1/2} \mathbf{W}(\mathcal{X}) \mathbf{D}(\mathcal{X})^{-1/2} \mathbf{y}_k$$

$$= \sum_{\mathbf{x} \in \mathcal{X}} w_{\mathbf{x}} \cdot \mathbb{1}\{y_*(\mathbf{x}) = k\} - \sum_{\mathbf{x} \in \mathcal{X}} \sum_{\mathbf{x}' \in \mathcal{X}} w_{\mathbf{x}\mathbf{x}'} \cdot \mathbb{1}\{y_*(\mathbf{x}) = k\} \cdot \mathbb{1}\{y_*(\mathbf{x}') = k\}$$

$$= \sum_{\mathbf{x} \in \mathcal{X}} \mathbb{1}\{y_*(\mathbf{x}) = k\} \left( w_{\mathbf{x}} - \sum_{\mathbf{x}' \in \mathcal{X}} w_{\mathbf{x}\mathbf{x}'} \cdot \mathbb{1}\{y_*(\mathbf{x}) = y_*(\mathbf{x}')\} \right)$$

$$= \sum_{\mathbf{x} \in \mathcal{X}} \mathbb{1}\{y_*(\mathbf{x}) = k\} \sum_{\mathbf{x}' \in \mathcal{X}} w_{\mathbf{x}\mathbf{x}'} \cdot \mathbb{1}\{y_*(\mathbf{x}) \neq y_*(\mathbf{x}')\}.$$

Meanwhile, since $\sum_{\mathbf{x} \in \mathcal{X}} \sum_{\mathbf{x}' \in \mathcal{X}} w_{\mathbf{x}\mathbf{x}'} = 1$, we have $\alpha = \frac{1}{2} \sum_{k \in [K]} \sum_{\mathbf{x} \in \mathcal{X}_k} \sum_{\mathbf{x}' \notin \mathcal{X}_k} w_{\mathbf{x}\mathbf{x}'}$ and

$$\sum_{k \in [K]} \mathbf{y}_k^\top \mathbf{L}(\mathcal{X}) \mathbf{y}_k = \sum_{\mathbf{x} \in \mathcal{X}} \sum_{\mathbf{x}' \in \mathcal{X}} w_{\mathbf{x}\mathbf{x}'} \cdot \mathbb{1}\{y_*(\mathbf{x}) \neq y_*(\mathbf{x}')\} \sum_{k \in [K]} \mathbb{1}\{y_*(\mathbf{x}) = k\}$$

$$= \sum_{\mathbf{x} \in \mathcal{X}} \sum_{\mathbf{x}' \in \mathcal{X}} w_{\mathbf{x}\mathbf{x}'} \cdot \mathbb{1}\{y_*(\mathbf{x}) \neq y_*(\mathbf{x}')\}$$

$$= \frac{1}{2} \sum_{k \in [K]} \sum_{\mathbf{x} \in \mathcal{X}_k} \sum_{\mathbf{x}' \notin \mathcal{X}_k} w_{\mathbf{x}\mathbf{x}'} = \alpha.$$

$\square$

**Claim C.3.** *For all* $f \in \mathcal{F}_{\mathbf{L}(\mathcal{X})}$ *minimizing Equation* (1), $\text{Range}\left(\mathbf{D}(\mathcal{X})^{\frac{1}{2}} f(\mathcal{X})\right) = \text{span}\{\mathbf{v}_1, \ldots, \mathbf{v}_K\}$.

*Proof of Claim C.3.* Since $0 \preccurlyeq \mathbf{L}(\mathcal{X}) \preccurlyeq 2\mathbf{I}$, we have $-\mathbf{I} \preccurlyeq \overline{\mathbf{W}}(\mathcal{X}) \preccurlyeq \mathbf{I}$. Meanwhile, let $k_\psi(\mathcal{X}, \mathcal{X}) \in \mathbb{R}^{|\mathcal{X}| \times |\mathcal{X}|}$ be the population kernel matrix such that $k_\psi(\mathcal{X}, \mathcal{X})_{\mathbf{xx}'} = k_\psi(\mathbf{x}, \mathbf{x}')$. Then by defintion, $k_\psi(\mathcal{X}, \mathcal{X}) = \mathbf{D}(\mathcal{X})^{-1}\mathbf{W}(\mathcal{X})\mathbf{D}(\mathcal{X})^{-1}$, and therefore

$$\mathbf{I} \succcurlyeq \overline{\mathbf{W}}(\mathcal{X}) = \mathbf{D}(\mathcal{X})^{-\frac{1}{2}}\mathbf{W}(\mathcal{X})\mathbf{D}(\mathcal{X})^{-\frac{1}{2}} = \mathbf{D}(\mathcal{X})^{\frac{1}{2}}k_\psi(\mathcal{X}, \mathcal{X})\mathbf{D}(\mathcal{X})^{\frac{1}{2}} \succcurlyeq 0.$$

That is, the eigenvalues of $\mathbf{L}(\mathcal{X})$ and those of $\overline{\mathbf{W}}(\mathcal{X})$ satisfies

$$\lambda_i \triangleq \lambda_i\left(\mathbf{L}(\mathcal{X})\right) = 1 - \lambda_{|\mathcal{X}|-i+1}\left(\overline{\mathbf{W}}(\mathcal{X})\right) \quad \forall\, i \in [|\mathcal{X}|],$$

while the eigenvectors are shared such that, with the spectral decomposition of $\mathbf{L}(\mathcal{X})$ in Equation (13),

$$\overline{\mathbf{W}}(\mathcal{X}) = \sum_{i=1}^{|\mathcal{X}|} (1 - \lambda_i) \cdot \mathbf{v}_i \mathbf{v}_i^\top.$$

Therefore, the $K$ eigenvalues of $\overline{\mathbf{W}}(\mathcal{X})$ of maximal modulus correspond exactly to the $K$ eigenvalues of $\mathbf{L}(\mathcal{X})$ that are closest to 0, the eigenvalues whose eigenspaces are relevant for spectral clustering.

Recalling $\mathbf{F}_\mathcal{X} \triangleq \mathbf{D}(\mathcal{X})^{\frac{1}{2}} f(\mathcal{X})$, we can rewrite Equation (1) as

$$R(f) = \left\|\overline{\mathbf{W}}(\mathcal{X}) - \mathbf{F}_\mathcal{X}\mathbf{F}_\mathcal{X}^\top\right\|_F^2.$$

It follows directly from the Eckart-Young-Mirsky theorem [Eckart and Young, 1936] that $\text{Range}(\mathcal{F}_\mathcal{X})$ is the subspace spanned by the eigenvectors $\mathbf{V}_K = [\mathbf{v}_1, \ldots, \mathbf{v}_K] \in \mathbb{R}^{|\mathcal{X}| \times K}$ associated with the maximum $K$ eigenvalues of $\overline{\mathbf{W}}(\mathcal{X})$:

$$\mathcal{F}_{\mathbf{L}(\mathcal{X})} = \left\{ f \in \mathcal{F} \,\middle|\, \exists\, \mathbf{Z} \in \mathbb{R}^{K \times K} \text{ s.t. } \mathbf{D}(\mathcal{X})^{\frac{1}{2}} f(\mathcal{X}) = \mathbf{V}_K \mathbf{Z} \right\}.$$

$\square$

*Proof of Theorem 4.1.* Recall $\mathbf{F}_\mathcal{X} \triangleq \mathbf{D}(\mathcal{X})^{1/2} f(\mathcal{X})$, $\mathbf{Y}_\mathcal{X} \triangleq \mathbf{D}(\mathcal{X})^{1/2} \overrightarrow{y_*}(\mathcal{X})$, and let $\mathbf{y}_k \in [0,1]^{|\mathcal{X}|}$ be the $k$-th column of $\mathbf{Y}_\mathcal{X}$. Leveraging Lemma C.1, we have for any $f \in \mathcal{F}$,

$$P(M(f)) \leq 2\max\left(\frac{\beta^2}{\gamma^2}, 1\right) \min_{\mathbf{Z} \in \mathbb{R}^{K \times K}} \|\mathbf{Y}_\mathcal{X} - \mathbf{F}_\mathcal{X}\mathbf{Z}\|_F^2$$

$$= 2\max\left(\frac{\beta^2}{\gamma^2}, 1\right) \min_{\{\mathbf{z}_k \in \mathbb{R}^K\}_{k \in [K]}} \sum_{k \in [K]} \|\mathbf{y}_k - \mathbf{F}_\mathcal{X}\mathbf{z}_k\|_2^2.$$

Recall that the orthonormal eigenvectors $\{\mathbf{v}_i \in \mathbb{R}^{|\mathcal{X}|}\}_{i \in [|\mathcal{X}|]}$ associated with eigenvalues $0 = \lambda_1 \leq \cdots \leq \lambda_{|\mathcal{X}|}$ of $\mathbf{L}(\mathcal{X})$ form a basis of $\mathbb{R}^{|\mathcal{X}|}$. For every $k \in [K]$, $(\mathbf{y}_k - \mathbf{F}_\mathcal{X}\mathbf{z}_k) \in \mathbb{R}^{|\mathcal{X}|}$ can be decomposed as the linear combination of $\{\mathbf{v}_i\}_{i \in [|\mathcal{X}|]}$ such that

$$\|\mathbf{y}_k - \mathbf{F}_\mathcal{X}\mathbf{z}_k\|_2^2 = \sum_{i=1}^{|\mathcal{X}|} \left(\mathbf{v}_i^\top (\mathbf{y}_k - \mathbf{F}_\mathcal{X}\mathbf{z}_k)\right)^2$$

Meanwhile, Claim C.3 implies that, for $f \in \mathcal{F}_{\mathbf{L}(\mathcal{X})}$, with $\mathbf{V}_K = [\mathbf{v}_1, \ldots, \mathbf{v}_K] \in \mathbb{R}^{|\mathcal{X}| \times K}$, we have $\text{Range}(\mathbf{F}_\mathcal{X}) = \text{Range}(\mathbf{V}_K)$. That is, $\mathbf{v}_i^\top \mathbf{F}_\mathcal{X}\mathbf{z}_k = 0$ for all $i > K$, and by taking

$$\mathbf{z}_k = \left(\mathbf{V}_K^\top \mathbf{F}_\mathcal{X}\right)^{-1} \mathbf{V}_K^\top \mathbf{y}_k \quad \Rightarrow \quad \sum_{i=1}^{K} \left(\mathbf{v}_i^\top (\mathbf{y}_k - \mathbf{F}_\mathcal{X}\mathbf{z}_k)\right)^2 = 0.$$

Overall, for every $k \in [K]$, there exists $\mathbf{z}_k \in \mathbb{R}^K$ such that

$$\|\mathbf{y}_k - \mathbf{F}_\mathcal{X}\mathbf{z}_k\|_2^2 = \sum_{i=K+1}^{|\mathcal{X}|} \left(\mathbf{v}_i^\top \mathbf{y}_k\right)^2 \leq \frac{1}{\lambda_{K+1}} \sum_{i=K+1}^{|\mathcal{X}|} \lambda_i \cdot \left(\mathbf{v}_i^\top \mathbf{y}_k\right)^2 \leq \frac{\mathbf{y}_k^\top \mathbf{L}(\mathcal{X}) \mathbf{y}_k}{\lambda_{K+1}},$$

and Lemma C.2 implies that

$$\min_{\{\mathbf{z}_k \in \mathbb{R}^K\}_{k \in [K]}} \sum_{k \in [K]} \|\mathbf{y}_k - \mathbf{F}_{\mathcal{X}} \mathbf{z}_k\|_2^2 \leq \sum_{k \in [K]} \frac{\mathbf{y}_k^\top \mathbf{L}(\mathcal{X}) \mathbf{y}_k}{\lambda_{K+1}} = \frac{\alpha}{\lambda_{K+1}},$$

which completes the proof. □

## C.2 Limitation of Spectral Clustering Alone

To ground the discussion in Remark 4.1, here, we raise a toy counter-example where spectral clustering alone fails to satisfy Assumption 4.2 and suffers from a large clustering error. Intuitively, such a pitfall of spectral clustering is caused by a suboptimal choice of basis for the predictions $f(\mathcal{X})$ in Equation (1) due to the inherited rotation invariance of the minimizers of $R(f)$.

**Example C.1** (Pitfall of spectral clustering alone). *We consider a binary classification problem with two balanced ground truth classes $\{\mathcal{X}_1, \mathcal{X}_2\}$ perfectly partitioned in $G_{\mathcal{X}}$ as disconnected components. Then, when the prediction $y_f(\mathbf{x}) \triangleq \operatorname{argmax}_{k \in [K]} f(\mathbf{x})_k$ is made with a uniformly random break of ties, Equation (1) alone suffers from $\mu(\mathcal{F}_{\mathbf{L}(\mathcal{X})}) \geq \frac{1}{4}$ in expectation, due to the violation of Assumption 4.2.*

*Nevertheless, with as few as one labeled sample per class $\{(\mathbf{x}_i, i) \mid i = 1, 2\}$, weak supervision via the cross-entropy loss*

$$\min_{f \in \mathcal{F}_{\mathbf{L}(\mathcal{X})}} \left\{ \widehat{\mathcal{L}}_{\mathrm{CE}}(f) \triangleq -\sum_{i=1}^{2} \log\left(\mathtt{softmax}(f(\mathbf{x}_i))_i\right) \right\}$$

*facilitates satisfaction of Assumption 4.2 with a provably large margin $\gamma$. For instance, assuming $\|f(\mathbf{x})\|_2 \leq \|f(\mathbf{S})\|_2 \leq \beta$ and $w_{\mathbf{x}} = 1/|\mathcal{X}|$ for all $\mathbf{x} \in \mathcal{X}$ (simplified for illustration purposes), for any $f \in \mathcal{F}_{\mathbf{L}(\mathcal{X})}$ under weak supervision such that*

$$f \in \left\{ f \in \mathcal{F}_{\mathbf{L}(\mathcal{X})} \;\middle|\; \log\left(1 + e^{-\sqrt{2}\beta}\right) \leq \widehat{\mathcal{L}}_{\mathrm{CE}}(f) < \log\left(1 + e^{-\beta}\right) \right\},$$

*Assumption 4.2 is satisfied with*

$$\gamma \geq -\log\left(e^{\widehat{\mathcal{L}}_{\mathrm{CE}}(f)} - 1\right) - \sqrt{2\beta^2 - \log\left(e^{\widehat{\mathcal{L}}_{\mathrm{CE}}(f)} - 1\right)^2}.$$

*In particular, with the minimum feasible cross-entropy loss $\widehat{\mathcal{L}}_{\mathrm{CE}}(f) = \log\left(1 + e^{-\sqrt{2}\beta}\right)$, we have $\gamma \geq \sqrt{2}\beta$.*

*We remark that with DAC regularizations like FixMatch [Sohn et al., 2020] and MixMatch [Berthelot et al., 2019], model predictions are matched against pseudo-labels that are gauged based on data augmentations, usually via the cross-entropy loss. Therefore, DAC has analogous effects on the margins of the learned prediction functions $f$ as the additional supervision described above.*

*Moreover, when coupling Equation (1) with the conventional KD (i.e., feature matching), the learned prediction functions $f$ inherit the boundedness and margin of the teacher model, which are presumably satisfactory.*

*Rationale for Example C.1.* Recall that $\lambda_1 \leq \cdots \leq \lambda_{|\mathcal{X}|}$ are the eigenvalues of the graph Laplacian $\mathbf{L}(\mathcal{X}) = \mathbf{I} - \overline{\mathbf{W}}(\mathcal{X})$, and let $\{\mathbf{v}_1, \cdots, \mathbf{v}_{|\mathcal{X}|}\}$ be the associated normalized eigenvectors. For a binary classification with $\{\mathcal{X}_1, \mathcal{X}_2\}$ perfectly separated in $G_{\mathcal{X}}$, we can write $\lambda_1 = \lambda_2 = 0$, while $\mathbf{v}_1, \mathbf{v}_2$ are the scaled identity vectors of $\mathcal{X}_1, \mathcal{X}_2$, respectively, such that $\mathbf{v}_i(\mathbf{x}) = \frac{1}{\sqrt{|\mathcal{X}_i|}} \mathbb{1}\{\mathbf{x} \in \mathcal{X}_i\}$ for $i = 1, 2$.

By denoting $\mathbf{V}_2 = [\mathbf{v}_1, \mathbf{v}_2] \in \mathbb{R}^{|\mathcal{X}| \times 2}$, the Eckart-Young-Mirsky theorem [Eckart and Young, 1936] implies that, for any orthogonal matrix $\mathbf{Q} = [\mathbf{q}_1; \mathbf{q}_2] \in \mathbb{R}^{2 \times 2}$,

$$f(\mathcal{X}) = \mathbf{D}(\mathcal{X})^{-\frac{1}{2}} \mathbf{V}_2 \operatorname{diag}(1 - \lambda_1, 1 - \lambda_2)^{\frac{1}{2}} \mathbf{Q} = \mathbf{D}(\mathcal{X})^{-\frac{1}{2}} \mathbf{V}_2 \mathbf{Q}$$

is a minimizer of $R(f)$ in Equation (1). That is, $f(\mathbf{x}) = \frac{1}{\sqrt{w_{\mathbf{x}} |\mathcal{X}_i|}} \mathbf{q}_i$ for $\mathbf{x} \in \mathcal{X}_i$, $i = 1, 2$.

**Spectral clustering alone.** However, by taking $\mathbf{q}_1 = \frac{1}{\sqrt{2}}[1;1]$ and $\mathbf{q}_2 = \frac{1}{\sqrt{2}}[-1;1]$, with a uniformly random break of ties, the corresponding $f \in \mathcal{F}_{\mathbf{L}(\mathcal{X})}$ will misclassify half of $\mathbf{x} \in \mathcal{X}_1$ as $\mathcal{X}_2$ in expectation. Then, since $P(\mathcal{X}_1) = P(\mathcal{X}_2) = \frac{1}{2}$, by Definition 3.1, we have $M(f) = \{\mathbf{x} \in \mathcal{X}_1 \mid y_f(\mathbf{x}) = 2\}$ such that $\mu\left(\mathcal{F}_{\mathbf{L}(\mathcal{X})}\right) \geq P(M(f)) = \frac{1}{4}$ in expectation.

Spectral clustering in Example C.1 fails to yield low clustering error due to the potential violation of Assumption 4.2. For example, consider any prediction function $f \in \mathcal{F}_{\mathbf{L}(\mathcal{X})}$ described in Example C.1.

(i) When $\operatorname{argmin}_{\mathbf{x} \in \mathcal{X} \setminus M(f)} w_{\mathbf{x}} \subseteq \mathcal{X}_1$, recalling $\mathbf{s}_k = \operatorname{argmax}_{\mathbf{x} \in \mathcal{X} \setminus M(f)} f(\mathbf{x})_k$ and $f(\mathbf{x}) = \frac{1}{\sqrt{w_{\mathbf{x}} |\mathcal{X}_i|}} \mathbf{q}_i$ for $\mathbf{x} \in \mathcal{X}_i$ such that, since $|\mathcal{X}_1| = |\mathcal{X}_2|$,

$$\mathbf{s}_1 \in \mathcal{X}_1, \qquad \mathbf{s}_2 \in \operatorname*{argmax}_{\mathbf{x} \in \mathcal{X} \setminus M(f)} f(\mathbf{x})_2 = \operatorname*{argmax}_{\mathbf{x} \in \mathcal{X} \setminus M(f)} \frac{1}{\sqrt{w_{\mathbf{x}} |\mathcal{X}_i|}} = \operatorname*{argmin}_{\mathbf{x} \in \mathcal{X} \setminus M(f)} w_{\mathbf{x}} \subseteq \mathcal{X}_1,$$

there does not exist a skeleton subset $\mathbf{S} = [\mathbf{s}_1; \mathbf{s}_2]$ as described in Assumption 4.2.

(ii) Otherwise, suppose one can find a skeleton subset $\mathbf{S} = [\mathbf{s}_1; \mathbf{s}_2]$ as described in Assumption 4.2. Without loss of generality, by taking $(\operatorname{argmin}_{\mathbf{x} \in \mathcal{X}} w_{\mathbf{x}}) \cap M(f) \neq \emptyset$ in the problem setup, there exists $\mathbf{x} \in M(f) \subset \mathcal{X}_1$ (with $y_f(\mathbf{x}) = 2$) such that $w_{\mathbf{x}} \leq w_{\mathbf{s}_1}$. Therefore,

$$\gamma_1 \triangleq f(\mathbf{s}_1)_1 - \max_{\mathbf{x} \in M(f): y_f(\mathbf{x}) \neq 1} f(\mathbf{x})_1 \leq \frac{1}{\sqrt{2 w_{\mathbf{s}_1} |\mathcal{X}_1|}} - \frac{1}{\sqrt{2 w_{\mathbf{x}} |\mathcal{X}_1|}} \leq 0$$

implies a trivial margin $\gamma \leq 0$, which violates $\gamma > 0$ in Assumption 4.2.

**With additional weak supervision.** We first observe that for both $i \in [2]$,

$$-\log\left(\mathtt{softmax}\left(f(\mathbf{x}_i)\right)_i\right) \leq e^{\widehat{\mathcal{L}}_{\mathrm{CE}}(f)},$$

which implies that

$$f(\mathbf{x}_1)_1 - f(\mathbf{x}_1)_2 \geq -\log\left(e^{\widehat{\mathcal{L}}_{\mathrm{CE}}(f)} - 1\right), \quad f(\mathbf{x}_2)_2 - f(\mathbf{x}_2)_1 \geq -\log\left(e^{\widehat{\mathcal{L}}_{\mathrm{CE}}(f)} - 1\right)$$

where $-\log\left(e^{\widehat{\mathcal{L}}_{\mathrm{CE}}(f)} - 1\right) > 0$ since $\widehat{\mathcal{L}}_{\mathrm{CE}}(f) < \log\left(1 + e^{-\beta}\right) < \log(2)$.

Since we assume $\|f(\mathbf{x})\|_2 \leq \beta$ for all $\mathbf{x} \in \mathcal{X}$, $f(\mathbf{x}_i)_1^2 + f(\mathbf{x}_i)_2^2 \leq \beta^2$ for both $i = 1, 2$, we therefore have

$$f(\mathbf{x}_1)_1 - f(\mathbf{x}_2)_1 \geq -\log\left(e^{\widehat{\mathcal{L}}_{\mathrm{CE}}(f)} - 1\right) - \sqrt{2\beta^2 - \log\left(e^{\widehat{\mathcal{L}}_{\mathrm{CE}}(f)} - 1\right)^2},$$

This similarly holds for $f(\mathbf{x}_2)_2 - f(\mathbf{x}_1)_2$.

Recall now that $f(\mathbf{x}) = \frac{1}{\sqrt{w_{\mathbf{x}} |\mathcal{X}_i|}} \mathbf{q}_i$ for $\mathbf{x} \in \mathcal{X}_i$, $i = 1, 2$ and $\mathbf{x}_i \in \mathcal{X}_i$ for $i = 1, 2$. Assuming $w_{\mathbf{x}} = 1/|\mathcal{X}|$ for all $\mathbf{x} \in \mathcal{X}$, we have

$$f(\mathbf{x}) = \begin{cases} \sqrt{\frac{w_{\mathbf{x}_1}}{w_{\mathbf{x}}}} f(\mathbf{x}_1) = f(\mathbf{x}_1), & \mathbf{x} \in \mathcal{X}_1 \\ \sqrt{\frac{w_{\mathbf{x}_2}}{w_{\mathbf{x}}}} f(\mathbf{x}_2) = f(\mathbf{x}_2), & \mathbf{x} \in \mathcal{X}_2 \end{cases}.$$

Therefore, $\gamma_1 \geq f(\mathbf{x}_1)_1 - f(\mathbf{x}_2)_1$ and $\gamma_2 \geq f(\mathbf{x}_2)_2 - f(\mathbf{x}_1)_2$, which together imply that

$$\gamma \geq -\log\left(e^{\widehat{\mathcal{L}}_{\mathrm{CE}}(f)} - 1\right) - \sqrt{2\beta^2 - \log\left(e^{\widehat{\mathcal{L}}_{\mathrm{CE}}(f)} - 1\right)^2}.$$

Now examine the two extremes of $\log\left(1 + e^{-\sqrt{2}\beta}\right) \leq \widehat{\mathcal{L}}_{\mathrm{CE}}(f) < \log\left(1 + e^{-\beta}\right)$. When the cross-entropy loss achieves its minimum feasible value $\widehat{\mathcal{L}}_{\mathrm{CE}}(f) = \log\left(1 + e^{-\sqrt{2}\beta}\right)$, we have that $\gamma \geq \sqrt{2}\beta$, whereas the margin can be nearly trivial $\gamma \geq \epsilon_{\mathrm{CE}}$ with $\epsilon_{\mathrm{CE}} \to 0$ when the cross-entropy loss is on the larger side $\widehat{\mathcal{L}}_{\mathrm{CE}}(f) \to \log\left(1 + e^{-\beta}\right)$. □

## C.3 Connection with Sparsest $k$-partition

We now identify the connection between the eigenvalue $\lambda_{K+1}$ and spectral graph theoretic notions of clusteredness of the underlying graph. We recall the notion of sparsest $k$-partitions [Louis and Makarychev, 2014] based on the Dirichlet conductance of subgraphs.

**Definition C.1** (Sparsest $k$-partition, HaoChen et al. [2021] Definition 3.4, Louis and Makarychev [2014] Problem 1.1). *Given a weighted undirected graph $G = (\mathcal{X}, w)$, for any $k \in [|\mathcal{X}|]$, let $\mathcal{S} = \left\{ \{S_i\}_{i \in [k]} \mid \cup_{i \in [k]} S_i = \mathcal{X}, \ S_i \cap S_j = \emptyset, \ S_i \neq \emptyset \ \forall i \neq j, \ i, j \in [k] \right\}$ be the set of all $k$-partitions of $G$. The sparsest $k$-partition of $G$ is defined as*

$$\phi_k \triangleq \min_{\{S_i\}_{i \in [k]} \in \mathcal{S}} \max_{i \in [k]} \phi_G(S_i),$$

*where $\phi_G(S) \triangleq \left( \sum_{\mathbf{x} \in S} \sum_{\mathbf{x}' \notin S} w_{\mathbf{x}\mathbf{x}'} \right) / \left( \sum_{\mathbf{x} \in S} w_{\mathbf{x}} \right)$ is the Dirichlet conductance of $S \subseteq \mathcal{X}$.*

It is worth mentioning that $\phi_k$ is a non-decreasing function in $k$ [HaoChen et al., 2021]. In particular, $\phi_k = 0$ when the graph has at least $k$ disconnected components; whereas $\lambda_{K+1} > 0$ (Assumption 4.1) implies that $\phi_{K+1} > 0$.

Meanwhile, Louis and Makarychev [2014] unveils the following connection between the sparsest $k$-partition of the graph and the $k'$th smallest eigenvalue of its graph Laplacian:

**Lemma C.4** (Louis and Makarychev [2014] Proposition 1.2). *Given any weighted undirected graph $G = (\mathcal{X}, w)$, let $0 = \lambda_1 \leq \cdots \leq \lambda_{|\mathcal{X}|}$ be the eigenvalues of the normalized graph Laplacian in the ascending order. For any $k, k' \in [|\mathcal{X}|]$, $k < k'$, there exists a partition $\{S_i\}_{i \in [k]} \in \mathcal{S}$ of $\mathcal{X}$ such that, for all $i \in [k]$,*

$$\phi_G(S_i) \lesssim \text{poly}\left( \frac{k}{k' - k} \right) \sqrt{\log(k) \lambda_{k'}}.$$

Leveraging the existing result Lemma C.4 [Louis and Makarychev, 2014] from spectral graph theory, we have the following corollary.

**Corollary C.5** (Clustering in terms of sparsest $k$-partitions). *Under Assumption 4.1 and Assumption 4.2 for every $f \in \mathcal{F}_{\mathbf{L}(\mathcal{X})}$, error of clustering with the population (Equation (1)) satisfies that, if $\alpha > 0$ and $\phi_k > 0$ for all $k \in [K]$, then*

$$\mu\left( \mathcal{F}_{\mathbf{L}(\mathcal{X})} \right) \lesssim \max\left( \frac{\beta^2}{\gamma^2}, 1 \right) \cdot \text{poly}\left( \frac{k}{K + 1 - k} \right) \log(k) \frac{\alpha}{\phi_k^2}.$$

We also notice that when $\alpha = 0$, Theorem 4.1 automatically implies that $\mu\left( \mathcal{F}_{\mathbf{L}(\mathcal{X})} \right) = 0$. Intuitively, Corollary C.5 implies that except for the partition of the $K$ ground truth classes $\{\mathcal{X}\}_{k \in [K]}$ (i.e., when $\alpha > 0$), $G_{\mathcal{X}}$ cannot be partitioned into other $K$ components by removing a sparse set of edges from $G_{\mathcal{X}}$.

*Proof of Corollary C.5.* Lemma C.4 implies that for any $k \in [K]$, there exists a partition $\{S_i\}_{i \in [k]} \in \mathcal{S}$ of $\mathcal{X}$ such that

$$\phi_k \leq \max_{i \in [k]} \phi_G(S_i) \lesssim \text{poly}\left( \frac{k}{K + 1 - k} \right) \sqrt{\log(k) \lambda_{K+1}},$$

and therefore

$$\frac{1}{\lambda_{K+1}} \lesssim \text{poly}\left( \frac{k}{K + 1 - k} \right) \log(k) \frac{1}{\phi_k^2},$$

which completes the proof when recalling Theorem 4.1. $\qquad\square$

## D Proofs and Discussions for Section 4.2

Here, we show that $\widehat{R}_{\mathbf{X}^u}(f)$ shares the same minimizer as the population Laplacian regularizer $R(f)$ in expectation (Proposition D.1) and provide generalization bounds for $\widehat{R}_{\mathbf{X}^u}(f)$ (Theorem 4.2) accordingly.

## D.1 Unbiased Estimations of Population Laplacian

**Proposition D.1.** $\widehat{R}_{\mathbf{X}^u}(f)$ *(Equation* (3)*) serves as an unbiased estimate for* $R(f)$ *(Equation* (1)*):*

$$R\left(f\right) = \mathbb{E}_{\mathbf{X}^u \sim P(\mathbf{x})^N}\left[\widehat{R}_{\mathbf{X}^u}\left(f\right)\right].$$

*Proof of Proposition D.1.* We first observe that $R(f)$ can be expanded as following:

$$
\begin{aligned}
R\left(f\right) &\triangleq \left\|\overline{\mathbf{W}}(\mathcal{X}) - \mathbf{D}(\mathcal{X})^{\frac{1}{2}} f\left(\mathcal{X}\right) f\left(\mathcal{X}\right)^{\top} \mathbf{D}(\mathcal{X})^{\frac{1}{2}}\right\|_F^2 \\
&= \left\|\mathbf{D}(\mathcal{X})^{-\frac{1}{2}} \mathbf{W}(\mathcal{X}) \mathbf{D}(\mathcal{X})^{-\frac{1}{2}}\right\|_F^2 + \left\|\mathbf{D}(\mathcal{X})^{\frac{1}{2}} f\left(\mathcal{X}\right) f\left(\mathcal{X}\right)^{\top} \mathbf{D}(\mathcal{X})^{\frac{1}{2}}\right\|_F^2 \\
&\quad - 2 \operatorname{tr}\left(\mathbf{W}(\mathcal{X}) f\left(\mathcal{X}\right) f\left(\mathcal{X}\right)^{\top}\right) \\
&= \sum_{\mathbf{x} \in \mathcal{X}} \sum_{\mathbf{x}' \in \mathcal{X}} \frac{w_{\mathbf{x}\mathbf{x}'}^2}{w_{\mathbf{x}} w_{\mathbf{x}'}} - 2 w_{\mathbf{x}\mathbf{x}'} f(\mathbf{x})^{\top} f(\mathbf{x}') + w_{\mathbf{x}} w_{\mathbf{x}'} \left(f(\mathbf{x})^{\top} f(\mathbf{x}')\right)^2 \\
&= \sum_{\mathbf{x} \in \mathcal{X}} \sum_{\mathbf{x}' \in \mathcal{X}} w_{\mathbf{x}} w_{\mathbf{x}'} \cdot \left(\left(\frac{w_{\mathbf{x}\mathbf{x}'}}{w_{\mathbf{x}} w_{\mathbf{x}'}}\right)^2 - \frac{2 w_{\mathbf{x}\mathbf{x}'}}{w_{\mathbf{x}} w_{\mathbf{x}'}} f(\mathbf{x})^{\top} f(\mathbf{x}') + \left(f(\mathbf{x})^{\top} f(\mathbf{x}')\right)^2\right).
\end{aligned}
$$

Then, with $k_\psi\left(\mathbf{x}, \mathbf{x}'\right) = \frac{w_{\mathbf{x}\mathbf{x}'}}{w_{\mathbf{x}} w_{\mathbf{x}'}}$, $\widehat{R}_{\mathbf{X}^u}\left(f\right)$ in Equation (3), $w_{\mathbf{x}} = P\left(\mathbf{x}\right)$, and $w_{\mathbf{x}'} = P\left(\mathbf{x}'\right)$, we have

$$
\begin{aligned}
R\left(f\right) &= \mathbb{E}_{\mathbf{x}, \mathbf{x}' \sim P(\mathbf{x})^2}\left[k_\psi\left(\mathbf{x}, \mathbf{x}'\right)^2 - 2 k_\psi\left(\mathbf{x}, \mathbf{x}'\right) f(\mathbf{x})^{\top} f(\mathbf{x}') + \left(f(\mathbf{x})^{\top} f(\mathbf{x}')\right)^2\right] \\
&= \mathbb{E}_{\mathbf{x}, \mathbf{x}' \sim P(\mathbf{x})^2}\left[\left(f(\mathbf{x})^{\top} f(\mathbf{x}') - k_\psi\left(\mathbf{x}, \mathbf{x}'\right)\right)^2\right] \\
&= \mathbb{E}_{\mathbf{X}^u \sim P(\mathbf{x})^N}\left[\frac{2}{N} \sum_{i=1}^{N/2} \left(f\left(\mathbf{x}_{2i-1}^u\right)^{\top} f\left(\mathbf{x}_{2i}^u\right) - k_\psi\left(\mathbf{x}_{2i-1}^u, \mathbf{x}_{2i}^u\right)\right)^2\right] \\
&= \mathbb{E}_{\mathbf{X}^u \sim P(\mathbf{x})^N}\left[\widehat{R}_{\mathbf{X}^u}\left(f\right)\right].
\end{aligned}
$$

$\square$

## D.2 Proof of Theorem 4.2

*Proof of Theorem 4.2.* By defining

$$\widehat{r}_{\mathbf{x}, \mathbf{x}'}(f) \triangleq \left(f(\mathbf{x})^{\top} f(\mathbf{x}') - k_\psi\left(\mathbf{x}, \mathbf{x}'\right)\right)^2,$$

we have $R(f) = \mathbb{E}_{\mathbf{x}, \mathbf{x}' \sim P(\mathbf{x})^2}\left[\widehat{r}_{\mathbf{x}, \mathbf{x}'}(f)\right]$ and $\widehat{R}_{\mathbf{X}^u}\left(f\right) = \frac{2}{N} \sum_{i=1}^{N/2} \widehat{r}_{\mathbf{x}_{2i-1}^u, \mathbf{x}_{2i}^u}(f)$ as

$$\widehat{R}_{\mathbf{X}^u}\left(f\right) = \frac{2}{N} \sum_{i=1}^{N/2} \left(f\left(\mathbf{x}_{2i-1}^u\right)^{\top} f\left(\mathbf{x}_{2i}^u\right) - k_\psi\left(\mathbf{x}_{2i-1}^u, \mathbf{x}_{2i}^u\right)\right)^2,$$

where $\left\{\left(\mathbf{x}_{2i-1}^u, \mathbf{x}_{2i}^u\right)\right\}_{i=1}^{N/2}$ are *i.i.d.* given $\mathbf{X}^u \sim P(\mathbf{x})^N$. Consider the function class

$$\widehat{r}_{\cdot, \cdot} \circ \mathcal{F} = \left\{\widehat{r}_{\cdot, \cdot}(f) : \mathcal{X} \times \mathcal{X} \to \mathbb{R} | f \in \mathcal{F}\right\}.$$

Since $0 \le k_\psi(\mathbf{x}, \mathbf{x}') \le B_{k_\psi}$, and by Cauchy-Schwarz inequality, $f(\mathbf{x})^{\top} f(\mathbf{x}') \le \|f(\mathbf{x})\|_2 \|f(\mathbf{x}')\|_2 \le B_f$, we have $0 \le \widehat{r}_{\mathbf{x}, \mathbf{x}'}(f) \le \left(B_{k_\psi} + B_f\right)^2$ for all $\mathbf{x}, \mathbf{x}' \in \mathcal{X}$, which means that $\widehat{r}_{\cdot, \cdot} \circ \mathcal{F}$ is $\left(B_{k_\psi} + B_f\right)^2$-bounded. Leveraging Lemma F.1 gives that, with probability at least $1 - \delta/2$ over $\mathbf{X}^u$,

$$R\left(f_{|\mathbf{X}^u}\right) - R\left(f_{|\mathcal{X}}\right) \le 4 \mathfrak{R}_{N/2}\left(\widehat{r}_{\cdot, \cdot} \circ \mathcal{F}\right) + 2 \left(B_{k_\psi} + B_f\right)^2 \sqrt{\frac{\log\left(4/\delta\right)}{N}}.$$

Now, it remains to show that $\Re_{N/2}\left(\widehat{r}_{\cdot,\cdot}\circ\mathcal{F}\right)\leq 4\sqrt{2B_f}\left(B_f+B_{k_\psi}\right)\Re_{N/2}\left(\mathcal{F}\right)$. For this, we recall Equation (4) and observe that

$$\Re_{N/2}\left(\widehat{r}_{\cdot,\cdot}\circ\mathcal{F}\right)=\mathbb{E}_{\substack{\mathbf{X}^u\sim P(\mathbf{x})^N\\\boldsymbol{\rho}\sim\mathrm{Rad}^{\frac{N}{2}}}}\left[\sup_{f\in\mathcal{F}}\frac{2}{N}\sum_{i=1}^{N/2}\rho_i\cdot\left(f\left(\mathbf{x}_{2i-1}^u\right)^\top f\left(\mathbf{x}_{2i}^u\right)-k_\psi\left(\mathbf{x}_{2i-1}^u,\mathbf{x}_{2i}^u\right)\right)^2\right],$$

where with Talagrand's lemma [Ledoux and Talagrand, 2013, Theorem 4.12] for compositions with scalar Lipschitz functions,

$$\Re_{N/2}\left(\widehat{r}_{\cdot,\cdot}\circ\mathcal{F}\right)\leq 2\left(B_f+B_{k_\psi}\right)\mathbb{E}_{\mathbf{X}^u,\boldsymbol{\rho}}\left[\sup_{f\in\mathcal{F}}\frac{2}{N}\sum_{i=1}^{N/2}\rho_i\cdot f\left(\mathbf{x}_{2i-1}^u\right)^\top f\left(\mathbf{x}_{2i}^u\right)\right].$$

Since $f\left(\mathbf{x}_{2i-1}^u\right)^\top f\left(\mathbf{x}_{2i}^u\right)\leq\frac{1}{2}\left(\left\|f\left(\mathbf{x}_{2i-1}^u\right)\right\|_2^2+\left\|f\left(\mathbf{x}_{2i}^u\right)\right\|_2^2\right)$,

$$\mathbb{E}_{\mathbf{X}^u,\boldsymbol{\rho}}\left[\sup_{f\in\mathcal{F}}\frac{2}{N}\sum_{i=1}^{N/2}\rho_i\cdot f\left(\mathbf{x}_{2i-1}^u\right)^\top f\left(\mathbf{x}_{2i}^u\right)\right]$$

$$\leq\mathbb{E}_{\mathbf{X}^u,\boldsymbol{\rho}}\left[\sup_{f\in\mathcal{F}}\frac{2}{N}\sum_{i=1}^{N/2}\rho_i\cdot\frac{1}{2}\left(\left\|f\left(\mathbf{x}_{2i-1}^u\right)\right\|_2^2+\left\|f\left(\mathbf{x}_{2i}^u\right)\right\|_2^2\right)\right]$$

$$\leq\frac{1}{2}\left(\mathbb{E}_{\mathbf{X}^u,\boldsymbol{\rho}}\left[\sup_{f\in\mathcal{F}}\frac{2}{N}\sum_{i=1}^{N/2}\rho_i\cdot\left\|f\left(\mathbf{x}_{2i-1}^u\right)\right\|_2^2\right]+\mathbb{E}_{\mathbf{X}^u,\boldsymbol{\rho}}\left[\sup_{f\in\mathcal{F}}\frac{2}{N}\sum_{i=1}^{N/2}\rho_i\cdot\left\|f\left(\mathbf{x}_{2i}^u\right)\right\|_2^2\right]\right)$$

$$=\mathbb{E}_{\mathbf{X}^u,\boldsymbol{\rho}}\left[\sup_{f\in\mathcal{F}}\frac{2}{N}\sum_{i=1}^{N/2}\rho_i\cdot\left\|f\left(\mathbf{x}_{2i}^u\right)\right\|_2^2\right].$$

By the vector-contraction inequality for Rademacher complexities [Maurer, 2016, Corollary 4], since $\left\|f\left(\mathbf{x}\right)\right\|_2^2$ is $\left(2\sqrt{B_f}\right)$-Lipschitz in $f(\mathbf{x})$, with $f(\mathbf{x})_k$ denoting the $k$-th entry of $f(\mathbf{x})\in\mathbb{R}^K$,

$$\mathbb{E}_{\mathbf{X}^u,\boldsymbol{\rho}}\left[\sup_{f\in\mathcal{F}}\frac{2}{N}\sum_{i=1}^{N/2}\rho_i\cdot\left\|f\left(\mathbf{x}_{2i}^u\right)\right\|_2^2\right]\leq 2\sqrt{2B_f}\cdot\mathbb{E}_{\substack{\mathbf{X}^u\sim P(\mathbf{x})^{\frac{N}{2}}\\\boldsymbol{\rho}\sim\mathrm{Rad}^{\frac{N}{2}\times K}}}\left[\sup_{f\in\mathcal{F}}\frac{2}{N}\sum_{i=1}^{N/2}\sum_{k=1}^{K}\rho_{ik}\cdot f\left(\mathbf{x}_i^u\right)_k\right]$$

$$=2\sqrt{2B_f}\Re_{N/2}\left(\mathcal{F}\right).$$

Overall, we have $\Re_{N/2}\left(\widehat{r}_{\cdot,\cdot}\circ\mathcal{F}\right)\leq 4\sqrt{2B_f}\left(B_f+B_{k_\psi}\right)\Re_{N/2}\left(\mathcal{F}\right)$ which completes the proof. $\qquad\square$

### D.3 Unlabeled Sample Complexity with Deep Neural Networks

Here, we ground Theorem 4.2 by exemplifying $\Re_{N/2}(\mathcal{F})$ leveraging the existing generalization bound [Golowich et al., 2018] for deep neural networks.

**Claim D.2** (Rademacher complexity of deep neural networks [Golowich et al., 2018]). *Recall the notion of Rademacher complexity for a vector-valued function class $\mathcal{F}\ni f:\mathcal{X}\to\mathbb{R}^K$ from Equation (4). Adopting the existing results in Golowich et al. [2018] and following the notations in Section 6, we consider $\mathcal{F}$ as a class of deep neural networks:*

$$\mathcal{F}=\left\{f(\mathbf{x})=\mathbf{A}_p\phi\left(\cdots\mathbf{A}_2\phi(\mathbf{A}_1\mathbf{x})\cdots\right)\mid\left\|\mathbf{A}_\iota\right\|_F\leq B_{\mathcal{F},\iota}\,\forall\,\iota\in[p]\right\}$$

*with weight matrices $\{\mathbf{A}_\iota\mid\iota\in[p]\}$ and a 1-Lipschitz, positive-homogeneous activation function $\phi\left(\cdot\right)$ (e.g., ReLU) applied entry-wisely. Let $B_{\mathcal{F}}=\prod_{\iota=1}^{p}B_{\mathcal{F},\iota}$ and $\|\mathbf{x}\|_2\leq B_{\mathcal{X}}$ for all $\mathbf{x}\in\mathcal{X}$. Then, by Golowich et al. [2018] Theorem 1, we have*

$$\Re_{N/2}(\mathcal{F})\leq\frac{\left(2\sqrt{p\log(2)}+\sqrt{2}\right)KB_{\mathcal{X}}B_{\mathcal{F}}}{\sqrt{N}}.$$

### D.4  Proof of Theorem 4.3

*Proof of Theorem 4.3.* Let $\mathbf{U}_{\mathcal{X}} \triangleq \mathbf{D}\left(\mathcal{X}\right)^{1/2} f_{|\mathbf{X}^u}\left(\mathcal{X}\right), \mathbf{F}_{\mathcal{X}} \triangleq \mathbf{D}\left(\mathcal{X}\right)^{1/2} f_{|\mathcal{X}}\left(\mathcal{X}\right)$, and $\mathbf{y}_k \in [0,1]^{|\mathcal{X}|}$ be the $k$-th column of $\mathbf{Y}_{\mathcal{X}} \triangleq \mathbf{D}\left(\mathcal{X}\right)^{1/2} \overrightarrow{y_*}(\mathcal{X})$. We denote $\mathbf{P}_{\mathbf{U}} \triangleq \mathbf{U}_{\mathcal{X}} \mathbf{U}_{\mathcal{X}}^{\dagger}$ as the orthogonal projection onto $\mathrm{Range}\left(\mathbf{U}_{\mathcal{X}}\right)$ and $\mathbf{P}_{\mathbf{U}}^{\perp} \triangleq \mathbf{I} - \mathbf{P}_{\mathbf{U}}$ as the orthogonal complement of $\mathbf{P}_{\mathbf{U}}$.

Recalling from the proof of Theorem 4.1, Lemma C.1 implies that for all $f_{|\mathbf{X}^u} \in \mathcal{F}_{\mathbf{L}(\mathbf{X}^u)}$,

$$P\left(M(f_{|\mathbf{X}^u})\right) \leq 2\max\left(\frac{\beta^2}{\gamma^2}, 1\right) \sum_{k \in [K]} \min_{\mathbf{z}_k \in \mathbb{R}^K} \|\mathbf{y}_k - \mathbf{U}_{\mathcal{X}}\mathbf{z}_k\|_2^2 = 2\max\left(\frac{\beta^2}{\gamma^2}, 1\right) \sum_{k \in [K]} \left\|\mathbf{P}_{\mathbf{U}}^{\perp}\mathbf{y}_k\right\|_2^2,$$

where the second equality comes from taking $\mathbf{z}_k = \mathbf{U}_{\mathcal{X}}^{\dagger}\mathbf{y}_k$ for every $k \in [K]$.

Consider the spectral decomposition of $\mathbf{L}\left(\mathcal{X}\right)$ in Equation (13). Since the orthonormal eigenvectors $\left\{\mathbf{v}_i \in \mathbb{R}^{|\mathcal{X}|}\right\}_{i \in [|\mathcal{X}|]}$ associated with eigenvalues $0 = \lambda_1 \leq \cdots \leq \lambda_{|\mathcal{X}|} \leq 1$ form a basis of $\mathbb{R}^{|\mathcal{X}|}$, by denoting $\mathbf{V}_K \triangleq [\mathbf{v}_1, \ldots, \mathbf{v}_K] \in \mathbb{R}^{|\mathcal{X}| \times K}$ and $\mathbf{V}_K^{\perp} \triangleq [\mathbf{v}_{K+1}, \ldots, \mathbf{v}_{|\mathcal{X}|}] \in \mathbb{R}^{|\mathcal{X}| \times K}$, for every $k \in [K]$,

$$\left\|\mathbf{P}_{\mathbf{U}}^{\perp}\mathbf{y}_k\right\|_2^2 = \sum_{i=1}^{|\mathcal{X}|} \left(\mathbf{v}_i^{\top}\mathbf{P}_{\mathbf{U}}^{\perp}\mathbf{y}_k\right)^2 = \left\|\mathbf{V}_K^{\top}\mathbf{P}_{\mathbf{U}}^{\perp}\mathbf{y}_k\right\|_2^2 + \left\|\left(\mathbf{V}_K^{\perp}\right)^{\top}\mathbf{P}_{\mathbf{U}}^{\perp}\mathbf{y}_k\right\|_2^2.$$

It is shown in the proof of Theorem 4.1 that

$$\sum_{k \in [K]} \left\|\left(\mathbf{V}_K^{\perp}\right)^{\top}\mathbf{P}_{\mathbf{U}}^{\perp}\mathbf{y}_k\right\|_2^2 \leq \sum_{k \in [K]} \left\|\left(\mathbf{V}_K^{\perp}\right)^{\top}\mathbf{y}_k\right\|_2^2 \leq \frac{\alpha}{\lambda_{K+1}},$$

whereas by observing that $\|\mathbf{y}_k\|_2^2 = \sum_{\mathbf{x} \in \mathcal{X}} w_{\mathbf{x}} \cdot \mathbb{1}\left\{y_*(\mathbf{x}) = k\right\} = P\left(\mathcal{X}_k\right)$,

$$\sum_{k \in [K]} \left\|\mathbf{V}_K^{\top}\mathbf{P}_{\mathbf{U}}^{\perp}\mathbf{y}_k\right\|_2^2 \leq \sum_{k \in [K]} \|\mathbf{y}_k\|_2^2 \left\|\mathbf{P}_{\mathbf{U}}^{\perp}\mathbf{V}_K\right\|_F^2 = \left\|\mathbf{P}_{\mathbf{U}}^{\perp}\mathbf{V}_K\right\|_F^2 \sum_{k \in [K]} P\left(\mathcal{X}_k\right) = \left\|\mathbf{P}_{\mathbf{U}}^{\perp}\mathbf{V}_K\right\|_F^2.$$

Overall, we have

$$P\left(M(f_{|\mathbf{X}^u})\right) \leq 2\max\left(\frac{\beta^2}{\gamma^2}, 1\right)\left(\frac{\alpha}{\lambda_{K+1}} + \left\|\mathbf{P}_{\mathbf{U}}^{\perp}\mathbf{V}_K\right\|_F^2\right). \tag{16}$$

**Upper bounding $\left\|\mathbf{P}_{\mathbf{U}}^{\perp}\mathbf{V}_K\right\|_F^2$.** We first recall from Equation (13) that $\overline{\mathbf{W}}(\mathcal{X}) = \sum_{i=1}^{|\mathcal{X}|} (1-\lambda_i) \mathbf{v}_i\mathbf{v}_i^{\top}$. For $f_{|\mathcal{X}} \in \mathcal{F}_{\mathbf{L}(\mathcal{X})}$, Eckart-Young-Mirsky theorem [Eckart and Young, 1936] implies that $\mathbf{F}_{\mathcal{X}}\mathbf{F}_{\mathcal{X}}^{\top} = \sum_{i=1}^{K} (1-\lambda_i) \mathbf{v}_i\mathbf{v}_i^{\top}$, and

$$R\left(f_{|\mathcal{X}}\right) = \left\|\overline{\mathbf{W}}(\mathcal{X}) - \mathbf{F}_{\mathcal{X}}\mathbf{F}_{\mathcal{X}}^{\top}\right\|_F^2 = \left\|\sum_{i>K} (1-\lambda_i) \mathbf{v}_i\mathbf{v}_i^{\top}\right\|_F^2 = \sum_{i>K} (1-\lambda_i)^2.$$

Therefore, $R\left(f_{|\mathbf{X}^u}\right) \leq R\left(f_{|\mathcal{X}}\right) + \Delta$ implies that

$$\sum_{i>K} (1-\lambda_i)^2 + \Delta \geq \left\|\overline{\mathbf{W}}(\mathcal{X}) - \mathbf{U}_{\mathcal{X}}\mathbf{U}_{\mathcal{X}}^{\top}\right\|_F^2 = \left\|\left(\overline{\mathbf{W}}(\mathcal{X}) - \mathbf{P}_{\mathbf{U}}\overline{\mathbf{W}}(\mathcal{X})\right) + \left(\mathbf{P}_{\mathbf{U}}\overline{\mathbf{W}}(\mathcal{X}) - \mathbf{U}_{\mathcal{X}}\mathbf{U}_{\mathcal{X}}^{\top}\right)\right\|_F^2$$

$$= \left\|\mathbf{P}_{\mathbf{U}}^{\perp}\overline{\mathbf{W}}(\mathcal{X}) + \mathbf{P}_{\mathbf{U}}\left(\overline{\mathbf{W}}(\mathcal{X}) - \mathbf{U}_{\mathcal{X}}\mathbf{U}_{\mathcal{X}}^{\top}\right)\right\|_F^2 \quad \text{(since $\mathbf{P}_{\mathbf{U}}$ and $\mathbf{P}_{\mathbf{U}}^{\perp}$ are orthogonal)}$$

$$= \left\|\mathbf{P}_{\mathbf{U}}^{\perp}\overline{\mathbf{W}}(\mathcal{X})\right\|_F^2 + \left\|\mathbf{P}_{\mathbf{U}}\left(\overline{\mathbf{W}}(\mathcal{X}) - \mathbf{U}_{\mathcal{X}}\mathbf{U}_{\mathcal{X}}^{\top}\right)\right\|_F^2$$

$$\geq \left\|\mathbf{P}_{\mathbf{U}}^{\perp}\overline{\mathbf{W}}(\mathcal{X})\right\|_F^2 = \left\|\sum_{i=1}^{|\mathcal{X}|} (1-\lambda_i) \left(\mathbf{P}_{\mathbf{U}}^{\perp}\mathbf{v}_i\right) \mathbf{v}_i^{\top}\right\|_F^2$$

$$= \mathrm{tr}\left(\sum_{i=1}^{|\mathcal{X}|}\sum_{j=1}^{|\mathcal{X}|} (1-\lambda_i)(1-\lambda_j) \left(\mathbf{P}_{\mathbf{U}}^{\perp}\mathbf{v}_i\right) \mathbf{v}_i^{\top}\mathbf{v}_j \left(\mathbf{P}_{\mathbf{U}}^{\perp}\mathbf{v}_j\right)^{\top}\right)$$

$$= \sum_{i=1}^{|\mathcal{X}|} (1-\lambda_i)^2 \left\|\mathbf{P}_{\mathbf{U}}^{\perp}\mathbf{v}_i\right\|_2^2. \tag{17}$$

Meanwhile, the assumption $\Delta < (1 - \lambda_K)^2$ implies that $\mathrm{rank}\left(\mathbf{P}_{\mathbf{U}}^{\perp}\right) = |\mathcal{X}| - K$, and thus

$$\sum_{i=1}^{|\mathcal{X}|} \left\|\mathbf{P}_{\mathbf{U}}^{\perp} \mathbf{v}_i\right\|_2^2 = \left\|\mathbf{P}_{\mathbf{U}}^{\perp}\right\|_F^2 = \mathrm{tr}\left(\mathbf{P}_{\mathbf{U}}^{\perp}\right) = \mathrm{rank}\left(\mathbf{P}_{\mathbf{U}}^{\perp}\right) = |\mathcal{X}| - K. \tag{18}$$

Otherwise by contradiction, suppose $\mathrm{rank}\left(\mathbf{P}_{\mathbf{U}}\right) = \mathrm{rank}\left(\mathbf{U}_{\mathcal{X}}\right) < K$, then Eckart-Young-Mirsky theorem [Eckart and Young, 1936] implies that

$$\min_{\mathrm{rank}(\mathbf{U}_{\mathcal{X}}) < K} \left\|\overline{\mathbf{W}}(\mathcal{X}) - \mathbf{U}_{\mathcal{X}} \mathbf{U}_{\mathcal{X}}^{\top}\right\|_F^2 = \sum_{i=K}^{|\mathcal{X}|} (1 - \lambda_i)^2 > \sum_{i>K} (1 - \lambda_i)^2 + \Delta = R\left(f_{|\mathcal{X}}\right) + \Delta,$$

which contradicts $R\left(f_{|\mathbf{X}^u}\right) \leq R\left(f_{|\mathcal{X}}\right) + \Delta$.

Furthermore, since $\|\mathbf{v}_i\|_2 = 1$ and $\mathbf{P}_{\mathbf{U}}^{\perp}$ is an orthogonal projection, for all $i \in [|\mathcal{X}|]$,

$$0 \leq \left\|\mathbf{P}_{\mathbf{U}}^{\perp} \mathbf{v}_i\right\|_2^2 \leq 1. \tag{19}$$

For every $i \in [|\mathcal{X}|]$, we denote $\xi_i \triangleq \left\|\mathbf{P}_{\mathbf{U}}^{\perp} \mathbf{v}_i\right\|_2^2$. Then, upper bounding $\left\|\mathbf{P}_{\mathbf{U}}^{\perp} \mathbf{V}_K\right\|_F^2 = \sum_{i=1}^{K} \xi_i$ can be recast as a linear programming problem:

$$\begin{aligned}
(\mathrm{P}): \quad & \max_{\xi \in \mathbb{R}^{|\mathcal{X}|}} \sum_{i=1}^{K} \xi_i \\
& \text{s.t.} \quad 0 \leq \xi_i \leq 1, \\
& \qquad \sum_{i=1}^{|\mathcal{X}|} \xi_i = |\mathcal{X}| - K, \\
& \qquad \sum_{i=1}^{|\mathcal{X}|} (1 - \lambda_i)^2 \xi_i \leq \sum_{i>K} (1 - \lambda_i)^2 + \Delta,
\end{aligned} \tag{20}$$

whose dual can be expressed as

$$\begin{aligned}
(\mathrm{D}): \quad & \min_{(\omega_1, \cdots, \omega_{|\mathcal{X}|}, \omega_s, \omega_\Delta)} \sum_{i=1}^{|\mathcal{X}|} \omega_i - (|\mathcal{X}| - K)\omega_s + \left(\Delta + \sum_{i>K} (1 - \lambda_i)^2\right) \omega_\Delta \\
& \text{s.t.} \quad \omega_1, \cdots, \omega_{|\mathcal{X}|}, \omega_s, \omega_\Delta \geq 0, \\
& \qquad \omega_i - \omega_s + (1 - \lambda_i)^2 \omega_\Delta \geq 1 \quad \forall\, i \leq K, \\
& \qquad \omega_i - \omega_s + (1 - \lambda_i)^2 \omega_\Delta \geq 0 \quad \forall\, i > K.
\end{aligned} \tag{21}$$

By taking

$$\begin{aligned}
\omega_i &= 0 \quad \forall\, i \leq K_0, \\
\omega_i &= \frac{(1 - \lambda_{K_0})^2 - (1 - \lambda_i)^2}{(1 - \lambda_{K_0})^2 - (1 - \lambda_{K+1})^2} \quad \forall\, K_0 < i \leq K, \\
\omega_i &= \frac{(1 - \lambda_{K+1})^2 - (1 - \lambda_i)^2}{(1 - \lambda_{K_0})^2 - (1 - \lambda_{K+1})^2} \quad \forall\, K+1 \leq i \leq |\mathcal{X}|, \\
\omega_s &= \frac{(1 - \lambda_{K+1})^2}{(1 - \lambda_{K_0})^2 - (1 - \lambda_{K+1})^2}, \\
\omega_\Delta &= \frac{1}{(1 - \lambda_{K_0})^2 - (1 - \lambda_{K+1})^2},
\end{aligned}$$

the dual optimum can be upper bounded by

$$
(D) = \sum_{i=1}^{K} \omega_i + \Delta \cdot \omega_\Delta
$$

$$
= \frac{\Delta}{\left(1 - \lambda_{K_0}\right)^2 - \left(1 - \lambda_{K+1}\right)^2} + \sum_{i=K_0+1}^{K} \frac{\left(1 - \lambda_{K_0}\right)^2 - \left(1 - \lambda_i\right)^2}{\left(1 - \lambda_{K_0}\right)^2 - \left(1 - \lambda_{K+1}\right)^2}
$$

$$
\leq \frac{\left(1 + (K - K_0) C_{K_0}\right) \Delta}{\left(1 - \lambda_{K_0}\right)^2 - \left(1 - \lambda_{K+1}\right)^2},
$$

given $C_{K_0} \triangleq \frac{\left(1 - \lambda_{K_0}\right)^2 - \left(1 - \lambda_K\right)^2}{\Delta} = O(1)$.

With both the primal Equation (20) and the dual Equation (21) being feasible and bounded, the strong duality then implies that

$$
\left\| \mathbf{P}_{\mathbf{U}}^{\perp} \mathbf{V}_K \right\|_F^2 = \sum_{i=1}^{K} \xi_i = (P) = (D) \leq \frac{\left(1 + (K - K_0) C_{K_0}\right) \Delta}{\left(1 - \lambda_{K_0}\right)^2 - \left(1 - \lambda_{K+1}\right)^2},
$$

which, when plugging back into Equation (16), completes the proof. □

# E  Proofs and Discussions for Section 6

## E.1  Expansion-based Data Augmentation

For the detailed analysis, we further recall the following notion of constant expansion and its relation with the concept of $c$-expansion in Definition 6.1 from Wei et al. [2021].

**Definition E.1** (($q, \xi$)-constant expansion [Wei et al., 2021]). *We say the expansion-based data augmentation (Definition 6.1) satisfies ($q, \xi$)-constant expansion if for any $S \subseteq \mathcal{X}$ such that $P(S) \geq q$ and $P\left(S \cap \mathcal{X}_k\right) \leq P\left(\mathcal{X}_k\right)/2$ for all $k \in [K]$,*

$$
P\left(\mathrm{NB}(S)\right) > \min\{P(S), \xi\} + P(S).
$$

**Lemma E.1** ($c$-expansion implies constant expansion [Wei et al., 2021]). *The $c$-expansion property (Definition 6.1) implies $\left(\frac{\xi}{c-1}, \xi\right)$-constant expansion (Definition E.1) for any $\xi > 0$.*

*Proof of Lemma E.1.* Consider a subset $S \subseteq \mathcal{X}$ such that $P(S) \geq q$ and $P\left(S \cap \mathcal{X}_k\right) \leq P\left(\mathcal{X}_k\right)/2$. Let $S_k \triangleq S \cap \mathcal{X}_k$ for every $k \in [K]$.

(i) If $c \in (1, 2)$, since $P\left(\mathcal{X}_k\right) - P\left(S_k\right) \geq P\left(S_k\right) \geq (c-1)P\left(S_k\right)$

$$
P\left(\mathrm{NB}\left(S\right) \setminus S\right) \geq \sum_{k=1}^{K} P\left(\mathrm{NB}\left(S_k\right)\right) - P\left(S_k\right)
$$

$$
> \sum_{k=1}^{K} \min\left\{(c-1)P\left(S_k\right), P\left(\mathcal{X}_k\right) - P\left(S_k\right)\right\}
$$

$$
\geq \sum_{k=1}^{K} (c-1)P(S_k) = (c-1)P(S) \geq (c-1)q = \xi
$$

(ii) If $c \geq 2$, since $c - 1 \geq 1$ and $P\left(\mathcal{X}_k\right) - P\left(S_k\right) \geq P\left(S_k\right)$, we have

$$
P\left(\mathrm{NB}\left(S\right) \setminus S\right) > \sum_{k=1}^{K} \min\left\{(c-1)P\left(S_k\right), P\left(\mathcal{X}_k\right) - P\left(S_k\right)\right\} \geq \sum_{k=1}^{K} P\left(S_k\right) = P(S)
$$

Overall, we have

$$
P\left(\mathrm{NB}(S)\right) = P\left(\mathrm{NB}\left(S\right) \setminus S\right) + P(S) > \min\{P(S), \xi\} + P(S).
$$

□

## E.2 Data Augmentation Consistency Regularization

To bridge Equation (8) with common DAC regularization algorithms in practice, in Example E.1, we instantiate FixMatch [Sohn et al., 2020] – a state-of-the-art semi-supervised learning algorithm that leverages DAC regularization by encouraging similar predictions of proper weak and strong data augmentations $\mathbf{x}^w, \mathbf{x}^s \in \mathcal{A}(\mathbf{x})$ of the same sample $\mathbf{x}$.

**Example E.1** (FixMatch [Sohn et al., 2020]). *FixMatch minimizes the loss between the outputs of the strong augmentation $f(\mathbf{x}^s)$ and the pseudo-label of the weak augmentation $y_f(\mathbf{x}^w)$ when the confidence of such pseudo-labeling is high (i.e., $\max_{k \in [K]} f(\mathbf{x}^w)_k$ is larger than a threshold).*

*To connect FixMatch with Equation (8), we consider a fixed (unknown) margin-robust neighborhood $\mathcal{F}' \supseteq \mathcal{F}^\tau_{\mathbf{X}^u} \ni f_*$ where the strong augmentation characterizes the worst-case (minimum) robust margin $\mathbf{x}^s \in \bigcap_{f \in \mathcal{F}'} \operatorname{argmin}_{\mathbf{x}' \in \mathcal{A}(\mathbf{x})} m(f, \mathbf{x}', y_f(\mathbf{x}))$; while the weak augmentation preserves the classification $y_f(\mathbf{x}^w) = y_f(\mathbf{x})$.*

*Then, for any $f \in \mathcal{F}'$ during learning, minimizing the cross-entropy loss between $\texttt{softmax}(f(\mathbf{x}^s))$ and the one-hot label $\overrightarrow{y_f}(\mathbf{x}^w)$ is equivalent to enforcing a large enough margin $m(f, \mathbf{x}^s, y_f(\mathbf{x})) \geq \tau$, whereas $m(f, \mathbf{x}', y_f(\mathbf{x})) \geq m(f, \mathbf{x}^s, y_f(\mathbf{x})) \geq \tau$ for all $\mathbf{x}' \in \mathcal{A}(\mathbf{x})$ by construction implies $m_{\mathcal{A}}(f, \mathbf{x}) \geq \tau$ as required by the constraints in Equation (8).*

## E.3 Proof of Theorem 6.2

*Proof of Theorem 6.2.* We first recall from Lemma E.1 that $c$-expansion implies $\left( \frac{2\nu(f)}{c-1}, 2\nu(f) \right)$-constant expansion (Definition E.1) for any $f \in \mathcal{F}$. Therefore, it is sufficient to show that with $(q, 2\nu(f))$-constant expansion (*i.e.*, $q = \frac{2\nu(f)}{c-1}$),

$$P(M(f)) \leq \max\{q, 2\nu(f)\} \quad \forall f \in \mathcal{F}^\tau_{\mathbf{X}^u}. \tag{22}$$

For any $f \in \mathcal{F}^\tau_{\mathbf{X}^u}$, we notice that Equation (22) is automatically satisfied when $P(M(f)) < q$. Otherwise, when $P(M(f)) \geq q$, then along with $P(S \cap \mathcal{X}_k) \leq P(\mathcal{X}_k)/2$ for all $k \in [K]$, $(q, 2\nu(f))$-constant expansion implies

$$P(\mathrm{NB}(M(f))) > \min\{P(M(f)), 2\nu(f)\} + P(M(f)). \tag{23}$$

Meanwhile, we observe that by union bound,

$$P(\mathrm{NB}(M(f)) \setminus M(f)) \leq 2\mathbb{P}[\exists\, \mathbf{x}' \in \mathcal{A}(\mathbf{x}) \text{ s.t. } y_f(\mathbf{x}) \neq y_f(\mathbf{x}')] = 2\nu(f) \tag{24}$$

because for any $\mathbf{x} \in \mathrm{NB}(M(f)) \setminus M(f)$, there exists $\mathbf{x}' \in M(f)$ such that one can find some $\mathbf{x}'' \in \mathcal{A}(\mathbf{x}) \cap \mathcal{A}(\mathbf{x}')$; and therefore exactly one of the follows must hold

(i)  if $\mathbf{x}'' \in M(f)$, then $\widetilde{y}_f(\mathbf{x}'') \neq y_*(\mathbf{x}) = \widetilde{y}_f(\mathbf{x})$ implies $y_f(\mathbf{x}'') \neq y_f(\mathbf{x})$ for $\mathbf{x}'' \in \mathcal{A}(\mathbf{x})$;
(ii) if $\mathbf{x}'' \notin M(f)$, then $\widetilde{y}_f(\mathbf{x}'') = y_*(\mathbf{x}) \neq \widetilde{y}_f(\mathbf{x}')$ implies $y_f(\mathbf{x}'') \neq y_f(\mathbf{x}')$ for $\mathbf{x}'' \in \mathcal{A}(\mathbf{x}')$.

Leveraging Equation (23) and Equation (24), we have

$$\begin{aligned} P(M(f)) &\geq P(\mathrm{NB}(M(f))) - P(\mathrm{NB}(M(f)) \setminus M(f)) \\ &> \min\{P(M(f)), 2\nu(f)\} + P(M(f)) - 2\nu(f), \end{aligned}$$

which leads to $\min\{P(M(f)), 2\nu(f)\} < 2\nu(f)$ and implies $P(M(f)) < 2\nu(f)$. $\qquad\square$

# F Technical Lemmas

Here, we briefly review the standard Rademacher-complexity-based generalization analysis based on McDiarmid's inequality and a classical symmetrization argument Wainwright [2019], Bartlett and Mendelson [2003].

**Lemma F.1** (Generalization guarantee with Rademacher complexity). *Given a distribution over a set $\mathcal{Z}$[13], we consider a set of i.i.d. samples $\{\mathbf{z}_i\}_{i \in [n]} \sim \mathcal{Z}^n$ with $\mathbf{Z} = [\mathbf{z}_1, \ldots, \mathbf{z}_n]$ and a $B$-bounded function class $\mathcal{H} = \{h : \mathcal{Z} \to \mathbb{R} \mid |h(\mathbf{z}) - h(\mathbf{z}')| \leq B \ \forall\, \mathbf{z}, \mathbf{z}' \in \mathcal{Z}\}$. By denoting*

$$H(h) \triangleq \mathbb{E}_{\mathbf{z} \sim \mathcal{Z}}[h(\mathbf{z})] \quad and \quad \widehat{H}_{\mathbf{Z}}(h) \triangleq \frac{1}{n} \sum_{i=1}^{n} h(\mathbf{z}_i) \quad s.t. \quad \mathbb{E}_{\mathbf{Z} \sim \mathcal{Z}^n}\left[ \widehat{H}_{\mathbf{Z}}(h) \right] = H(h),$$

---

[13]Without ambiguity, we overload the notation $\mathcal{Z}$ as the set as well as the distribution over the set.

*then for $h^* \in \arg\min_{h\in\mathcal{H}} H(h)$ and $\widehat{h} \in \arg\min_{h\in\mathcal{H}} \widehat{H}_{\mathbf{Z}}(h)$, we have that with probability at least $1 - \delta/2$ over $\mathbf{Z}$,*

$$H(\widehat{h}) - H(h^*) \le 4\mathfrak{R}_n\left(\mathcal{H}\right) + \sqrt{\frac{2B^2 \log(4/\delta)}{n}}.$$

*where $\mathfrak{R}_n\left(\mathcal{H}\right)$ is the Rademacher complexity of $\mathcal{H}$,*

$$\mathfrak{R}_n\left(\mathcal{H}\right) \triangleq \mathbb{E}_{\substack{\mathbf{Z}\sim\mathcal{Z}^n \\ \boldsymbol{\rho}\sim\mathrm{Rad}^n}} \left[\sup_{h\in\mathcal{H}} \frac{1}{n}\sum_{i=1}^n \rho_i \cdot h(\mathbf{z}_i)\right], \quad \mathrm{Rad}(\rho) = \begin{cases} 1/2, & \rho = -1 \\ 1/2, & \rho = 1 \\ 0, & \rho \ne \pm 1 \end{cases}.$$

*Proof of Lemma F.1.* Let $g\left(\mathbf{z}_1, \ldots, \mathbf{z}_n\right) \triangleq g(\mathbf{Z}) \triangleq \sup_{h\in\mathcal{H}} H(h) - \widehat{H}_{\mathbf{Z}}(h)$. Then $h \in \mathcal{H}$ being $B$-bounded implies that for any $\mathbf{z} \sim \mathcal{Z}$ and $i \in [n]$,

$$|g(\mathbf{z}_1, \ldots, \mathbf{z}_i, \ldots, \mathbf{z}_n) - g(\mathbf{z}_1, \ldots, \mathbf{z}, \ldots, \mathbf{z}_n)| \le \frac{B}{n}.$$

Therefore, by McDiarmid's inequality [Bartlett and Mendelson, 2003],

$$\mathbb{P}\left[g(\mathbf{Z}) \ge \mathbb{E}[g(\mathbf{Z})] + t\right] \le \exp\left(-\frac{2nt^2}{B^2}\right),$$

where $\exp\left(-\frac{2nt^2}{B^2}\right) \le \frac{\delta}{4}$ when $t \ge \sqrt{\frac{B^2\log(4/\delta)}{2n}}$.

Meanwhile, by a classical symmetrization argument (e.g., proof of Wainwright [2019] Theorem 4.10), we can bound the expectation $\mathbb{E}[g(\mathbf{Z})]$: for an independent sample set $\mathbf{Z}' \sim \mathcal{Z}^n$ that is also independent of $\mathbf{Z}$,

$$\begin{aligned}
\mathbb{E}\left[g(\mathbf{Z})\right] &= \mathbb{E}_{\mathbf{Z}}\left[\sup_{h\in\mathcal{H}} H(h) - \widehat{H}_{\mathbf{Z}}(h)\right] \\
&= \mathbb{E}_{\mathbf{Z}}\left[\sup_{h\in\mathcal{H}} \mathbb{E}_{\mathbf{Z}'}\left[\widehat{H}_{\mathbf{Z}'}(h) - \widehat{H}_{\mathbf{Z}}(h) \,\Big|\, \mathbf{Z}\right]\right] \\
&\le \mathbb{E}_{\mathbf{Z}}\left[\mathbb{E}_{\mathbf{Z}'}\left[\sup_{h\in\mathcal{H}} \widehat{H}_{\mathbf{Z}'}(h) - \widehat{H}_{\mathbf{Z}}(h) \,\Big|\, \mathbf{Z}\right]\right] \\
&= \mathbb{E}_{(\mathbf{Z},\mathbf{Z}')}\left[\sup_{h\in\mathcal{H}} \widehat{H}_{\mathbf{Z}'}(h) - \widehat{H}_{\mathbf{Z}}(h)\right],
\end{aligned}$$

where in the last line we have used the law of iterated conditional expectation. Then, with Rademacher random variables $\boldsymbol{\rho} = (\rho_1, \ldots, \rho_n) \sim \mathrm{Rad}^n$, we have

$$\begin{aligned}
\mathbb{E}\left[g(\mathbf{Z})\right] &\le \mathbb{E}_{(\mathbf{Z},\mathbf{Z}',\boldsymbol{\rho})}\left[\sup_{h\in\mathcal{H}} \frac{1}{n}\sum_{i=1}^n \rho_i \cdot (h(\mathbf{z}'_i) - h(\mathbf{z}_i))\right] \\
&\le 2\mathbb{E}_{(\mathbf{Z},\boldsymbol{\rho})}\left[\sup_{h\in\mathcal{H}} \frac{1}{n}\sum_{i=1}^n \rho_i \cdot h(\mathbf{z}_i)\right] \\
&\le 2\mathfrak{R}_n\left(\mathcal{H}\right).
\end{aligned}$$

Therefore, with probability at least $1 - \delta/4$ over $\mathbf{Z}$, we have

$$\sup_{h\in\mathcal{H}} H(h) - \widehat{H}_{\mathbf{Z}}(h) \le 2\mathfrak{R}_n\left(\mathcal{H}\right) + \sqrt{\frac{B^2\log(4/\delta)}{2n}},$$

while the same tail bound holds for $\sup_{h\in\mathcal{H}} -H(h) + \widehat{H}_{\mathbf{Z}}(h)$ via an analogous argument.

By recalling the definition of $\widehat{h}$ and $h^*$, we can decompose the excess risk $H(\widehat{h}) - H(h^*)$ as

$$\begin{aligned}
H(\widehat{h}) - H(h^*) &= \left(H(\widehat{h}) - \widehat{H}_{\mathbf{Z}}(\widehat{h})\right) + \left(\widehat{H}_{\mathbf{Z}}(\widehat{h}) - \widehat{H}_{\mathbf{Z}}(h^*)\right) + \left(\widehat{H}_{\mathbf{Z}}(h^*) - H(h^*)\right) \\
&\le \left(H(\widehat{h}) - \widehat{H}_{\mathbf{Z}}(\widehat{h})\right) + \left(\widehat{H}_{\mathbf{Z}}(h^*) - H(h^*)\right)
\end{aligned}$$

where $\widehat{H}_{\mathbf{Z}}(\widehat{h}) - \widehat{H}_{\mathbf{Z}}(h^*) \leq 0$ by the basic inequality. With both $\widehat{h}, h^* \in \mathcal{H}$, we have

$$H(\widehat{h}) - H(h^*) \leq 2 \sup_{h \in \mathcal{H}} \left| H(h) - \widehat{H}_{\mathbf{Z}}(h) \right|,$$

and therefore

$$\mathbb{P}\left[ H(\widehat{h}) - H(h^*) \geq t \right] \leq \mathbb{P}\left[ \sup_{h \in \mathcal{H}} H(h) - \widehat{H}_{\mathbf{Z}}(h) \geq \frac{t}{2} \right] + \mathbb{P}\left[ \sup_{h \in \mathcal{H}} -H(h) + \widehat{H}_{\mathbf{Z}}(h) \geq \frac{t}{2} \right].$$

Overall, with probability at least $1 - \delta/2$ over $\mathbf{Z}$,

$$H(\widehat{h}) - H(h^*) \leq 4\mathfrak{R}_n\left(\mathcal{H}\right) + \sqrt{\frac{2B^2 \log(4/\delta)}{n}}.$$

$\square$

