# OpenReview forum: "Cluster-aware Semi-supervised Learning: Relational Knowledge Distillation Provably Learns Clustering"
_NeurIPS.cc/2023/Conference — NeurIPS 2023 poster_

### Official Review · Reviewer_T9nK · 2023-07-01

**Soundness:** 3 good
**Presentation:** 3 good
**Contribution:** 3 good
**Rating:** 6
**Confidence:** 3

**Summary:**

This paper studies relational knowledge distillation (RKD) on semi-supervised learning setting. This is different from the previous theoretical results on knowledge distillation, which mostly focus on feature matching (not relational). They introduce a new notion called low clustering error, which quantifies the difference betweeen the predicted and ground truth clusterings, and show that RKD can provably produce low clustering error. They also have a few other results, including the label efficiency theorm on a cluster-aware semi-supervised learning framework with low clustering errors, and unifying data augmentation consistency regularization with RKD, showing that consistency reuglarization focuses on the local perspective, while RKD improves over the global perspective.

**Strengths:**

Originality: Since we already have seen a few papers on connecting spectral clustering with contrastive learning or other learning methods, the spectral clustering part of this paper is not particularly surprising. However, I do like the relational part, which focuses on the relationships between the features, rather than the feature matching itself. I think this is a very interesting and important direction to pursue. Moreover, the authors also investigate the benefit of clustering awareness in the semi-supervised learning scenario, which is novel.

Quality: I think this is a well written paper with rigorous definitions, theorems and proofs.

Clarity: Since this is a theory paper, I will not say it is straightforward to read. However, I think the authors did a very good job in the introduction to demonstrate the main contributions of this paper, and present the technical theorems and assumptions in a structured way. Therefore I would say this paper is well written.

Significance: I think this paper is indeed interesting, because it provides an interesting connection between relational knowledge distillation learning and spectral clustering, and also provide theoretical characterization on various aspects.



**Weaknesses:**

It is a bit difficult for me to understand the intuition of Assumption 4.2, and the toy example after assumption 4.2 is too special, which does not explain much about the complicated definitions. It would be nice to further elaborate on the intuitions, and perhaps give a non-trivial example of this assumption, and why this assumption is a mild assumption.

Personally I like Theorem 4.1 better, and I do not feel it is really necessary to consider the limited unlabeled samples setting (because usually we may assume the unlabeled samples are cheap). Therefore, it might improve the presentation of this paper, if the author can add more motivations on Sec 4.2. (if there is no such motivation, it's also fine, I think it is a minor spot.)

The current connections between different results are weak. Specifically, I cannot easily understand how the last DAC result connects with the previous two. I might be wrong, but I initially felt that the third result says the combination of DAC and RKD can improve the performance. However, the statement of Theorem 6.2 seems only contains DAC, not RKD. Therefore, it seems to me that the first and third results are in parallel: both of them are providing upper bounds of clustering error. Is it true?
-- if this is true, I think the presentation of the introduction should be updated, to avoid potential misconceptions.



**Questions:**

I do not have other questions.

**Limitations:**

Not applicable.

---

> ### Author Rebuttal · Authors · 2023-08-08
>
> We would like to thank the reviewer for appreciating the novelty and presentation of this work, as well as for the insightful comments and questions.
>
> __[Intuition of Ass 4.2]__ Ass 4.2 can be intuitively summarized as assumptions on the __boundedness $\beta$__ and __reasonably large margins $\gamma$__ of the prediction functions $f$. In other words, Ass 4.2 pivots $f$ down to a reasonable prediction function.
>
> In the submitted version, we tailor Ass 4.2 to provide better upper bounds for the clustering error. Thanks to the reviewer’s feedback, we realize that this can lead to non-standard descriptions. We will include further clarifications on these in the revision:
> * __Skeleton subset__: Given a prediction function $f: \mathcal{X} \to \mathbb{R}^K$ for K-class classification, the skeleton subset $S = [s_1; \cdots; s_K]$ can be intuitively viewed as $K$ samples where $f$ makes the __“most confident” prediction__ in each class, i.e., $f(s_k)_k = \max_x f(x)_k$. For $f$ to be a reasonable prediction function, we assume
> 1. $f$ indeed assigns $s_k$ to class $k$ and
> 2. $f(S) \in \mathbb{R}^{K \times K}$ is full rank (i.e., $f(s_k)$ for different $k \in [K]$ are linearly independent).
> The latter is a milder assumption than other alternatives like “$f(S)$ is strictly diagonally dominant” (which is a sufficient condition for being full rank while generally holds for reasonable prediction functions). Intuitively, this enforces each prediction $f(s_k)$ in the skeleton subset to have a dominating k-th entry $f(s_k)_k$.
> * __Boundedness $\beta$__: $\sigma_1(f(S))\le\beta$ is again a milder version of some arguably more common forms of boundedness, e.g., $|f(x)_k|<\beta/K$ for all $x \in \mathcal{X}$, $k \in [K]$ (a sufficient condition).
> * __Margin $\gamma$__: This ensures that for each class $k \in [K]$, $f$ can easily distinguish its “most confident” prediction $s_k$ from those samples predicted to lie in other classes by at least a margin $\gamma$. Notice that this is a slightly simplified description–but a stronger assumption–than what appeared in Ass 4.2.
>
> It is worth highlighting that these assumptions __cannot be guaranteed with RKD alone__ but are generally __satisfied with additional supervision/regularization__. Following the toy example after Ass 4.2, we refer the reader to __Example C.1__, which demonstrates such __pitfall of RKD alone (violating Ass 4.2)__ and how __additional supervision/regularization can help__. This also provides an explanation of why, RKD is never applied in practice as a standalone algorithm for end-to-end learning but is commonly coupled with supervision and/or standard KD. We will further emphasize this critical point in the revision.
>
> __[Importance of the limited unlabeled sample setting]__ While we agree with the reviewer that considering the limited unlabeled sample setting inevitably leads to more complicated results, we kindly argue that such effort is worthwhile beyond theoretical completeness. Although the unlabeled samples are oftentimes much more accessible than the labeled ones, in practice, we generally only have __finite (limited) unlabeled samples__ but not access to the __unlabeled population__ (what is assumed in Thm 4.1).
>
> More importantly, analyzing the limited unlabeled sample setting provides insight into the **unlabeled sample complexity**—how many unlabeled samples are needed to achieve a low clustering error. In particular, Thm 4.2 and 4.3 imply that the unlabeled sample complexity is dominated by the Rademacher complexity of the function class. Intuitively, larger neural networks will then require more unlabeled samples to sufficiently learn the clustering structure.
>
> __[Connection and difference between RKD and DAC]__ At a high level, RKD, cluster-aware SSL, and DAC are motivated and connected through the three research questions raised in the introduction:
> * In Sec 4, we show that RKD leads to low clustering error via spectral clustering.
> * In Sec 5, we show how such cluster awareness brings label efficiency through a more general cluster-aware SSL framework.
> * As the reviewer mentioned, __clustering is a generic notion__ that is widely studied from various perspectives. Then, it is natural to ask __what is special about spectral clustering as learned by RKD?__ In Sec 6, the analysis of DAC is leveraged as __another example of cluster-aware SSL that facilitates clustering from a different perspective complementary to RKD__. By __comparing the distinct mechanisms of RKD and DAC in providing upper bounds for the same notion of clustering error__, we show that __DAC learns “locally”__ through expansion of neighborhoods, whereas __RKD learns “globally”__ through spectral clustering.
>
> The reviewer is right about that Thm 6.2 connects low DAC error to low clustering error and does not involve RKD. However, we kindly point out that the key message conveyed in Sec 6 is not the upper bound on clustering error itself, __but the different mechanism that leads to clustering—the “local” expansion of neighborhoods through data augmentation (Def. 6.1 and 6.2)__. Based on such distinct mechanisms, we further discuss the complementary perspectives of RKD and DAC in __Remark 6.1__.
>
> We will try to further clarify these connections and address the potential confusion in the revision.
>
> As a minor remark from the technical perspective, Thm 6.2 does not involve RKD because the analysis on DAC is conducted in a different setting from that for RKD (e.g., data population $\mathcal{X}$ is a set in Sec 6 without a graph structure), exactly because of their distinct perspectives of learning clustering. Although it is possible to combine both settings and derive unified bounds, this would lead to unnecessary complications for our goal of demonstrating the complementary perspectives of RKD and DAC.
>
> We hope this will help address the confusion and concerns raised in the review, and if so, we would genuinely appreciate a re-evaluation accordingly.

---

> > ### Comment · Reviewer_T9nK · 2023-08-12
> >
> > Thank you for your rebuttal. I have adjusted my score accordingly.
> > However, I do hope that you can update your introduction and your sections in a more organized way, as I see other reviewers have similar concerns.

---

> > > ### Author Response · Authors · 2023-08-15
> > > **Thanks for the response and feedback**
> > >
> > > We really appreciate the valuable feedback and will make sure to incorporate corresponding clarifications in our revision. In particular, we will include the following:
> > > - We will further clarify the motivation of exemplifying DAC as another instance of cluster-aware SSL with a complementary perspective to RKD in both the introduction and the associated Sec 6 (as elaborated in our response to Reviewer T9nK “__[Connection and difference between RKD and DAC]__”).
> > > - We will provide more intuitions for Ass 4.2 and further highlight the associated example concerning the pitfall of RKD alone (as elaborated in our response to Reviewer T9nK “__[Intuition of Ass 4.2]__”).
> > > - We will update the experiments and the corresponding discussion in Appendix A (as presented in our general response).

---

### Official Review · Reviewer_APrr · 2023-07-06

**Soundness:** 3 good
**Presentation:** 3 good
**Contribution:** 3 good
**Rating:** 5
**Confidence:** 3

**Summary:**

This paper points out that RKD over a population-induced graph given by a teacher model leads to low clustering error (Spectral Clustering), which provably benefits to semi-supervised learning and other downstream tasks. Consequently, it provides a theoretical analysis of relational knowledge distillation (RKD) in the semi-supervised classification setting from a clustering perspective.

**Strengths:**

This paper mainly focuses on expounding the close relationship between RKD and Spectral Clustering from a theoretic perspective, which definitely contributes to the increase of attention on RKD. According to this seminal work, RKD can be regarded as a principle choice to extract knowledge from a pre-trained large model.

**Weaknesses:**

The weakness of this paper is obvious: The experimental part is too weak. There is no empirical study in the main text, and the experimental results in the appendix are limited. In addition, in some cases, RKD even deteriorates the Top-1 performance. The authors provided no explanation on these cases. Besides, CIFAR-10 and CIFAR-100 are not quite challenging datasets, and they are too similar. It is suggested conducting experiments on the large-scale datasets to better demonstrate the effectiveness of the proposed method.

**Questions:**

My principle concern is the inadequate empirical study. As mentioned above, there are mainly two issues should be addressed. First, in the current results (Table 1 and Table 2), why the performance of the proposed approach cannot consistently outperform the baseline? Second, what is the performance of the proposed method on other large-scale datasets? If these issues can be resolved, I will recommend this work.

**Limitations:**

I agree with the limitations provided in the last section. Besides, please see the Weaknesses and Questions parts for my concern. I hope the review is helpful for this work.

---

> ### Author Rebuttal · Authors · 2023-08-08
>
> Many thanks for appreciating our novelty, as well as for raising constructive suggestions and questions concerning our experiments.
>
> Nevertheless, from a high-level perspective, we would like to kindly point out that this work is __a novel theoretical study on a family of existing methods__ that is empirically well-known—RKD and cluster-aware semi-supervised learning. Therefore, __the goal of our experiments is to ground the insights unveiled by our theoretical results__, instead of demonstrating the power of a novel algorithm on a broad spectrum of large-scale benchmarks.
>
> Moreover, as we briefly touched on in the introduction (line 27-31), an essential virtue of (data-free) knowledge distillation in general is to leverage the power of existing teacher models for more affordable learning in both computational and sample complexities. Therefore when designing the experiments, with limited computational resources, we purposefully put our focus on fully investigating the more relevant factors to our theory (e.g., the unlabeled sample sizes and the choices of data augmentations, as explained in the next paragraphs).
>
> __[Consistency of performance I: when and how RKD helps improve DAC]__ Thanks for the great question. This leads to the interesting point on when and how RKD helps improve DAC. The answer can be provided from both the analysis and experiment perspectives as follows.
>
> From the __analysis perspective__, as elaborated in __Remark 6.1__, DAC learns a “local” clustering structure through the expansion of neighborhoods characterized by the data augmentations. This is complementary to RKD which learns the spectral clustering from a “global” perspective. Therefore, RKD helps improve DAC when the data augmentations on the unlabeled samples fail to reveal the “local” clustering structure:
> 1. when the data augmentations are not expansive enough (characterized by the $c$-expansion property in Def. 6.1), and/or
> 2. when there are insufficient numbers of unlabeled samples.
> That is, in our experiments, RKD brings more improvements to FixMatch when the augmentation strength is weaker or the unlabeled sample size $N$ is smaller, whereas when the data augmentations are strong and $N$ is large, the gap between FixMatch and FixMatch+RKD may not be noticeable.
>
> From the __experiment perspective__, the choice of teacher model in practice becomes another critical factor for the performance of RKD. Recall that in theory, we assume the access to a teacher model that reveals the underlying structure of the data, whose accuracy is characterized by a constant $\alpha$ in Def. 4.1 such that $\alpha \ll 1$ for a good teacher model. However, in practice, different pretrained teacher models can lead to different $\alpha$. For example, in our experiments, the teacher model for CIFAR-10 pretrained with supervised ERM on CIFAR-10 and that for CIFAR-100 pretrained with unsupervised contrastive learning (SwAV) on ImageNet can have different $\alpha$’s that lead to different performance.
>
> __[Consistency of performance II: experiments with multiple random seeds]__ Thanks to the reviewer's suggestion, we ran more experiments with multiple random seeds to improve the reliability of our results. Please refer to the general response for details.
>
> We hope this will help address the concerns raised in the review, and if so, we would genuinely appreciate a re-evaluation accordingly.

---

> > ### Comment · Reviewer_APrr · 2023-08-13
> > **Please provide more explanation on the experimental results that do not quite match your claim**
> >
> > Thanks for offering the additional experimental results and more explanation for my questions. After reading these, most of my concerns have been resolved. However, I still have problems in understanding your rebuttal: RKD brings more improvements to FixMatch when the augmentation strength is weaker or the unlabeled sample size N is smaller, whereas when the data augmentations are strong and N is large, the gap between FixMatch and FixMatch+RKD may not be noticeable. According to your new empirical results, I noticed that this conclusion may not be true, especially when considering the **unlabeled sample size**. Besides, I found in your experimental results, the higher augmentations and more unlabeled data do not consistently lead to better results (e.g., in the Uniform&Weak case on CIFAR10 and the StochasticGreedy&50000 case on CIFAR100), which is a little strange to me. Do you have any further explanation on this issue?

---

> > > ### Author Response · Authors · 2023-08-15
> > > **Thanks for the response**
> > >
> > > Many thanks for the follow-up questions, as well as for the time and effort.
> > >
> > > Before diving into the detailed explanations for specific observations, we would like to kindly highlight the key message of our experiments: __The incorporation of RKD with the “global” perspective via spectral clustering brings improvement to DAC with the “local” perspective.__ In the updated experiments with multiple random seeds, we observe that this generally holds for different combinations of augmentation strengths and unlabeled sample sizes.
> > >
> > > Moreover, our theoretical analysis implies that the “local” perspective of DAC tends to be deficient while adding the “global” perspective of RKD can be helpful when data augmentations are not suitable (e.g., not “expansive” enough in terms of satisfying $c$-expansion property in Def 6.1 with a small $c \approx 1$) and/or when the unlabeled samples are limited. As an illustrative (but arguably oversimplified) intuition, RKD tends to bring observable improvement to DAC with weak augmentations and/or insufficient unlabeled samples. This explains why, for example in the updated experiments with $N=50000$ and high/medium augmentation strengths, the gaps between FixMatch v.s. FixMatch+RKD are not significant.
> > >
> > > __[When and how, but not how much, RKD can help]__ Although our analysis casts insight on when and how incorporating RKD brings benefits, inferring the magnitude of such improvement when scaling with the augmentation strength and unlabeled sample size is a much trickier problem in practice.
> > >
> > > In particular, for the unlabeled sample size, $N$, recall that the clustering error bounds for both RKD and DAC (Thm 4.2, 4.3, and 6.1, 6.2) depend on $N$. In other words, with insufficient unlabeled samples, the clustering accuracies of both RKD and DAC are compromised. Although their complementary perspectives suggest the advantage of RKD+DAC over DAC alone (as we observed in the experiments), such improvement may not monotonically increase as $N$ decreases. This depends on how much the clustering errors of RKD v.s. DAC are affected by decreasing $N$, which is affected not only by the algorithms themselves but also by the randomness in the training data and data augmentations.
> > >
> > > We will clarify this point in the discussion on experiments in the revision.
> > >
> > > __[Stronger augmentations/slightly more unlabeled samples do not necessarily lead to better performance]__ Despite being intuitive, we kindly point out the subtle but critical difference between “strong” and “suitable” data augmentations. Recall that in Def 6.1, data augmentations are required to be class-invariant (i.e., preserving semantic information) in addition to the expansion property. Intuitively, an overly high augmentation strength can distort the semantic information in the data and therefore hurt the performance.
> > >
> > > For the unlabeled sample size, although the accuracies of both FixMatch and FixMatch+RKD generally increase as $N$ increases, the results can fluctuate within the standard deviations (as the reviewer mentioned) since the gap between $N=40000$ and $N=50000$ in our experiments is relatively small for models like WideResNet.

---

> > > > ### Comment · Reviewer_APrr · 2023-08-21
> > > >
> > > > Thanks for the reply. My concerns are partially addressed, and I will retain my score.

---

### Official Review · Reviewer_mtgh · 2023-07-08

**Soundness:** 3 good
**Presentation:** 2 fair
**Contribution:** 3 good
**Rating:** 6
**Confidence:** 3

**Summary:**

The paper provides a comprehensive theoretical proof of the relationship between RKD and clustering from the perspective of spectral clustering. The theoretical proof presented in the paper is detailed and reasonable. Firstly, RKD is regarded as a spectral clustering problem of the teacher model, and then the introduced clustering error is used to prove that RKD leads to low clustering error. Finally, the paper presents the sampling complexity bound of RKD in the case of limited unlabeled samples.

**Strengths:**

1.The theoretical proof presented in the paper is detailed and reasonable.
2. Label efficiency of cluster-aware SSL is demonstrated as well.


**Weaknesses:**

1. It appears that the author attempts to convey numerous viewpoints in this paper, resulting in a somewhat confused logical structure. As a result, the paper is somewhat difficult to read and understand, possibly due to my unfamiliarity with the writing style of purely theoretical papers.
2. While the author's arguments are generally reasonable, there seems to be a lack of strong correspondence between the presented problem and the method of reasoning. It would be beneficial to understand the author's motivations and thought process behind exploring RKD at a theoretical level, particularly in relation to demonstrating its label efficiency in the SSL setting.

**Questions:**

As I am less familiar with purely theoretical work, I find that in Section 7, the author continues to discuss the theoretical analysis of RKD in different settings. While providing relevant theoretical proofs is important based on previous empirical work, it would be beneficial to consider adding a discussion section that explores potential new directions for practical applications.

**Limitations:**

1. The research presented in the paper holds significant theoretical value. It would be beneficial to enhance the overall clarity and organization of the paper's structure. This will improve the readability and friendliness of the paper for fellow researchers in the field.
2. In the field of machine learning, it is highly recommended to conduct necessary experiments to strengthen the scientific rigor of the study.

---

> ### Author Rebuttal · Authors · 2023-08-08
>
> We would like to first thank the reviewer for appreciating the theoretical contribution of this work, as well as for bringing up constructive questions and suggestions in the review. On the detailed concerns:
>
> __[Motivations and improvements on presentation]__
> We understand and value the reviewer’s concern and feedback on the accessibility of our presentation to the audience who are less familiar with theory works. In the following paragraphs, we clarify the motivations and outline of this work. Meanwhile, we will try to improve the presentation and accessibility of the paper in the revision.
> 1. __RKD learns spectral clustering__: Clustering is a well-established and fundamental idea in machine learning that suggests that data can be more efficiently learned from via the inherent geometric structure of the available data set of points. Inputs that are more “similar” to each other (in the appropriate sense of similarity for the machine learning task at hand) should have “similar” outputs. We establish the connection to RKD by considering that the teacher model, which has learned these inherent similarities, imparts knowledge regarding the clustering structure over the population data, $\mathcal{X}$ to the student model and its observed empirical data, $\mathbf{X}^u$. Viewing clustering from a graph perspective, the knowledge imparted from teacher to student can be interpreted as spectral clustering on an underlying population-induced graph.
> 2. __Cluster-awareness of RKD leads to better label efficiency__: One of the most celebrated virtues of (data-free) knowledge distillation in general, including RKD, is to leverage the power of existing pretrained models to __enable more efficient learning__ in terms of both the computation and sample complexity. With the established connection between RKD and spectral clustering, we then demonstrate the label efficiency achieved by RKD through a more general framework of “cluster-aware SSL”.
> 3. __Different perspectives of cluster-awareness: DAC v.s. RKD__: Given that the notion of clustering is quite broad, we ask a natural question: __what is special about spectral clustering as learned by RKD?__ To answer this question, we further investigate data augmentation consistency (DAC) regularization, another example of cluster-aware SSL that learns clustering from a perspective that is complementary to RKD.  By __comparing the distinct mechanisms of RKD and DAC in providing upper bounds for the same notion of clustering error__, we show that __DAC learns clustering “locally”__ through “expansion of neighborhoods”, whereas __RKD learns clustering “globally”__ through spectral clustering.
>
> __[Future works from an empirical perspective]__ There are two immediate insights from our theoretical analysis. First, by unveiling the mechanism of RKD as a form of spectral clustering from a “global” perspective, we illustrate __when and how we can incorporate RKD efficiently in practice__. For example, semi-supervised learning with DAC encourages clustering from a “local” perspective (Sec. 6), but when the data augmentations are not strong (expansive as defined in Def. 6.1) enough to reveal the clustering structure through the local connectivity/overlap of neighborhoods (cf. the “weak” augmentations in Appendix A), the additional “global” perspective offered by RKD generally helps the student model better capture the underlying clustering structure and achieve better accuracy. Our experiments in Appendix A empirically demonstrate this.
>
> Second, through the analysis of RKD as spectral clustering, we identify a pitfall of applying RKD alone in end-to-end learning (Example C.1) that violates the boundedness and margin assumptions. This provides __a theoretical explanation for the practical wisdom that RKD is generally used to reinforce standard KD__, instead of as a standalone end-to-end learning algorithm. Through the lenses of “global” and “local” perspectives of RKD and DAC, along with the experiments in Appendix A, we show that __RKD can be leveraged not only to boost standard KD but also to improve upon DAC in semi-supervised learning__.
>
> From a broader perspective and motivated by the improvement brought by RKD to DAC, especially in the data-scarce setting, it would be particularly interesting to __design practical (semi-supervised) learning algorithms that leverage both the local and global perspectives on the clustering structure__.
>
> __[(More) experiments]__ Due to the page limit, we defer the experiments to Appendix A. For various label-rate (i.e., amounts of labeled vs unlabeled data) and augmentation strength levels, we evaluate the utility of RKD for improving a baseline SSL method. We use the recent state-of-the-art FixMatch as our baseline SSL method, which can be considered as a DAC method in our theoretical framework. Furthermore, for each of the datasets that we include in our experimental evaluation, we consider the effect of the selection of the limited amount of labeled data on the performance in each setting. We find that RKD does indeed help to improve student model performance, with the greatest gains in performance in especially data-scarce settings wherein smaller amounts of labeled and unlabeled data are available to the student model. This reflects the importance of the global perspective of the clustering structure offered by RKD for data-scarce applications of data-free knowledge distillation (where the student model does not have access to the data that the teacher model was trained on).
>
> While we think that more empirical evaluation would be very useful, we kindly suggest that our main contributions in this work lie in the important steps we have taken to establish a more rigorous theoretical understanding of the empirical success of RKD.
>
> We hope this will help address the confusion and concerns raised in the review, and if so, we would truly appreciate a re-evaluation accordingly.

---

### Author Rebuttal · Authors · 2023-08-08

First, we would like to thank all the reviewers for their efforts and valuable suggestions. We will share an update on our experiments (in Appendix A) here globally while addressing individual questions and concerns posed in the reviews separately.

__[Additional experiments with multiple random seeds]__
Thanks to the suggestion of Reviewer APrr, we ran more experiments with multiple random seeds to improve the reliability of our results. We will update the following results in the revision: (Here, "High", "Medium", and "Weak" refer to the augmentation strength described in Appendix A.)

**Table 1: Top-1 accuracy on CIFAR-10.**
| Label selection | Sample sizes $(n, N)$ | SSL algorithm | **High** | **Medium** | **Weak** |
|:---------------:|:----------------------:|:-------------:|:--------:|:----------:|:--------:|
| **Uniform**     | (40, 50000)            | FixMatch      | 89.47 ± 3.83 | 87.89 ± 1.43 | 81.36 ± 1.41 |
|                 |                        | FixMatch + RKD| **89.88 ± 0.41** | **89.10 ± 1.14** | **84.61 ± 1.23** |
|                 | (40, 40000)            | FixMatch      | 87.60 ± 1.21 | 85.30 ± 1.35 | 83.74 ± 3.07 |
|                 |                        | FixMatch + RKD| **90.19 ± 0.19** | **88.65 ± 1.81** | **86.68 ± 1.06** |
|                 | (40, 25000)            | FixMatch      | 81.73 ± 1.27 | 81.43 ± 2.98 | 76.84 ± 1.98 |
|                 |                        | FixMatch + RKD| **87.06 ± 2.04** | **87.24 ± 0.39** | **82.75 ± 2.35** |
| **StochasticGreedy** | (40, 50000)      | FixMatch      | 85.48 ± 1.67 | 89.39 ± 0.82 | 78.98 ± 0.52 |
|                 |                        | FixMatch + RKD| **91.19 ± 0.26** | **90.00 ± 0.64** | **87.11 ± 4.33** |
|                 | (40, 40000)            | FixMatch      | 84.51 ± 1.34 | 84.88 ± 6.68 | 70.70 ± 5.32 |
|                 |                        | FixMatch + RKD| **85.68 ± 1.19** | **88.31 ± 1.36** | **85.12 ± 3.49** |
|                 | (40, 25000)            | FixMatch      | 79.74 ± 8.02 | 73.97 ± 1.98 | 73.86 ± 3.29 |
|                 |                        | FixMatch + RKD| **83.72 ± 4.73** | **82.09 ± 3.27** | **80.65 ± 3.67** |

**Table 2: Top-1 accuracy on CIFAR-100.**
| Label selection | Sample sizes $(n, N)$ | SSL algorithm | **High** | **Medium** | **Weak** |
|:---------------:|:----------------------:|:-------------:|:--------:|:----------:|:--------:|
| **Uniform**     | (400, 50000)           | FixMatch      | 46.85 ± 1.68 | 45.67 ± 0.51 | 45.84 ± 0.35 |
|                 |                        | FixMatch + RKD| **49.33 ± 0.42** | **47.97 ± 0.37** | **47.65 ± 0.17** |
|                 | (400, 40000)           | FixMatch      | 44.11 ± 0.51 | 42.98 ± 0.56 | 42.74 ± 0.42 |
|                 |                        | FixMatch + RKD| **46.24 ± 0.64** | **45.78 ± 0.94** | **45.13 ± 1.08** |
|                 | (400, 25000)           | FixMatch      | 36.60 ± 0.03 | 35.86 ± 0.51 | 33.59 ± 1.54 |
|                 |                        | FixMatch + RKD| **37.87 ± 0.98** | **36.69 ± 0.41** | **35.69 ± 1.51** |
| **StochasticGreedy** | (400, 50000)     | FixMatch      | 47.45 ± 0.30 | 48.64 ± 0.35 | 46.79 ± 1.05 |
|                 |                        | FixMatch + RKD| **50.41 ± 1.41** | **50.67 ± 1.62** | **49.09 ± 1.31** |
|                 | (400, 40000)           | FixMatch      | 45.93 ± 0.21 | 44.78 ± 1.51 | 44.62 ± 0.72 |
|                 |                        | FixMatch + RKD| **47.62 ± 0.67** | **47.28 ± 0.56** | **46.07 ± 0.86** |
|                 | (400, 25000)           | FixMatch      | 39.40 ± 0.31 | 39.01 ± 0.68 | 38.22 ± 0.42 |
|                 |                        | FixMatch + RKD| **40.88 ± 0.61** | **40.73 ± 0.46** | **39.20 ± 0.73** |

---

### Decision · Program_Chairs · 2023-09-21

**Decision:**

Accept (poster)

**Comment:**

The paper explores the issue of relational knowledge distillation. It frames the problem as that of Spectral clustering and draws theoretical insights. This paper should be of interest to Neurips audience.